# HW-GPT-Bench: Hardware-Aware Architecture Benchmark for Language Models

**Rhea Sanjay Sukthanker**[1]    **Arber Zela**[1]    **Benedikt Staffler**[3]

**Aaron Klein**[4]    **Lennart Purucker**[1]    **Jörg K. H. Franke**[1]    **Frank Hutter**[2,1]

[1]University of Freiburg, [2]ELLIS Institute Tübingen,
[3]Bosch Center for Artificial Intelligence (BCAI), Germany,
[4]ScaDS.AI, Leipzig,
{sukthank, zelaa, frankej, fh}@cs.uni-freiburg.de
benediktsebastian.staffler@de.bosch.com

## Abstract

The increasing size of language models necessitates a thorough analysis across multiple dimensions to assess trade-offs among crucial hardware metrics such as latency, energy consumption, GPU memory usage, and performance. Identifying optimal model configurations under specific hardware constraints is becoming essential but remains challenging due to the computational load of exhaustive training and evaluation on multiple devices. To address this, we introduce HW-GPT-Bench, a hardware-aware benchmark that utilizes surrogate predictions to approximate various hardware metrics across 13 devices of architectures in the GPT-2 family, with architectures containing up to 1.55B parameters. Our surrogates, via calibrated predictions and reliable uncertainty estimates, faithfully model the heteroscedastic noise inherent in the energy and latency measurements. To estimate perplexity, we employ weight-sharing techniques from Neural Architecture Search (NAS), inheriting pretrained weights from the largest GPT-2 model. Finally, we demonstrate the utility of HW-GPT-Bench by simulating optimization trajectories of various multi-objective optimization algorithms in just a few seconds.

## 1 Introduction

*Language models (LMs)* based on the transformer architectures [73] mark the current state-of-the-art [52] in most natural language understanding tasks, including text summarization, question-answering and language generation. This has led to a surge in research, with models [9, 13, 66] and training data [33, 44] growing in size. Consequently, inference costs have also risen significantly, making it often challenging to deploy these models in practice. For instance, ChatGPT utilizes over half a million kilowatt-hours of electricity daily, a consumption sufficient to handle approximately two hundred million requests. This energy usage is comparable to that of around 180,000 U.S. households, each consuming approximately twenty-nine kilowatt-hours [27].

There is a natural trade-off (Pareto frontier) between latency and performance of LLMs. While techniques such as KV-Cache optimization [75] and pruning [79, 86] have been used to improve inference efficiency, they do not explicitly balance performance and latency. Hence, discovering the inference-optimal frontier of language models is a multi-objective optimization problem [17], where we are interested in the Pareto set of architectures, i.e. the set of all dominating architectures that have a lower loss value on at least one objective and no higher loss value on any other objective.

Neural architecture search (NAS) [21] is a powerful framework to derive Pareto-optimal neural network architectures in an automated data-driven way. However, as pointed out by Wan et al. [76],

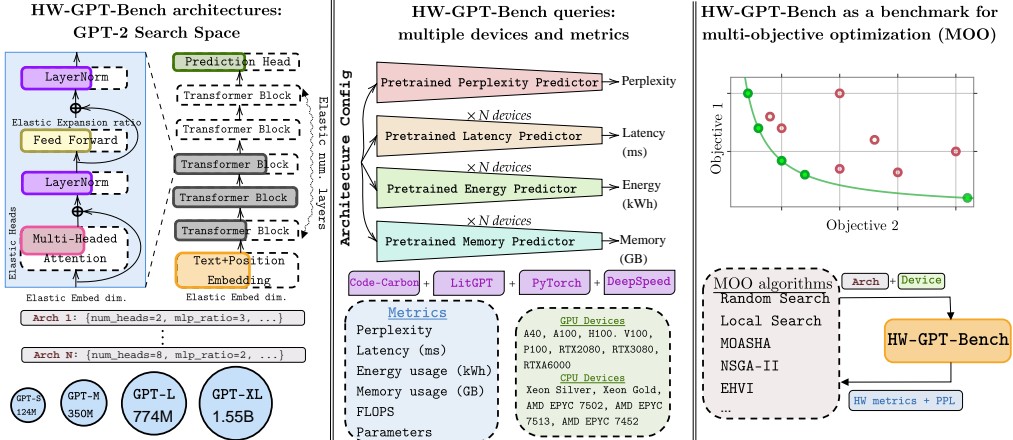

Figure 1: **HW-GPT-Bench Overview**. Illustration of the search space (*left*), hardware devices and metrics (*middle*) and multi-objective algorithms (*right*) used in the HW-GPT-Bench framework.

training a single language model can require millions of GPU hours, making the use of simple multi-objective NAS strategies, such as NSGA-II [45], that need to train multiple architectures from scratch, impractical. To foster the development of more efficient NAS methods, surrogate [19, 31, 83, 88] and tabular [18, 39, 72, 84] NAS benchmarks have been proposed – particularly for convolutional networks and image classification tasks. These benchmarks have significantly aided the development of search algorithms to replace manual heuristics. Surrogate benchmarks such as Once-for-all [31] or HAT [78] follow the idea of two-stage weight-sharing based NAS [6], which trains a single *supernet* subsuming a large number of architectures into a single model, followed by a gradient-free search to select the Pareto optimal *sub-networks*. While some benchmarks focus on natural language understanding tasks, such as machine translation [78] and speech recognition [49], the efficacy of these techniques does not directly transfer to architectures for causal language modelling problems and across various hardware metrics (e.g., FLOPS, latency) and devices (e.g., CPUs, GPUs). Therefore, a hardware-aware benchmark for evaluating multi-objective NAS methods is crucial for advancing the design of inference-optimal LM architectures. In this paper, we introduce **HW-GPT-Bench** (see Figure 1 for an overview), a hardware-aware LM benchmark, based on the GPT-2 [58] architecture, for multi-objective NAS across 13 devices, 5 hardware metrics, and 4 model scales. Our contributions include:

- **Benchmark Creation** (Section 3): Establishing a benchmark across small, medium, and large model scales with surrogate supernets, performance predictors, and hardware metric predictors.

- **Faithful Latency and Energy Estimates** (Section 3): Contrary to previous in works in the NAS literature, we use surrogate predictors that provide calibrated predictions and faithfully model the uncertainties inherent in latency and energy profiling.

- **Metric Interaction Analysis and Algorithm Evaluation** (Sections 4 and 5): Studying interaction effects between different hardware and performance metrics, importance of architectural choices and evaluations of various multi-objective optimization algorithms, providing out-of-the-box baselines for future development.

We provide an open-source API [1], for latency and perplexity predictors, the supernetwork weights and different baselines studied, making integration of new methods into this benchmark straightforward. Through HW-GPT-Bench, we aim to accelerate research in hardware-aware NAS for language models, ultimately advancing the development of efficient and high-performing language model architectures.

## 2   Related work

**Structural Pruning, Computational Efficiency, and Neural Architecture Search.** Network pruning [42], a model compression technique, reduces model complexity with minimal performance loss. Empirical studies [35, 51, 60, 74] have examined the impact of pruning different layers and

---

[1]Code at: https://github.com/automl/HW-GPT-Bench/

modules in pretrained transformers on validation loss. Techniques such as KV-Caching [57] and Quantization [4, 32, 46, 70] improve inference time, memory footprint, and energy usage. These methods complement structural pruning and can be incorporated for further speedups. *Structured pruning* removes structured parameter blocks from pretrained models, while *unstructured pruning* introduces sparsity by zeroing out unimportant weights. Adaptive pruning methods [3, 23], which prune based on task difficulty, have also been proposed. Recently, Klein et al. [36] and Sarah et al. [64] used neural architecture search (NAS) [81] for automated structural pruning of pretrained LLMs. Similarly Muñoz et al. [53] and Munoz et al. [54] study, starting from pretrained models and parameter-efficient finetuning for NAS. Training multiple architectures from scratch is computationally expensive [59, 67, 89], so efficient NAS methods employ one-shot models (or supernetworks) [7, 31, 43] as performance proxies by inheriting weights and fine-tuning individual architectures. In our benchmark, we follow this procedure to create perplexity proxies from the largest GPT-2 [58] model. As a second stage, many multi-objective NAS methods [31, 78] run gradient-free search routines efficiently, such as evolutionary strategies, to optimize performance and hardware metrics. Recently, Sukthanker et al. [69] proposed a method that generates the entire Pareto set in a single stage based on preference vectors of objectives and hardware type.

**(Hardware-aware) NAS Benchmarks.** NAS benchmarks, both tabular and surrogate, emerged due to the high computational costs and reproducibility challenges in developing and evaluating NAS algorithms [18, 49, 84, 87, 88]. *Tabular* benchmarks [18, 20, 49, 84] evaluate all architectures in the search space upfront, but this becomes infeasible as search spaces and training times grow. To address this, *surrogate* benchmarks [83, 88] use model-based performance predictors, overcoming the limitations of tabular benchmarks and providing more realistic search space evaluations [88]. One-shot models [10, 29, 31, 78] can also act as surrogates by inheriting weights and evaluating performance on validation sets. Most NAS benchmarks focus on convolutional spaces and computer vision tasks [18, 20, 50, 84], with some targeting natural language processing (NLP) [37, 78] and speech recognition[49]. Given the rising computational costs of training, deploying, and searching models via multi-objective NAS, extended tabular NAS benchmarks now include hardware-specific metrics like FLOPS, on-device latency, and energy consumption [5, 19, 38, 39, 78]. Unlike these benchmarks, HW-GPT-Bench focuses on language modeling with decoder-only transformers [58, 73]. Additionally, our surrogates offer calibrated predictions by modeling the intrinsic heteroscedastic noise in latency and energy usage, rather than relying on single measurements [19, 38, 39].

## 3   HW-GPT-Bench Design Choices

In this section we provide details on design choices , such as the architecture search space, data collection procedure, performance and hardware metrics, as well as the surrogate model types.

### 3.1   Architecture Search Space

To construct our architecture search space, we pick the GPT-2 [58] language model, which is an autoregressive decoder-only transformer [73] composed by three primary components (see Figure 1, *left*): (i) **Embedding layers** that map input tokens to learnable representations and encode their position; (ii) **Transformer blocks** stacked multiple times; (iii) a **Prediction head** that predicts the next token for a given sequence. Moreover, each of the transformer blocks consist of: (a) a **Causal Self-Attention Block** that weights the significance of different input tokens (b) a **MLP block** containing two layers that project the input to a higher dimension and back to a lower one. We denote the ratio of the higher projection dimension to the transformer dimension as MLP ratio. In addition, we apply the following enhancements to the original architecture:

- **Rotary positional embeddings (RoPE)** [68]: A form of position embedding that captures absolute positional details using rotation matrices while seamlessly integrating explicit relative positional relationships into the self-attention mechanism. Importantly, RoPE offers several advantages, including adaptability to sequences of varying lengths, diminishing token interactions over greater relative distances, and the ability to enhance linear self-attention via relative positional encoding.

- **Parallel residual**: Following PaLM [13, 77], in contrast to the standard serialized formulation, we use a parallel formulation in each transformer block. Specifically, if $x$ is the input to the block, the

Table 1: HW-GPT-Bench search space. We pretrain 7 supernetworks with different sizes and search space strides: GPT-S, -M and -L, -S-wide, -M-wide, -L-wide, -XL-wide. On each of them we parameterize the dimensinality of the embedding layer, number of stacked layers (transformers blocks), number of self-attention heads and MLP ratio for every active layer, as well as if the bias is on or off.

| Supernet Type | Embedding Dim. | Layer No. | Head No. | MLP Ratio | Bias | No. of Archs | Supernet Size |
|---|---|---|---|---|---|---|---|
| GPT-S | [192, 384, 768] | [10, 11, 12] | [4, 8, 12] | [2, 3, 4] | [On, Off] | $\sim 10^{12}$ | 124M |
| GPT-M | [256, 512, 1024] | [22, 23, 24] | [8, 12, 16] | [2, 3, 4] | [On, Off] | $\sim 10^{24}$ | 350M |
| GPT-L | [320, 640, 1280] | [34, 35, 36] | [8, 16, 20] | [2, 3, 4] | [On, Off] | $\sim 10^{36}$ | 774M |
| GPT-S-wide | [192, 384, 768] | [3, 6, 12] | [3, 6, 12] | [1, 2, 4] | [On, Off] | $\sim 10^{12}$ | 124M |
| GPT-M-wide | [256, 512, 1024] | [6, 12, 24] | [4, 8, 16] | [1, 2, 4] | [On, Off] | $\sim 10^{24}$ | 350M |
| GPT-L-wide | [320, 640, 1280] | [9, 18, 36] | [5, 10, 20] | [1, 2, 4] | [On, Off] | $\sim 10^{36}$ | 774M |
| GPT-XL-wide | [400,800,1600] | [12,24,48] | [6, 12, 25] | [1, 2, 4] | [On, Off] | $\sim 10^{48}$ | 1.55B |

standard and parallel formulations can be written as:

$$y_{serialized} = x + \texttt{MLP}(\texttt{LayerNorm}(x + \texttt{Attention}(\texttt{LayerNorm}(x))))$$

$$y_{parallel} = x + \texttt{MLP}(\texttt{LayerNorm}(x)) + \texttt{Attention}(\texttt{LayerNorm}(x))$$

As reported in PaLM [13], the parallel formulation is faster at larger scales as the MLP and attention input matrix multiplications can be fused.

**Architectural choices.** Consider a search space $\mathcal{S} = \mathcal{D}_e \times \mathcal{D}_l \times \mathcal{D}_h \times \mathcal{D}_b \times \mathcal{D}_m$, obtained by parameterizing the building blocks of the transformer architecture, where $\mathcal{D}_e := \{e_1, e_2, e_3\}$, $\mathcal{D}_l := \{l_1, l_2, l_3\}$, $\mathcal{D}_h := \{h_1, h_2, h_3\}$, $\mathcal{D}_b := \{\texttt{On}, \texttt{Off}\}$ and $\mathcal{D}_m := \{m_1, m_2, m_3\}$ correspond to the set of embedding dimension choices, number of layers, number of heads, choice of setting the bias in linear layers and the MLP ratio choices, respectively. Furthermore, we choose the MLP ratio and the number of heads on a per-layer basis, amounting to a search space size of $\sim 10^{36}$ possible architectures. We represent architecture configurations as a list of integers $s = \{e, l, h^1, \cdots, h^l, m^1, \cdots, m^l, b\}$, where $e \in \mathcal{D}_e$, $l \in \mathcal{D}_l$, $h^l \in \mathcal{D}_h$, $m^l \in \mathcal{D}_m$ and $b \in \mathcal{D}_b$. $h^l$ and $m^l$ denote the number of heads and MLP ratio of layer $l$, respectively. Given a set of $m$ metrics (objectives) $\mathcal{Y} = \{y_m \in \mathcal{R}^m : y = f(s), s \in \mathcal{S}\}$ e.g.: latency, perplexity, energy, memory consumption etc., a NAS algorithm searches for the (Pareto) optimal architectures, evaluated using these metrics, from the space $\mathcal{S}$.

**Four Transformer scales.** Based on the values we assign to the choices in every architectural block, we can obtain arbitrary number of search spaces. In HW-GPT-Bench, we construct 7 such spaces, namely, GPT-S, GPT-M, GPT-L, GPT-S-wide, GPT-M-wide, GPT-L-wide and GPT-XL-wide as defined in Table 1, with the largest model containing 1.55B parameters. Note that in every search space, the *supernetwork* is the largest possible model, e.g. in GPT-S that would be $s = \{784, 12, 12, \cdots, 4, \cdots, \texttt{On}\}$.

### 3.2 Dataset Collection

Building a tabular benchmark for our search spaces with cardinality ranging from $\sim 10^{12}$ to $\sim 10^{48}$ is infeasible even for objectives such as latency or energy usage that are faster to measure than performance. Therefore, following Zela et al. [88], **we sample 10000 unique architectures uniformly at random** from each of the search spaces (GPT-S, -M, -L, -S-wide, -M-wide, -L-wide, -XL-wide), and use observations from these architectures to train our hardware and performance surrogates.

**Performance data.** We evaluate the *perplexity and accuracy* of the sampled architectures at every scale, by inheriting the weights corresponding to a particular architecture from the *supernetwork*, which subsumes all possible architectures in a single network (see Section 2).

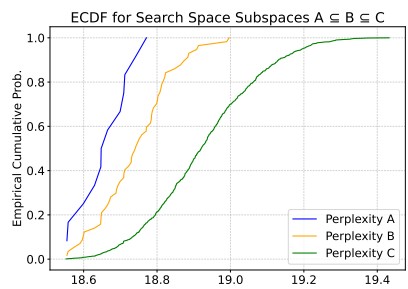

Figure 2: Empirical Cumulative distribution of different search space subspaces.

Since architectures index the same supernetwork to access their weights, all their individual weights are entangled [10, 31, 78]. Various strategies exist to pretrain the supernetwork, such as random

sampling, sandwich scheme, and pre-defined structured sampling [36, 85]. Following its effectiveness, as shown by [36], we employ the *sandwich scheme*, that at every mini-batch training iteration samples the largest, the smallest, and two random architectures from the search space. Similar to [36], the weights of different sub-networks are tightly coupled with each other and the memory footprint of the supernetwork is the same as the largest network in the space. This allows for extremely efficient training of about $10^{48}$ (for GPT-XL-wide) architectures by updating multiple architectures that share weights simultaneously. We train the supernetwork with the standard language modeling loss function, which is the average negative log likelihood of all tokens without label smoothing. We use the OpenWebText [2] dataset, split to train and test sets, for training the supernetwork and evaluating individual architectures' perplexity, respectively. We refer the reader to Appendix A for more details on the training pipeline and used hyperparameters. In Figure 2 we plot the *empirical cumulative distribution* of the 10k architectures evaluated using the pretrained supernetwork weights on the validation set. The green curve represents random sampling in a space of fixed embedding dimension of 1280 in the GPT-L space, the orange curve with the number of layers fixed to 36 as well, and the blue curve with the average number of heads and MLP ratio across layers greater than 16 and 3, respectively (with embedding dimension fixed to 1280 and number of layers to 36). From the cumulative distribution we can see that there is a lot to gain from searching for the right architectural choices instead of randomly sampling.

**Hardware metrics and devices.** In addition to perplexity, we also collect the following hardware-related metrics: (i) **number of parameters** (ii) **FLOPS** (Floating Point Operations) (iii) on-device **latency** (in ms) (iv) on-device **energy consumption** (in kWh) (v) **memory footprint** (in GB). We compute latencies and energies for all 10k sampled architectures on a variety of GPU and CPU types:

- **GPU devices:** NVidia RTX A6000, RTX 2080Ti, RTX 3080Ti, P100, A40, A100 and H100.
- **CPU devices:** Intel Xeon Silver, Xeon Gold, and AMD EPYC 7513, 7452, 7502.

For details on hardware specifications refer to Appendix B. To profile the energy usage, we use CodeCarbon[3] on CPU devices and Nvidia's visual profiling tool[4] on GPUs. For latency profiling we use the native PyTorch profiler[5] on both GPUs and CPUs. For FLOPs, we use the DeepSpeed library[6]. Furthermore, due to the high intrinsic measurement noise for energy and latencies, to get robust estimates, *we collect up to 10 observations per architecture for latency and up to 50 observations per architecture for energy* on GPUs. We then use these observations to incorporate the *aleatoric* uncertainty into the surrogates (see Section 3.3) and estimate the noisy latency and energy distributions more reliably. In all evaluations of every metric, we used a batch size of 8, 4, 1 and 1 for GPT-S, -M, -L and -XL scales respectively, and a sequence length of 1024.

In Figure 3 we show the computed ground-truth perplexity from the trained supernet, latency and energy usage values of all 10000 architectures. We can clearly see the *heteroscedastic noise in the latency and energy measurements*, with an increasing variance as the model perplexity improves. In the same figure, we show the Pareto fronts obtained by randomly sampling an observation (blue line) – 1 out of 10 for latency and 1 out of 50 for energy – while the best and worst possible Pareto fronts (red and black markers, respectively) are obtained by using the best and worst measured value, respectively. These results show that the high observation noise in the data can potentially affect the optimization trajectories of multi-objective algorithms, hence resulting in different Pareto fronts.

### 3.3 Surrogate Models

**Perplexity and Memory Usage Surrogates.** After pretraining the supernetwork, evaluating thousands of architectures on the test set can still be relatively expensive. To this end, similar to Han et al. [31], we train a MLP surrogate model on 80% of the collected datapoints, obtained by evaluating the supernetwork, to predict the perplexity given the architecture encoding as input. We also train a MLP surrogate to predict GPU memory usage. Evaluations on the unseen 2000 architectures, yielded a rank correlation of $> 0.90$ for every metric. Refer to Appendix C.1 for more details on the MLP architecture and training hyperparameters.

---

[2]https://skylion007.github.io/OpenWebTextCorpus/
[3]https://codecarbon.io/
[4]https://docs.nvidia.com/cuda/profiler-users-guide/index.html
[5]https://pytorch.org/tutorials/recipes/recipes/profiler_recipe.html
[6]https://github.com/microsoft/DeepSpeed

Trade-offs between perplexity, energies and latencies on RTX2080

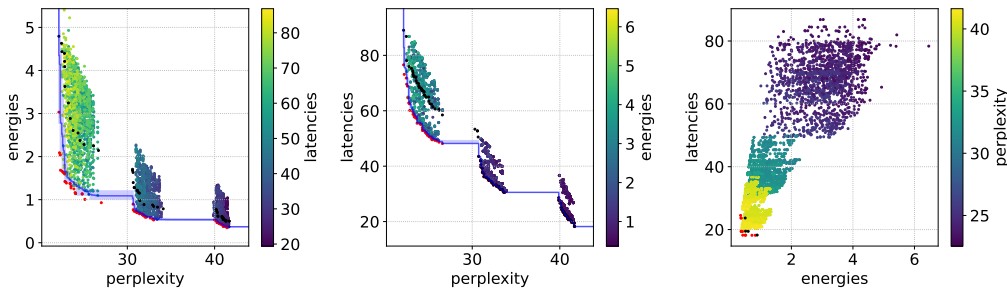

(a) Latency vs. Perplexity vs. Energy for GPT-S on RTX 2080Ti GPU

Trade-offs between perplexity, energy and latency on A100

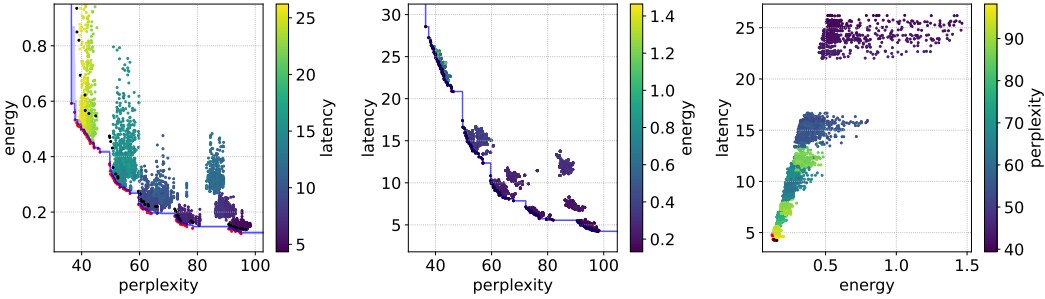

(b) Latency vs. Perplexity vs. Energy for GPT-S-wide on A100 GPU

Figure 3: Trade-offs between Energy, Latency, and Perplexity across architectures for different search spaces. The blue curve represents the Pareto front obtained by randomly sampling an observation, while the best and worst possible Pareto fronts (red and black markers, respectively) are obtained by using the best and worst measured value, respectively, for latencies and energies.

**Energy and Latency Surrogates.** From our initial collection of energy and latency observations for different architectures, we observe that on-device latencies and energies tend to be *very noisy*, and the median of observations is insufficient to capture this noisy distribution (see Figure 3). Moreover, we empirically observe that the distribution of energies and latencies is often normally distributed with a few outliers. A reliable surrogate model in such a case should not only be performant in terms of accuracy or ranking of the architectures, but on *uncertainty quantification and calibration* as well. As our surrogate model for predicting per-device latency and energy, we choose **AutoGluon** [22], the state-of-the-art automated machine learning system for tabular data [26] that has been shown to outperform (ensembles of) individual model types [61]. Specifically, we train two AutoGluon models on the 80% split of the sampled architectures to predict the first and second moments of the latency and energy distributions. AutoGluon builds stacking ensembles [82] to further enhance performance while using a portfolio [24, 61] of linear models, neural networks, and decision tree-based models to be robust to outliers and performant across diverse data distributions. To analyze different model choices, we compare AutoGluon to LightGBM [34] and XGBoost [11], state-of-the-art tabular regression models [28, 48, 61] as baselines. To enhance performance, for both baselines, we ensemble various configurations of LightGBM and XGBoost, and estimate the first and second moments using the individual baselearners' predictions. In addition, we also evaluate an ensemble mix of scitkit-learn's [56] Linear Regression, Ridge Regression, and Random Forest [8]. We refer the reader to Appendix C.2 for more details on the surrogate models.

After fitting AutoGluon and the baselines and computing evaluations on the testing data points, we compute various performance and calibration metrics from the *Uncertainty Toolbox*[7] [14, 71] to quantitatively compare AutoGluon to the other baselines. In Table 2, we report accuracy metrics, such as mean absolute error (MAE), Spearman rank correlation, etc., between predicted mean and true mean of observations (e.g. mean of the 50 energy observations per architecture), averaged

---

[7]https://github.com/uncertainty-toolbox/uncertainty-toolbox

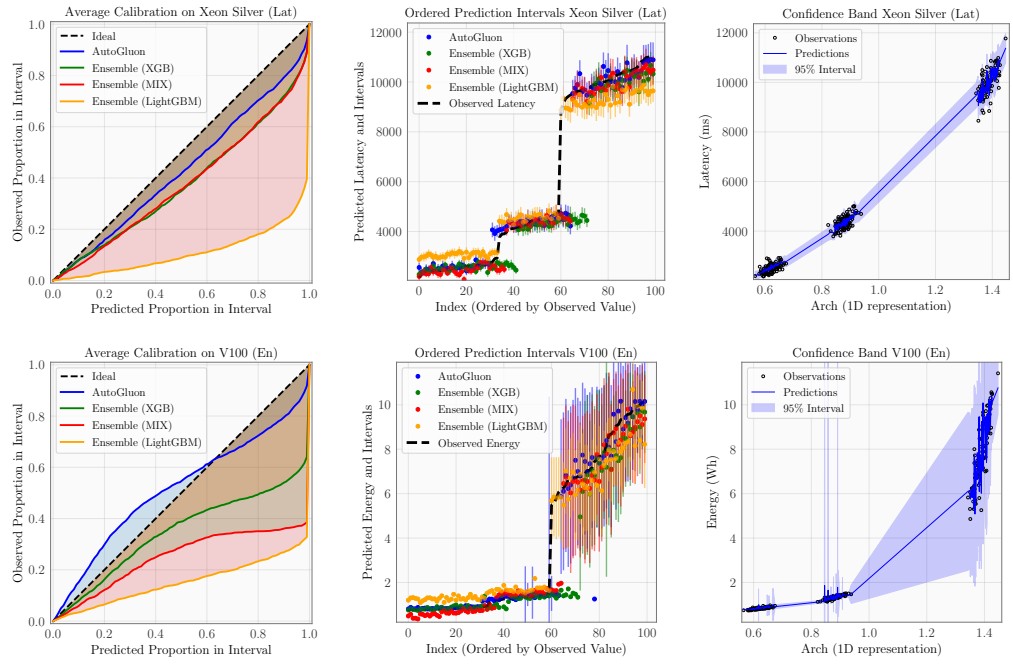

Figure 4: Calibration area, Prediction Intervals and Confidence Bounds for different surrogate types on Xeon Silver CPU (Latency) and V100 (Energy). The rightmost plots show only the predictions and confidence bands of AutoGluon.

Table 2: Various performance metrics of surrogates evaluated to predict H100 Latency and RTX2080 Energy. The arrow on the side of the metric name indicates if lower or higher is better.

| Surrogate | Accuracy | | | | | | | | | | | | Calibration | | | | | |
|---|---|---|---|---|---|---|---|---|---|---|---|---|---|---|---|---|---|---|
| | MAE ↓ | | RMSE ↓ | | MDAE ↓ | | MARPD ↓ | | R² ↑ | | Corr. ↑ | | RMS Cal. ↓ | | MA Cal. ↓ | | Miscal. Area ↓ | |
| | H100 | 2080 | H100 | 2080 | H100 | 2080 | H100 | 2080 | H100 | 2080 | H100 | 2080 | H100 | 2080 | H100 | 2080 | H100 | 2080 |
| **AutoGluon** | **0.153** | **1.80** | **0.211** | **2.576** | **0.111** | 1.121 | **0.153** | 10.677 | **0.999** | **0.904** | **0.999** | **0.950** | **0.223** | **0.244** | **0.198** | **0.217** | **0.199** | **0.220** |
| **Ensemble (Mix)** | 0.569 | 1.830 | 0.785 | 2.621 | 0.413 | **1.092** | 0.565 | 10.920 | 0.999 | 0.900 | 0.999 | 0.949 | 0.472 | 0.298 | 0.411 | 0.264 | 0.415 | 0.267 |
| **Ensemble (XGB)** | 0.620 | 1.832 | 0.827 | 2.628 | 0.475 | 1.154 | 0.629 | 10.919 | 0.990 | 0.899 | 0.990 | 0.948 | 0.481 | 0.286 | 0.417 | 0.251 | 0.421 | 0.254 |
| **Ensemble (LGB)** | 0.361 | 2.094 | 0.411 | 2.922 | 0.379 | 1.415 | 0.384 | 13.140 | 0.970 | 0.875 | 0.999 | 0.947 | 0.559 | 0.347 | 0.481 | 0.304 | 0.486 | 0.308 |

across architectures in the test set. Furthermore, we also compute various calibration metrics, such as average calibration error. In summary, from the results in Table 2 we can conclude that: **AutoGluon is the best model that provides both accurate and calibrated predictions, as well as reliable uncertainty estimates for energy usage and latency**. Refer to Appendix D for more details on these evaluation metrics.

Figure 4 shows the *average calibration plot* (left), *prediction interval plot* (middle) and *predicted mean and 95% confidence interval* (right) on two devices and two metrics. We can see from the rightmost plot that AutoGluon reliably predicts the mean of the observations and has calibrated uncertainty estimates (left plot: calibrated models have calibration curves that approach the ideal diagonal line). Following Tran et al. [71], we utilized the standard deviation of predictions from each surrogate model to generate Gaussian random variables for each test point. We then evaluated the residuals' adherence to their respective Gaussian random variables. Consequently, models considered "well-calibrated" had residuals that formed Gaussian distributions with standard deviations closely matching the model's predicted standard deviations. For more details on these plots see Appendix D.

## 4   Analysis and Interpretability on HW-GPT-Bench

**Correlations between hardware metrics.**   We now use our collected data to examine the relationships between performance and hardware metrics in order to gain insights on how these different metrics are correlated with each other and across devices. Interestingly, we observe that at smaller scales (Figure 62 in the Appendix), the energy and latency are highly correlated (assesed via the Kendall-$\tau$ correlation coefficient between the ground truth metrics across devices) with FLOPS, often used as a proxy for device-agnostic latency measurements. However, the correlation coefficient

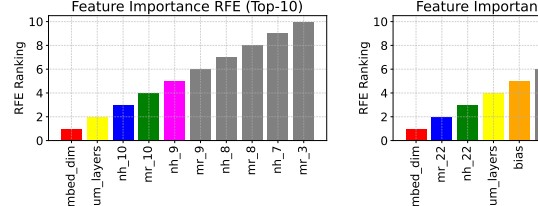 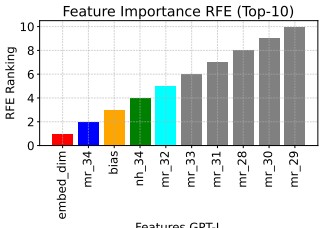

Figure 5: Feature ranking of architecture dimensions at different scales (lower rank is better). The embedding dimension (in red) is most important across scales and the number of layers (in yellow) is more important at smaller scales. MLP ratio (mr) and Number of heads (nh) at layer $N - 1$ is important across different scales (depicted in blue and green).

become progressively lower at larger model scales as shown in Figures 63 and 64 of the Appendix, mainly due to higher energy and latency measurement noise. As expected, per-device latency and energy are highly correlated, whilst latency/energy and perplexity are anti-correlated (see Figure 3).

**Importance of Architectural Choices.** For the collected perplexity data at each scale, to model the function of perplexity dependent on the architectural choices, we assume a power law of the form, for analysis purposes:

$$y = \mathcal{C} \times l^{\alpha} \times e^{\beta} \times \left( \frac{\sum_{i=1}^{l} h^i}{l} \right)^{\gamma} \times \left( \frac{\sum_{i=1}^{l} m^i}{l} \right)^{\delta} \times (b+1)^{\sigma},$$

where $\mathcal{C}, \alpha, \beta, \gamma, \delta, \sigma$ are data-dependent constants, $l$ is the number of layers, $e$ is the embedding dimension, $m^i$ and $h^i$ are the MLP ratio and number of heads on layer $i$, respectively, and b is the bias. After fitting the power law on the collected 10000 pairs (architecture, perplexity), we obtain the following estimated coefficients:

$$\text{GPT-S:} \quad y = 646.234 \cdot l^{-0.226} \cdot e^{-0.371} \cdot m^{-0.100} \cdot h^{-0.076} \cdot b^{-0.001},$$
$$\text{GPT-M:} \quad y = 404.456 \cdot l^{-0.104} \cdot e^{-0.343} \cdot m^{-0.091} \cdot h^{-0.049} \cdot b^{-0.005},$$
$$\text{GPT-L:} \quad y = 280.757 \cdot l^{-0.073} \cdot e^{-0.309} \cdot m^{-0.088} \cdot h^{-0.051} \cdot b^{-0.005}.$$

An ordinary-least-squares (OLS) fit on the log-transformed data indicates that all search dimensions are statistically significant with p-value $< 0.001$, with embedding dimension being the most important architecture parameter in the search space. The number of layers $l$ and MLP ratio become increasingly less important at larger scales and could potentially be pruned without significantly impacting perplexity. The importance of bias stays more or less constant across scales, while the MLP ratio is more important than the number of heads, indicating that a significant number of heads are possibly redundant and are amenable to pruning. In Figure 5, we study the ranking upon applying Recursive Feature Elimination (RFE) [12] to the architecture features and present the 10 top-ranked features. We observe that embedding dimension and number of layers are important across different model scales. Furthermore, the MLP ratio and the number of heads chosen in the transformer's later layers

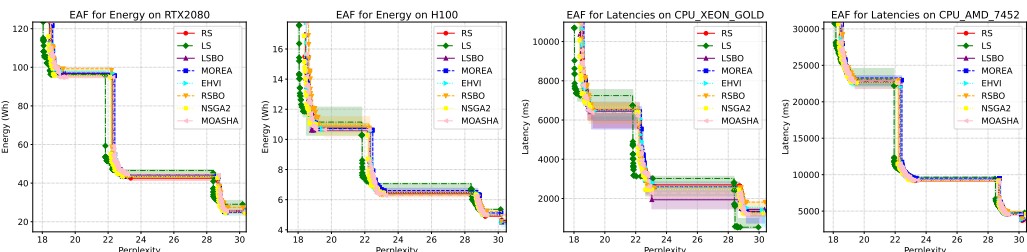

Figure 6: Different multi-objective NAS baselines on RTX2080Ti and H100 energy, Xeon Gold CPU Latency and CPU AMD 7452 Latency for GPT-L

(e.g., layer $N-1$ and layer $N-2$) are more important than their choices in earlier layers. We present additional results in Appendix F.

## 5 HW-GPT as a Benchmark for Multi-objective Optimization

In this section, we showcase how HW-GPT-Bench can be used as a benchmark for evaluating multi-objective NAS algorithms.

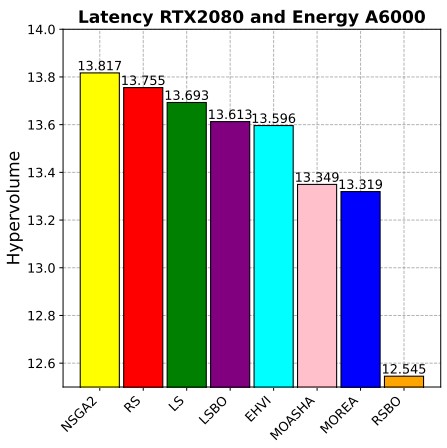

**Multi-objective NAS algorithms.** State-of-the-art multi-objective NAS (MO-NAS) methods aim to identify Pareto-optimal configurations that balance performance and efficiency. HW-GPT-Bench, enables simulations of their optimization trajectories *in just a few minutes* using the predictions from our surrogate models. We simulate multiple runs of the following MO-NAS methods implemented in Syne Tune [62]: (i) Random Search (RS) (ii) Multi-objective Regularized Evolution (MOREA) [59] (iii) Non-dominated Sorting Genetic Algorithm II (NSGA-II) [16] (iv) Local Search (LS) (v) Multi-objective Asynchronous Successive Halving (MOASHA) [65] (vi) Bayesian Optimization with Random Scalarizations (RSBO) and (vii) Linear Scalarizations (LSBO) [55] (viii) Expected Hypervolume Improvement (EHVI) [15]. Refer to Appendix K for a more detailed description of them.

Figure 7: HV of MO-NAS methods on 3 objectives (on GPT-S), namely, perplexity, RTX 2080Ti latency and A6000 energy usage.

### 5.1 Experiments with 2 objectives

We run all MO-NAS methods for a fixed (simulated) time budget of 16 CPU hours. We repeat each run 4 times to account for the noise in the latency and energy predictions. In Figure 8 shows the hypervolume indicator over number of surrogate evaluations. Notably, EHVI and NSGA-II achieve a higher hypervolume under smaller budgets compared to Local Search (LS) and Random Search (RS), underscoring the efficiency of model-based optimization algorithms in navigating the search space and identify Pareto-optimal architectures. To aggregate the resulting Pareto fronts from our multiple runs, we use the Empirical Attainment Function (EAF) [47], which represents a coherent way to capture the uncertainty in the multi-objective metric space (see Appendix G for details). We show the results of all methods in Figure 6. We can see that LS and NSGA-II generally yield more favorable trade-offs between the objectives, achieving lower perplexity for a given energy, latency or memory consumption. In contrast, Random Search (RS) shows wider variability, with worst case scenarios containing solutions in the Pareto set with relatively high energy usage and low perplexity. Appendix L.1 contains the rest of the experimental results across devices and metrics.

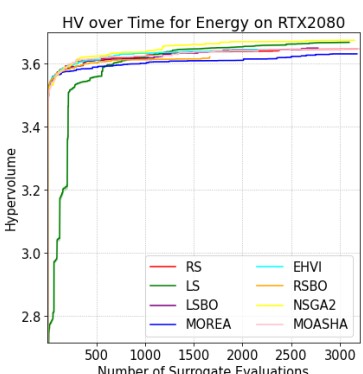

Figure 8: HV of blackbox optimizers over 200 surrogate evaluations on HW-GPT-Bench for RTX2080 Energy.

### 5.2 Experiments with more than 2 objectives

All the methods we run on HW-GPT-Bench on 2 objectives in the section above, are extendable to more than 2 objectives. HW-GPT-Bench enables optimizing for different hardware metrics, not necessarily measured on the same hardware device. Here, we showcase this by optimizing simultaneously for perplexity, latency on RTX 2080Ti and energy usage on the A6000 GPU. We picked these objectives due to their relatively low correlation, as we can see in Figure 62. We run the

```
from hwgpt.api import HWGPT
api = HWGPT(search_space="s") # initialize API
random_arch = api.sample_arch() # sample random arch
api.set_arch(random_arch) # set  arch
results = api.query() # query all
energy = api.query(metric="energy") # query energy
rtx2080 = api.query(device="rtx2080") # query device
# query perplexity based on mlp predictor
perplexity_mlp = api.query(metric="perplexity",predictor="mlp")
# query perplexity based on supernet
perplexity_supernet = api.query(metric="perplexity",predictor="supernet")
# run baseline and plot EAF
nsga2_results = api.run_baseline(method="nsga2", device="rtx2080", metrics=["energy","perplexity"],
    ppl_predictor="mlp")
# plot Pareto-front
api.plot_eaf(nsga2_results)
```

Snippet 1: A minimal example of the HW-GPT-Bench API.

baselines for 16 hrs on a single CPU and compute the hypervolume [8], which we show in Figure 7 (see Appendix L.2 for the other results.). Notably, NSGA-II achieves good trade-offs also in the 3-objective case, however, the Bayesian optimization methods perform worse than RS and LS here.

With this set of baselines and evaluations, we hope that HW-GPT-Bench will serve as a de-facto testbed for prototyping and benchmarking future hardware-aware optimization methods.

### 5.3  HW-GPT-Bench API

We provide a minimal API for users, that enables loading the benchmark and querying all or specific hardware and performance metrics across devices, and different search spaces with just a few lines of code.

Users can also select which perplexity surrogate to use – either the supernetwork itself or the MLP performance predictor – ensuring flexibility in performance assessment. Furthermore, the API supports the execution of multi-objective baselines, enabling rigorous benchmarking and comparative analysis. See code snippet 1 for a simple example. Additional examples can be found in Appendix O.

## 6  Conclusions, Broader Impact and Implications

We introduce HW-GPT-Bench, a hardware-aware surrogate benchmark for evaluating language models across various **hardware devices**, **metrics**, and **scales** on a single CPU *in just a few seconds*. By enabling efficient exploration of multi-objective NAS algorithms to achieve Pareto-optimal configurations across multiple hardware metrics and devices, our work has several broader societal implications: (i) **Energy Efficiency and Environmental Impact** - It promotes the development of energy-efficient language models, mitigating the environmental cost of large-scale AI systems and enhancing sustainability; (ii) **Enhanced Research and Development** - It accelerates research in NAS and structural pruning, leading to more energy-efficient architectures; (iii) **Accessibility and Democratization of AI** - Resource-efficient language models enable innovation for users and organizations with limited resources; (iv) **Economic Benefits** - Optimizing for hardware efficiency reduces training and deployment costs, benefiting industries reliant on extensive language model querying and improving user experience.

Overall, HW-GPT-Bench addresses critical challenges in developing and deploying algorithms that enhance the resource efficiency of language models, providing a more sustainable, accessible, and reliable benchmarking framework. It underscores the importance of considering hardware efficiency constraints alongside performance in advancing language models.

---

[8]computed with respect to the empirical nadir point for the 3 objectives

## Acknowledgments

This research was partially supported by the following sources: TAILOR, a project funded by EU Horizon 2020 research and innovation programme under GA No 952215; the Deutsche Forschungs-gemeinschaft (DFG, German Research Foundation) under grant number 417962828 and 499552394, SFB 1597 (SmallData); the European Research Council (ERC) Consolidator Grant "Deep Learning 2.0" (grant no. 101045765). Robert Bosch GmbH is acknowledged for financial support. The authors acknowledge support from ELLIS and ELIZA. Funded by the European Union. Views and opinions expressed are however those of the author(s) only and do not necessarily reflect those of the European Union or the ERC. Neither the European Union nor the ERC can be held responsible for them. Aaron Klein acknowledges the financial support by the Federal Ministry of Education and Research of Germany and by Sächsische Staatsministerium für Wissenschaft, Kultur und Tourismus in the programme Center of Excellence for AI-research „Center for Scalable Data Analytics and Artificial Intelligence Dresden/Leipzig", project identification number: ScaDS.AI. Frank Hutter acknowledges the financial support of the Hector Foundation.

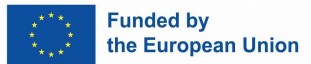

.

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

# A Pretraining details

We present all the training hyperparameters for the supernetwork (see Section 3.2 too) at different scales in Table 3. We train GPT-S, -M and -L scales for 4, 6 and 8 days, respectively, on 4 Nvidia A100 GPUs. GPT-XL is trained for 6 days across 40 A100 GPUs. As described in Section 3.2, we use the sandwich rule for the supernetwork training, which accumulates gradients from the smallest, largest and 2 random subnetworks of the supernetwork in each gradient update step.

| Hyperparameter | Value |
|---|---|
| **Trainer** | |
| num_gpus | 4 |
| gradient_clip_val | 1.0 |
| max_steps | 800000 |
| precision | BFloat16 Mixed precision |
| seed | 1234 |
| **Optimizer** | |
| optimizer | AdamW |
| lr | 0.000316 |
| weight_decay | 0.1 |
| betas | [0.9, 0.95] |
| eps | 1.0e-09 |
| seed | 1234 |
| **Optimizer Parameter Grouping** | |
| bias_weight_decay | False |
| normalization_weight_decay | False |
| **Scheduler** | |
| num_warmup_steps | 4000 |
| num_training_steps | 800000 |
| decay_factor | 0.1 |
| schedule | cosine |
| **Model** | |
| block_size | 1024 |
| vocab_size | 50254 |
| padding_multiple | 512 |
| scale_embeddings | False |
| padded_vocab_size | 50254 |
| head_size | 64 |
| lm_head_bias | False |
| attn_dropout | 0.1 |
| resi_dropout | 0.1 |
| embed_dropout | 0.1 |
| shared_attention_norm | False |
| norm | LayerNorm |
| rope_condense_ratio | 1 |
| scale_attn_weights | True |
| scale_attn_by_inverse_layer_idx | True |
| shared_embedding | True |
| pos_embedding | False |
| rel_pos_enc | True |
| rotary_percentage | 0.5 |
| norm_eps | 1e-5 |
| rope_base | 10000 |
| parallel_residual | True |
| initializer_range | 0.02 |
| **Data** | |
| dataset | "openwebtext" |
| num_cpu_worker | 32 |
| batch_size | 32(GPT-S), 8(GPT-M), 8(GPT-L), 1(GPT-XL-wide) |
| max_sample_len | 1024 |
| val_ratio | 0.0005 |
| val_split_seed | 2357 |

Table 3: Supernet training hyperparameters.

# B    Hardware specifications and Properties

In Table 4, we present details of the 8 GPU devices and 5 CPU devices we used for profiling our 3 search spaces GPT-S, GPT-M and GPT-L. The devices we study capture a wide range of micro-architectures, number of cores and GPU/CPU memory. The considered GPUs capture a wide range of inference latencies and throughput [9].

Table 4: Hardware specifications.

| Platform | Device Name | Micro Architecture | Number of Cores | Memory |
|---|---|---|---|---|
| GPU | NVIDIA RTX 2080 | Turing | CUDA 2944 | 8GB GDDR6 |
| | NVIDIA RTX 3080 | Ampere | CUDA 8704 | 12GB GDDR6X |
| | NVIDIA RTX A6000 | Ampere | CUDA 10752 | 48 GB GDDR6 |
| | NVIDIA A100 | Ampere | CUDA 6912 | 40GB HBM2e |
| | NVIDIA A40 | Ampere | CUDA 10752 | 48GB GDDR6 |
| | NVIDIA Tesla P100 | Pascal | CUDA 3584 | 12GB HBM2 |
| | NVIDIA Tesla V100 | Volta | CUDA 5120 | 24GB GDDR6 |
| | NVIDIA H100 | Hopper | CUDA 16896 | 80GB HBM3 |
| CPU | AMD 7452 | Zen2 | CPU Core 32 | Cache 128 MB |
| | AMD 7513 | Zen3 | CPU Core 32 | Cache 128MB |
| | AMD 7502 | Zen2 | CPU Core 32 | Cache 128MB |
| | Intel Xeon Silver 4114 | Intel P6 | CPU Core 10 | Cache 13.75MB |
| | Intel Xeon Gold 6242 | Intel P6 | CPU Core 16 | Cache 22MB |

# C    Surrogate models

In this section, we provide details of the latency and energy surrogate predictors used in HW-GPT-Bench. The surrogates are trained and evaluated on 8000 and 2000 fixed architectures, respectively.

## C.1    PPL Surrogates

**Data collection.**    We subsample 10% from the OpenWebText training set and use that as a validation set to compute the perplexity and accuracy metrics of architectures in the supernetwork, that inherit the trained supernetwork weights. The time to collect the perplexity pairs for 10k architectures was 8, 16, 24 and 32 days on 4 A100 GPUs for GPT-S, -M and -L and -XL-wide, respectively. Note that this computation is trivial to parallelize, as all architectures can be evaluated independently of each other, given the pretrained supernetwork weights. Afterwards, the $(x, y)$ pairs, where $x$ is the one-hot encoded architecture and $y$ the corresponding perplexity value, are used to train an MLP (Multi Layer Perceptron) using the MSE loss to predict the perplexity on unseen (test) architectures.

**Surrogate Architecture.**    The MLP contains 4 linear layers with hidden dimension of 128 and output dimension of 1 (the perplexity prediction). We use ReLU activations at every layer. The dimension of the one-hot encoding is 3 (3 choices for layers) + 3 (3 choices for embedding dimension) + max_layers×3 (3 choices for number of heads for every layer) + max_layers×3 (3 choices for MLP ratio for every layer). We train the MLP for 4000 epochs using the Adam optimizer with a learning rate of $10^{-3}$ and a batch size of 1024.

## C.2    Hardware Surrogates

We now describe the different surrogates we study to model uncertainties in energy and latency prediction. Each of the surrogates is trained to predict mean and standard deviation of energy or latency predictions of a given architecture. For FLOPs, memory and number of parameters, we compute and predict a single observation per architecture. We use 8000 architectures for training our surrogates and 2000 to test the performance and calibration properties of our surrogates.

---

[9]GPU benchmark

**Data collection.** We measure energy and latency pairs across different devices multiple times to capture the noise in the energy and latency profiling. Additionally, we also profile hardware metrics like Floating Point Operations (FLOPs), Memory consumption (Float16 and BFloat16) and number of parameters for an architecture. Specifically, for every architecture, we compute 50 observations per architecture for energies on GPU and 10 observations for energies on CPU. Similarly, we compute 10 observations per architecture for latencies on both CPUs and GPUs. We use 8 CPU cores (per GPU) for all the latency and energy evaluations.

**AutoGluon.** AutoGluon simplifies the process of developing, training, and deploying machine learning models by automating key tasks such as feature engineering, model selection, and hyperparameter tuning. AutoGluon has two major components. First one is Bagging (bootstrap aggregation) on wide varieties of models like decision trees, random forests, nearest neighbors, linear models and neural networks. Second one is stacking, where predictions of models on the first level are fed as input to models at the next level. We use an AutoGluon model per metric (latency or energy) with `dynamic_stacking = False`, `num_stack_levels = 1`, `number_of_bag_folds = 8` and `number_of_bag_sets = 4`. The input feature map to the model is simply a concatenation of chosen hyperparameters with choices for MLP ratio and number of heads set to -1 when a particular layer is not selected. We train two AutoGluon predictors, one to predict mean, and one to predict the standard deviation of the ground truth latency or energy measurements.

**Ensemble (XGB).** We use an ensemble of 27 Extreme Gradient Boosting (XGB) models with different hyperparameter combinations. Specifically, the models in the ensemble contain a cross product of `n_estimator` choices from [200, 500, 800], `depth` choices from [5,9,3] and `learning_rate` choices from [0.01,0.1,1.0]. We then fit this ensemble, and predict the mean and standard deviation values for the metric at hand by aggregating the predictions of different models using bagging.

**Ensemble (LightGBM).** We use an ensemble of 36 Light Gradient Boosting Machine (LightGBM) models with different hyperparameter combinations. Specifically, the models in the ensemble contain a cross product of `n_estimator` choices from [200, 500, 800], `depth` choices from [5,9,3] and `learning_rate` choices from [0.01,0.1,0.001,1.0]. We then fit this ensemble, and predict the mean and standard deviation values for the metric at hand by aggregating the predictions of different models using bagging.

**Ensemble (MIX).** In addition to ensembles based on XGB and LightGBM, we also fit an ensemble containing a mixture of different machine learning models. Specifically, this ensemble contains 4 XGB and LightGBM regressors (default: [`n_estimators = 500`,`max_depth = 5` ,`learning_rate = 0.01`],[`n_estimators = 800`,`max_depth = 3`,`learning_rate = 0.1`], [`n_estimators = 200`,`max_depth = 9`,`learning_rate = 1`]), as well as LinearRegression, Ridge and RandomForestRegressor from `sklearn` with their default hyperparameters.

### C.3 Memory Surrogate.

We model the memory consumption using a simple MLP (similar to perplexity). The MLP has 4 layers with a hidden dimension of 128 and uses ReLU activations. The input to the MLP is the normalized feature map corresponding to the concatenation of the architecture parameters. The MLP is trained using Adam, with learning rate of $10^{-4}$ and a batch size of 1024, for 4000 epochs.

## D  Accuracy and Calibration of Surrogate models

In this section we provide a more detailed description on the metrics we used to evaluate our surrogate models in Section 3.3, as well as additional results that complement results shown in Table 2 and Figure 4.

### D.1  Accuracy and Calibration Metrics

Given a finite dataset $D = \{(x_i, y_i)_{i=1}^N\}$ and a regression model that predicts the mean value $\hat{y}$, we use standard accuracy metrics:

**Root Mean Squared Error (RMSE):** $RMSE = \sqrt{\frac{1}{N}\sum_{i=1}^{N}(\hat{y_i} - y_i)^2}$

**Mean Absolute Error (MAE):** $MAE = \frac{1}{N}\sum_{i=1}^{N}|\hat{y_i} - y_i|$

**Median Absolute Error (MDAE):** Median of the absolute differences between predicted and actual values, in ascending order.

$$MDAE = med(\{|\hat{y_i} - y_i|\}_{i=1}^{N})$$

**Mean Absolute Percentage Relative Deviation (MARPD):** A similar metric to MAE but scaled by the sum of the absolute values of predicted and actual values. This scaling can be beneficial when dealing with a wider range of values, especially when the predicted or actual values are very close to zero.

$$MARPD = \frac{1}{N}\sum_{i=1}^{N}\left|\frac{2\cdot(\hat{y_i} - y_i)}{|\hat{y_i}| + |y_i|}\right|\cdot 100$$

**Coefficient of Determination ($R^2$):** Tells us about the proportion of variance in the dependent variable (y) explained by the independent variable (x) through the model. Denoting by $\bar{y} = \sum_{i=1}^{N} y_i$ the mean of the ground truth data, $R^2$ is defined as:

$$R^2 = 1 - \frac{\sum_{i=1}^{N}(\hat{y_i} - y_i)^2}{\sum_{i=1}^{N}(\bar{y} - y_i)^2}$$

**Correlation Coefficient:** The Pearson correlation coefficient ($r$) measures the strength and direction of the linear relationship between predicted values ($\hat{y_i}$) and actual values ($y_i$) of the data. It ranges from -1 to 1.

$$r = \frac{\sum_{i=1}^{N}(\hat{y_i} - \bar{y})(y_i - \bar{y})}{\sqrt{\sum_{i=1}^{N}(\hat{y_i} - \bar{y})^2 \sum_{i=1}^{N}(y_i - \bar{y})^2}}$$

**Root Mean Squared Calibration Error (RMS Cal.):** A metric used to evaluate the calibration of probabilistic predictions in regression models, particularly in the context of uncertainty quantification. It measures the discrepancy between predicted and observed quantiles. For a model to be well-calibrated, the proportion of observations below a given predicted quantile should match the quantile value. For example, the 90th percentile prediction should contain the true outcome 90% of the time. RMS Cal. is computed as the root mean square of the difference between the predicted quantile levels and the observed frequencies over all quantiles:

$$\text{RMS Cal.} = \sqrt{\frac{1}{K}\sum_{k=1}^{K}\left(P(Y \leq Q_k) - q_k\right)^2},$$

where $Q_k$ is the k-th predicted quantile, $q_k$ is the corresponding quantile level, and $P(Y \leq Q_k) = \frac{1}{N}\sum_{i=1}^{N}\mathbb{I}[y_i \leq Q_k(x_i)]$ is the empirical frequency of the target variables $Y = \{y_i\}_{i=1}^{N}$ being less than or equal to $Q_k$. $\mathbb{I}$ stands for the indicator function. A lower RMS Cal. indicates better calibration, reflecting that the predicted uncertainties are accurate and reliable.

**Mean Absolute Calibration Error (MA Cal.):** MA Cal. is computed as the average absolute difference between the predicted quantile levels and the observed frequencies over all quantiles:

$$\text{MA Cal.} = \frac{1}{K}\sum_{k=1}^{K}|P(Y \leq Q_k) - q_k|.$$

### D.2 Additional Results on the Accuracy and Calibration of Surrogates

In Tables 5-10 and Figures 9-34 we show additional results that complement results shown in Table 2 and Figure 4.

Table 5: Various performance metrics of surrogates evaluated to predict latencies and energies on GPT-S for different GPUs. The arrow on the side of the metric name indicates if lower or higher is better.

| GPU | Surrogate | Accuracy | | | | | | | | | | | | Calibration | | | | | |
| | | MAE ↓ | | RMSE ↓ | | MDAE ↓ | | MARPD ↓ | | $R^2$ ↑ | | Corr. ↑ | | RMS Cal. ↓ | | MA Cal. ↓ | | Miscal. Area ↓ | |
| | | latencies | energies | latencies | energies | latencies | energies | latencies | energies | latencies | energies | latencies | energies | latencies | energies | latencies | energies | latencies | energies |
|---|---|---|---|---|---|---|---|---|---|---|---|---|---|---|---|---|---|---|---|
| A100 | AutoGluon | **0.39** | **0.03** | **0.50** | **0.28** | **0.33** | **0.02** | **0.23** | **5.68** | **1.00** | **0.17** | **1.00** | **0.42** | **0.46** | 0.09 | **0.40** | 0.08 | **0.41** | 0.08 |
| | Ensemble (XGB) | 1.81 | 0.05 | 2.37 | 0.29 | 1.48 | 0.02 | 1.03 | 8.32 | 1.00 | 0.10 | 1.00 | 0.34 | 0.54 | **0.08** | 0.47 | 0.07 | 0.47 | 0.07 |
| | Ensemble (LGB) | 8.74 | 0.05 | 10.14 | 0.29 | 8.37 | 0.03 | 5.06 | 9.70 | 0.97 | 0.14 | 1.00 | 0.37 | 0.57 | 0.08 | 0.49 | **0.06** | 0.49 | **0.06** |
| | Ensemble (Mix) | 1.93 | 0.04 | 2.51 | 0.29 | 1.57 | 0.02 | 1.09 | 7.42 | 1.00 | 0.14 | 1.00 | 0.38 | 0.55 | 0.08 | 0.47 | 0.07 | 0.47 | 0.07 |
| A40 | AutoGluon | **0.52** | **0.09** | **0.76** | **0.15** | **0.32** | **0.03** | **0.18** | **6.16** | **1.00** | **0.93** | **1.00** | **0.96** | **0.35** | 0.21 | **0.31** | 0.18 | **0.31** | 0.18 |
| | Ensemble (XGB) | 3.16 | 0.09 | 4.19 | 0.16 | 2.43 | 0.03 | 1.26 | 6.56 | 1.00 | 0.92 | 1.00 | 0.96 | 0.54 | 0.16 | 0.46 | 0.14 | 0.47 | 0.15 |
| | Ensemble (LGB) | 14.90 | 0.12 | 17.40 | 0.18 | 14.80 | 0.08 | 6.21 | 11.26 | 0.97 | 0.90 | 1.00 | 0.96 | 0.57 | **0.09** | 0.49 | **0.07** | 0.49 | **0.08** |
| | Ensemble (Mix) | 3.16 | 0.09 | 4.32 | 0.15 | 2.30 | 0.03 | 1.20 | 6.49 | 1.00 | 0.93 | 1.00 | 0.96 | 0.54 | 0.17 | 0.46 | 0.15 | 0.47 | 0.15 |
| H100 | AutoGluon | **0.18** | **0.02** | **0.25** | **0.03** | **0.14** | **0.01** | **0.23** | **3.34** | **1.00** | **0.90** | **1.00** | **0.95** | **0.19** | **0.05** | **0.17** | **0.04** | **0.17** | **0.04** |
| | Ensemble (XGB) | 0.76 | 0.02 | 0.99 | 0.03 | 0.61 | 0.01 | 0.98 | 3.57 | 1.00 | 0.90 | 1.00 | 0.95 | 0.48 | 0.05 | 0.41 | 0.05 | 0.42 | 0.05 |
| | Ensemble (LGB) | 3.45 | 0.02 | 4.02 | 0.04 | 3.34 | 0.02 | 4.44 | 4.64 | 0.97 | 0.87 | 1.00 | 0.95 | 0.55 | 0.18 | 0.48 | 0.16 | 0.48 | 0.16 |
| | Ensemble (Mix) | 0.77 | 0.02 | 1.00 | 0.03 | 0.64 | 0.01 | 0.99 | 3.50 | 1.00 | 0.90 | 1.00 | 0.95 | 0.47 | 0.05 | 0.41 | 0.05 | 0.42 | 0.05 |
| RTX2080 | AutoGluon | **20.01** | **0.09** | **28.40** | **0.14** | **11.91** | **0.07** | **4.54** | **6.45** | 0.98 | **0.99** | 0.99 | **0.99** | **0.35** | 0.26 | **0.29** | 0.23 | **0.29** | 0.23 |
| | Ensemble (XGB) | 20.59 | 0.10 | 29.58 | 0.15 | 12.17 | 0.07 | 4.60 | 6.95 | 0.97 | 0.98 | 0.99 | 0.99 | 0.42 | 0.24 | 0.35 | 0.21 | 0.36 | 0.21 |
| | Ensemble (LGB) | 32.25 | 0.20 | 42.97 | 0.25 | 22.52 | 0.18 | 7.14 | 14.77 | 0.94 | 0.96 | 0.99 | 0.99 | 0.53 | **0.05** | 0.46 | **0.04** | 0.46 | **0.04** |
| | Ensemble (Mix) | 20.79 | 0.10 | 29.25 | 0.15 | 13.27 | 0.08 | 4.71 | 7.52 | 0.97 | 0.98 | 0.99 | 0.99 | 0.42 | 0.21 | 0.36 | 0.19 | 0.37 | 0.20 |
| RTX3080 | AutoGluon | **6.21** | **0.06** | **8.83** | **0.12** | **3.69** | **0.03** | **1.97** | **4.10** | **1.00** | **0.96** | **1.00** | **0.98** | **0.43** | 0.03 | **0.36** | 0.03 | **0.37** | 0.03 |
| | Ensemble (XGB) | 7.18 | 0.06 | 10.25 | 0.12 | 4.26 | 0.04 | 2.21 | 4.67 | 1.00 | 0.95 | 1.00 | 0.98 | 0.47 | **0.02** | 0.41 | **0.02** | 0.41 | **0.02** |
| | Ensemble (LGB) | 22.76 | 0.10 | 27.36 | 0.16 | 21.79 | 0.08 | 7.45 | 8.40 | 0.97 | 0.92 | 1.00 | 0.98 | 0.56 | 0.25 | 0.48 | 0.22 | 0.49 | 0.23 |
| | Ensemble (Mix) | 7.49 | 0.07 | 10.43 | 0.13 | 5.05 | 0.04 | 2.37 | 4.74 | 1.00 | 0.95 | 1.00 | 0.98 | 0.48 | 0.02 | 0.42 | 0.02 | 0.42 | 0.02 |
| A6000 | AutoGluon | **1.74** | **0.05** | **2.68** | **0.06** | **1.03** | **0.04** | **0.85** | **5.95** | **1.00** | **0.93** | **1.00** | **0.96** | **0.47** | 0.04 | **0.40** | 0.03 | **0.41** | 0.03 |
| | Ensemble (XGB) | 3.34 | 0.05 | 4.45 | 0.06 | 2.42 | 0.04 | 1.54 | 6.17 | 1.00 | 0.92 | 1.00 | 0.96 | 0.53 | **0.02** | 0.45 | **0.02** | 0.46 | **0.02** |
| | Ensemble (LGB) | 13.46 | 0.05 | 15.83 | 0.07 | 13.16 | 0.04 | 6.48 | 6.85 | 0.97 | 0.90 | 1.00 | 0.96 | 0.57 | 0.04 | 0.49 | 0.04 | 0.49 | 0.04 |
| | Ensemble (Mix) | 3.39 | 0.05 | 4.54 | 0.06 | 2.50 | 0.04 | 1.54 | 6.22 | 1.00 | 0.92 | 1.00 | 0.96 | 0.54 | 0.03 | 0.46 | 0.03 | 0.47 | 0.03 |
| V100 | AutoGluon | **7.49** | **0.02** | **10.51** | **0.14** | **3.94** | **0.01** | **2.21** | **2.64** | 0.99 | **0.70** | **1.00** | **0.84** | **0.44** | 0.36 | **0.38** | 0.32 | **0.39** | 0.33 |
| | Ensemble (XGB) | 8.03 | 0.02 | 11.42 | 0.15 | 4.60 | 0.01 | 2.41 | 3.68 | 0.99 | 0.64 | 1.00 | 0.80 | 0.47 | 0.37 | 0.41 | 0.33 | 0.41 | 0.33 |
| | Ensemble (LGB) | 18.87 | 0.04 | 23.54 | 0.15 | 16.73 | 0.03 | 6.32 | 7.22 | 0.96 | 0.66 | 1.00 | 0.83 | 0.55 | 0.52 | 0.47 | 0.45 | 0.48 | 0.45 |
| | Ensemble (Mix) | 8.10 | 0.02 | 11.40 | 0.14 | 4.82 | 0.01 | 2.43 | 3.40 | 0.99 | 0.69 | 1.00 | 0.83 | 0.48 | 0.37 | 0.42 | 0.32 | 0.42 | 0.33 |
| P100 | AutoGluon | **3.62** | **0.30** | **5.31** | **0.40** | **2.36** | **0.23** | **0.21** | **6.13** | **1.00** | **1.00** | **1.00** | **1.00** | **0.24** | 0.26 | **0.22** | 0.23 | **0.22** | 0.23 |
| | Ensemble (XGB) | 22.53 | 0.34 | 31.55 | 0.45 | 16.57 | 0.27 | 1.35 | 7.06 | 1.00 | 1.00 | 1.00 | 1.00 | 0.52 | 0.21 | 0.45 | 0.19 | 0.45 | 0.19 |
| | Ensemble (LGB) | 140.75 | 0.99 | 160.96 | 1.18 | 144.00 | 1.04 | 9.42 | 22.64 | 0.97 | 0.97 | 1.00 | 1.00 | 0.57 | 0.21 | 0.49 | **0.16** | 0.49 | **0.16** |
| | Ensemble (Mix) | 26.29 | 0.39 | 36.45 | 0.49 | 19.65 | 0.33 | 1.55 | 7.87 | 1.00 | 0.99 | 1.00 | 1.00 | 0.52 | **0.18** | 0.45 | 0.16 | 0.46 | 0.16 |

Table 6: Various performance metrics of surrogates evaluated to predict latencies and energies on GPT-S for different CPUs. The arrow on the side of the metric name indicates if lower or higher is better.

| CPU | Surrogate | Accuracy | | | | | | | | | | | | Calibration | | | | | |
| | | MAE ↓ | | RMSE ↓ | | MDAE ↓ | | MARPD ↓ | | $R^2$ ↑ | | Corr. ↑ | | RMS Cal. ↓ | | MA Cal. ↓ | | Miscal. Area ↓ | |
| | | latencies | energies | latencies | energies | latencies | energies | latencies | energies | latencies | energies | latencies | energies | latencies | energies | latencies | energies | latencies | energies |
|---|---|---|---|---|---|---|---|---|---|---|---|---|---|---|---|---|---|---|---|
| Xeon Silver | **AutoGluon** | **1534.66** | **8.94** | **1982.43** | **11.88** | **1242.68** | **6.52** | **3.89** | **11.69** | **0.99** | **0.91** | **1.00** | **0.95** | **0.14** | **0.50** | **0.13** | **0.43** | **0.13** | **0.43** |
| | **Ensemble (XGB)** | 1606.20 | 9.14 | 2107.51 | 12.21 | 1297.02 | 6.92 | 4.11 | 12.08 | 0.99 | 0.90 | 0.99 | 0.95 | 0.21 | 0.53 | 0.19 | 0.45 | 0.20 | 0.46 |
| | **Ensemble (LGB)** | 3243.75 | 10.09 | 4020.23 | 13.66 | 2833.39 | 8.42 | 9.15 | 14.42 | 0.96 | 0.88 | 0.99 | 0.95 | 0.39 | 0.54 | 0.34 | 0.46 | 0.35 | 0.47 |
| | **Ensemble (Mix)** | 1633.05 | 9.05 | 2146.05 | 12.09 | 1285.48 | 6.95 | 4.14 | 11.81 | 0.99 | 0.90 | 0.99 | 0.95 | 0.22 | 0.52 | 0.20 | 0.45 | 0.20 | 0.46 |
| Xeon Gold | **AutoGluon** | **1161.51** | **2.62** | **1634.77** | **3.37** | **833.97** | **2.08** | **6.49** | **9.40** | **0.96** | **0.92** | **0.98** | **0.96** | 0.12 | **0.02** | 0.10 | **0.02** | 0.10 | **0.02** |
| | **Ensemble (XGB)** | 1224.12 | 2.69 | 1695.22 | 3.47 | 896.84 | 2.19 | 6.88 | 9.66 | 0.95 | 0.92 | 0.98 | 0.96 | **0.06** | 0.04 | **0.05** | 0.04 | **0.05** | 0.04 |
| | **Ensemble (LGB)** | 1690.01 | 3.20 | 2144.80 | 3.98 | 1456.95 | 2.78 | 9.58 | 11.52 | 0.93 | 0.89 | 0.98 | 0.96 | 0.09 | 0.12 | 0.09 | 0.11 | 0.09 | 0.11 |
| | **Ensemble (Mix)** | 1200.12 | 2.65 | 1677.90 | 3.42 | 874.13 | 2.16 | 6.68 | 9.47 | 0.95 | 0.92 | 0.98 | 0.96 | 0.07 | 0.04 | 0.06 | 0.04 | 0.06 | 0.04 |
| AMD 7502 | **AutoGluon** | **1570.47** | **15.84** | **2297.45** | **22.15** | **1021.43** | **11.24** | **3.84** | **4.89** | **0.99** | **0.98** | **0.99** | **0.99** | **0.17** | **0.35** | **0.15** | **0.31** | **0.15** | **0.31** |
| | **Ensemble (XGB)** | 1697.98 | 16.83 | 2420.66 | 23.22 | 1107.53 | 12.16 | 4.12 | 5.16 | 0.99 | 0.98 | 0.99 | 0.99 | 0.20 | 0.39 | 0.17 | 0.34 | 0.17 | 0.35 |
| | **Ensemble (LGB)** | 3764.57 | 31.28 | 4531.32 | 38.53 | 3489.26 | 26.99 | 9.94 | 10.26 | 0.96 | 0.95 | 0.99 | 0.99 | 0.45 | 0.51 | 0.39 | 0.44 | 0.40 | 0.44 |
| | **Ensemble (Mix)** | 1693.33 | 16.82 | 2425.90 | 23.07 | 1110.10 | 12.23 | 4.13 | 5.21 | 0.99 | 0.98 | 0.99 | 0.99 | 0.20 | 0.41 | 0.17 | 0.36 | 0.17 | 0.36 |
| AMD 7513 | **AutoGluon** | **3321.83** | **150.24** | **5518.96** | **221.10** | **2174.08** | **103.02** | **2.76** | **10.55** | **0.99** | **0.93** | **1.00** | **0.96** | **0.38** | **0.53** | **0.34** | **0.46** | **0.34** | **0.46** |
| | **Ensemble (XGB)** | 3761.37 | 157.08 | 6054.88 | 230.02 | 2633.74 | 108.79 | 3.23 | 11.14 | 0.99 | 0.92 | 1.00 | 0.96 | 0.41 | 0.54 | 0.36 | 0.46 | 0.36 | 0.47 |
| | **Ensemble (LGB)** | 11204.40 | 187.74 | 13456.74 | 268.02 | 10465.21 | 150.15 | 10.68 | 14.91 | 0.96 | 0.89 | 1.00 | 0.96 | 0.55 | 0.54 | 0.47 | 0.47 | 0.48 | 0.47 |
| | **Ensemble (Mix)** | 3852.39 | 154.29 | 6098.78 | 226.36 | 2635.35 | 104.72 | 3.22 | 10.85 | 0.99 | 0.92 | 1.00 | 0.96 | 0.41 | 0.54 | 0.36 | 0.47 | 0.36 | 0.47 |
| AMD 7452 | **AutoGluon** | **4879.20** | **14.75** | **6100.78** | **18.95** | 3820.30 | 11.39 | **9.02** | **9.21** | **0.95** | **0.95** | **0.98** | **0.97** | **0.41** | **0.47** | **0.36** | **0.41** | **0.37** | **0.41** |
| | **Ensemble (XGB)** | 4930.64 | 15.06 | 6253.26 | 19.54 | 3817.53 | 11.60 | 9.16 | 9.46 | 0.95 | 0.95 | 0.98 | 0.97 | 0.44 | 0.47 | 0.39 | 0.41 | 0.39 | 0.42 |
| | **Ensemble (LGB)** | 5907.72 | 18.06 | 7569.56 | 23.38 | 4669.03 | 14.40 | 11.97 | 12.29 | 0.93 | 0.92 | 0.98 | 0.97 | 0.45 | 0.48 | 0.40 | 0.41 | 0.40 | 0.42 |
| | **Ensemble (Mix)** | 4903.38 | 14.94 | 6172.90 | 19.31 | **3738.50** | **11.25** | 9.12 | 9.36 | 0.95 | 0.95 | 0.98 | 0.97 | 0.46 | 0.48 | 0.40 | 0.42 | 0.40 | 0.42 |

Table 7: Various performance metrics of surrogates evaluated to predict latencies and energies on GPT-M for different GPUs. The arrow on the side of the metric name indicates if lower or higher is better.

| GPU | Surrogate | Accuracy | | | | | | | | | | | | Calibration | | | | | |
|---|---|---|---|---|---|---|---|---|---|---|---|---|---|---|---|---|---|---|---|
| | | MAE ↓ | | RMSE ↓ | | MDAE ↓ | | MARPD ↓ | | $R^2$ ↑ | | Corr. ↑ | | RMS Cal. ↓ | | MA Cal. ↓ | | Miscal. Area ↓ | |
| | | latencies | energies | latencies | energies | latencies | energies | latencies | energies | latencies | energies | latencies | energies | latencies | energies | latencies | energies | latencies | energies |
| A100 | AutoGluon | **2.31** | **0.10** | **3.59** | **0.36** | **1.17** | **0.06** | **0.74** | **6.50** | **1.00** | **0.84** | **1.00** | **0.92** | **0.45** | 0.40 | **0.39** | 0.35 | **0.39** | 0.36 |
| | Ensemble (XGB) | 3.35 | 0.13 | 4.66 | 0.39 | 2.30 | 0.07 | 1.17 | 7.94 | 1.00 | 0.81 | 1.00 | 0.90 | 0.50 | 0.36 | 0.43 | 0.32 | 0.44 | 0.32 |
| | Ensemble (LGB) | 13.88 | 0.16 | 16.12 | 0.39 | 13.82 | 0.13 | 5.37 | 12.00 | 0.97 | 0.81 | 1.00 | 0.91 | 0.57 | **0.24** | 0.49 | **0.21** | 0.49 | **0.22** |
| | Ensemble (Mix) | 3.21 | 0.12 | 4.50 | 0.37 | 2.21 | 0.07 | 1.11 | 7.32 | 1.00 | 0.82 | 1.00 | 0.91 | 0.48 | 0.37 | 0.42 | 0.33 | 0.43 | 0.33 |
| A40 | AutoGluon | **1.12** | **0.26** | **1.68** | **0.35** | **0.67** | **0.18** | **0.28** | **6.38** | **1.00** | **0.98** | **1.00** | **0.99** | **0.45** | 0.20 | **0.39** | 0.18 | **0.40** | 0.18 |
| | Ensemble (XGB) | 4.16 | 0.27 | 5.58 | 0.36 | 3.19 | 0.20 | 1.17 | 6.82 | 1.00 | 0.98 | 1.00 | 0.99 | 0.54 | 0.18 | 0.47 | 0.16 | 0.47 | 0.16 |
| | Ensemble (LGB) | 23.39 | 0.48 | 26.62 | 0.56 | 24.93 | 0.45 | 7.11 | 13.17 | 0.97 | 0.95 | 1.00 | 0.99 | 0.57 | **0.08** | 0.49 | **0.08** | 0.49 | **0.08** |
| | Ensemble (Mix) | 4.01 | 0.27 | 5.47 | 0.36 | 2.94 | 0.19 | 1.09 | 6.63 | 1.00 | 0.98 | 1.00 | 0.99 | 0.54 | 0.19 | 0.46 | 0.17 | 0.47 | 0.1 |
| H100 | AutoGluon | **0.27** | **0.05** | **0.37** | **0.07** | **0.21** | **0.04** | **0.24** | **4.57** | **1.00** | **0.95** | **1.00** | **0.98** | **0.22** | 0.24 | **0.19** | 0.21 | **0.20** | 0.21 |
| | Ensemble (XGB) | 0.91 | 0.05 | 1.19 | 0.07 | 0.74 | 0.04 | 0.81 | 4.72 | 1.00 | 0.95 | 1.00 | 0.98 | 0.45 | 0.24 | 0.39 | 0.22 | 0.40 | 0.22 |
| | Ensemble (LGB) | 4.49 | 0.07 | 5.14 | 0.09 | 4.54 | 0.06 | 4.04 | 6.25 | 0.97 | 0.92 | 1.00 | 0.98 | 0.56 | 0.31 | 0.48 | 0.27 | 0.48 | 0.27 |
| | Ensemble (Mix) | 0.87 | 0.05 | 1.15 | 0.07 | 0.68 | 0.04 | 0.76 | 4.68 | 1.00 | 0.95 | 1.00 | 0.98 | 0.45 | 0.24 | 0.39 | 0.21 | 0.40 | 0.22 |
| RTX2080 | AutoGluon | **9.13** | **0.17** | **12.76** | **0.23** | **6.70** | **0.12** | **1.36** | **3.04** | **1.00** | **1.00** | **1.00** | **1.00** | **0.11** | 0.35 | **0.07** | 0.31 | **0.07** | 0.31 |
| | Ensemble (XGB) | 11.91 | 0.19 | 16.15 | 0.27 | 8.86 | 0.14 | 1.83 | 3.35 | 0.99 | 1.00 | 1.00 | 1.00 | 0.16 | 0.32 | 0.12 | 0.28 | 0.12 | 0.29 |
| | Ensemble (LGB) | 34.99 | 0.60 | 41.31 | 0.70 | 34.08 | 0.57 | 5.73 | 9.74 | 0.97 | 0.97 | 1.00 | 1.00 | 0.37 | **0.09** | 0.33 | **0.07** | 0.33 | **0.07** |
| | Ensemble (Mix) | 11.58 | 0.20 | 15.82 | 0.27 | 8.78 | 0.15 | 1.76 | 3.43 | 0.99 | 1.00 | 1.00 | 1.00 | 0.16 | 0.32 | 0.12 | 0.28 | 0.13 | 0.28 |
| RTX3080 | AutoGluon | **2.90** | **0.82** | **4.62** | **1.06** | **1.70** | **0.59** | **0.54** | 16.97 | **1.00** | **0.93** | **1.00** | **0.97** | **0.14** | **0.20** | **0.11** | **0.18** | **0.11** | **0.19** |
| | Ensemble (XGB) | 5.82 | 0.83 | 8.10 | 1.08 | 4.52 | 0.63 | 1.28 | 17.18 | 1.00 | 0.93 | 1.00 | 0.96 | 0.26 | 0.24 | 0.21 | 0.21 | 0.21 | 0.21 |
| | Ensemble (LGB) | 34.62 | 1.00 | 38.99 | 1.28 | 36.60 | 0.87 | 7.87 | 22.33 | 0.97 | 0.90 | 1.00 | 0.96 | 0.55 | 0.32 | 0.47 | 0.27 | 0.48 | 0.27 |
| | Ensemble (Mix) | 5.95 | 0.83 | 8.17 | 1.07 | 4.63 | 0.63 | 1.30 | **16.84** | 1.00 | 0.93 | 1.00 | 0.96 | 0.27 | 0.24 | 0.22 | 0.21 | 0.22 | 0.21 |
| A6000 | AutoGluon | **0.70** | **0.11** | **1.06** | **0.18** | **0.46** | **0.07** | **0.21** | **3.97** | **1.00** | **0.99** | **1.00** | **0.99** | **0.32** | 0.20 | **0.28** | 0.18 | **0.28** | 0.18 |
| | Ensemble (XGB) | 3.66 | 0.12 | 4.93 | 0.19 | 2.82 | 0.07 | 1.16 | 4.25 | 1.00 | 0.99 | 1.00 | 0.99 | 0.52 | 0.18 | 0.45 | 0.16 | 0.46 | 0.16 |
| | Ensemble (LGB) | 20.92 | 0.27 | 23.81 | 0.34 | 22.15 | 0.23 | 7.19 | 10.84 | 0.97 | 0.96 | 1.00 | 0.99 | 0.57 | 0.21 | 0.49 | 0.19 | 0.49 | 0.19 |
| | Ensemble (Mix) | 3.52 | 0.14 | 4.86 | 0.20 | 2.57 | 0.10 | 1.08 | 5.25 | 1.00 | 0.99 | 1.00 | 0.99 | 0.52 | **0.10** | 0.45 | **0.09** | 0.46 | **0.09** |
| V100 | AutoGluon | **9.43** | **0.16** | **11.64** | **0.35** | **7.92** | **0.05** | **2.60** | **4.46** | **0.99** | **0.99** | **1.00** | **0.99** | **0.37** | **0.08** | **0.33** | **0.07** | **0.33** | **0.07** |
| | Ensemble (XGB) | 10.24 | 0.24 | 12.67 | 0.45 | 8.31 | 0.10 | 2.71 | 7.80 | 0.99 | 0.98 | 1.00 | 0.99 | 0.42 | 0.20 | 0.37 | 0.15 | 0.37 | 0.15 |
| | Ensemble (LGB) | 26.38 | 0.54 | 30.14 | 0.67 | 25.11 | 0.42 | 6.63 | 25.46 | 0.97 | 0.96 | 1.00 | 0.99 | 0.54 | 0.40 | 0.46 | 0.34 | 0.47 | 0.35 |
| | Ensemble (Mix) | 10.46 | 0.32 | 12.69 | 0.46 | 9.43 | 0.23 | 2.81 | 15.49 | 0.99 | 0.98 | 1.00 | 0.99 | 0.42 | 0.31 | 0.37 | 0.25 | 0.37 | 0.25 |
| P100 | AutoGluon | **6.87** | **0.66** | **10.69** | **0.96** | **4.52** | **0.42** | **0.28** | **2.38** | **1.00** | **1.00** | **1.00** | **1.00** | **0.32** | 0.06 | **0.28** | 0.04 | **0.28** | 0.04 |
| | Ensemble (XGB) | 33.86 | 0.73 | 48.02 | 1.06 | 24.22 | 0.46 | 1.39 | 2.63 | 1.00 | 1.00 | 1.00 | 1.00 | 0.52 | **0.03** | 0.45 | **0.03** | 0.46 | **0.03** |
| | Ensemble (LGB) | 213.74 | 2.55 | 241.02 | 3.03 | 224.82 | 2.33 | 10.04 | 11.36 | 0.97 | 0.97 | 1.00 | 1.00 | 0.57 | 0.43 | 0.49 | 0.38 | 0.49 | 0.38 |
| | Ensemble (Mix) | 38.65 | 0.89 | 52.59 | 1.21 | 28.81 | 0.69 | 1.60 | 3.61 | 1.00 | 0.99 | 1.00 | 1.00 | 0.53 | 0.08 | 0.46 | 0.06 | 0.47 | 0.07 |

Table 8: Various performance metrics of surrogates evaluated to predict latencies and energies on GPT-M for different CPUs. The arrow on the side of the metric name indicates if lower or higher is better.

| CPU | Surrogate | Accuracy | | | | | | | | | | | | Calibration | | | | | |
|---|---|---|---|---|---|---|---|---|---|---|---|---|---|---|---|---|---|---|---|
| | | MAE ↓ | | RMSE ↓ | | MDAE ↓ | | MARPD ↓ | | R² ↑ | | Corr. ↑ | | RMS Cal. ↓ | | MA Cal. ↓ | | Miscal. Area ↓ | |
| | | latencies | energies | latencies | energies | latencies | energies | latencies | energies | latencies | energies | latencies | energies | latencies | energies | latencies | energies | latencies | energies |
| Xeon Silver | **AutoGluon** | **1713.09** | **13.02** | **2355.46** | **16.40** | **1261.81** | 10.12 | **3.50** | **11.85** | **0.99** | **0.94** | **1.00** | **0.97** | **0.08** | 0.54 | **0.07** | 0.46 | **0.07** | 0.47 |
| | **Ensemble (XGB)** | 1867.00 | 13.19 | 2544.82 | 16.74 | 1379.22 | 10.25 | 3.78 | 12.04 | 0.99 | 0.93 | 1.00 | 0.97 | 0.14 | **0.54** | 0.12 | **0.46** | 0.12 | **0.47** |
| | **Ensemble (LGB)** | 5077.16 | 15.70 | 5981.26 | 20.00 | 4830.55 | 13.47 | 10.78 | 15.19 | 0.97 | 0.91 | 1.00 | 0.97 | 0.44 | 0.54 | 0.38 | 0.46 | 0.39 | 0.47 |
| | **Ensemble (Mix)** | 1913.78 | 13.05 | 2597.53 | 16.48 | 1464.40 | **9.92** | 3.87 | 11.93 | 0.99 | 0.94 | 1.00 | 0.97 | 0.14 | 0.54 | 0.12 | 0.46 | 0.12 | 0.47 |
| Xeon Gold | **AutoGluon** | **997.52** | **1.78** | **1342.44** | **2.36** | **749.29** | **1.37** | **4.66** | **5.51** | **0.99** | **0.98** | **0.99** | **0.99** | 0.21 | **0.02** | 0.19 | **0.02** | 0.19 | **0.02** |
| | **Ensemble (XGB)** | 1059.66 | 1.88 | 1399.18 | 2.46 | 835.65 | 1.49 | 5.00 | 5.82 | 0.99 | 0.98 | 0.99 | 0.99 | 0.19 | 0.02 | 0.17 | 0.02 | 0.17 | 0.02 |
| | **Ensemble (LGB)** | 2038.55 | 3.23 | 2439.66 | 3.88 | 1886.11 | 2.99 | 9.75 | 9.79 | 0.96 | 0.96 | 0.99 | 0.99 | **0.08** | 0.22 | **0.07** | 0.20 | **0.07** | 0.20 |
| | **Ensemble (Mix)** | 1042.62 | 1.88 | 1395.96 | 2.47 | 795.68 | 1.47 | 4.81 | 5.74 | 0.99 | 0.98 | 0.99 | 0.99 | 0.20 | 0.02 | 0.18 | 0.02 | 0.18 | 0.02 |
| AMD 7502 | **AutoGluon** | **2793.38** | **22.27** | **3832.68** | **29.64** | **2039.52** | **16.81** | **4.75** | **4.81** | **0.99** | **0.99** | **0.99** | **0.99** | **0.36** | **0.45** | **0.32** | **0.40** | **0.32** | **0.40** |
| | **Ensemble (XGB)** | 2889.71 | 22.93 | 3927.64 | 30.61 | 2083.31 | 17.06 | 4.95 | 4.98 | 0.99 | 0.99 | 0.99 | 0.99 | 0.41 | 0.47 | 0.36 | 0.41 | 0.36 | 0.41 |
| | **Ensemble (LGB)** | 5664.76 | 44.12 | 6951.41 | 54.57 | 5092.86 | 39.69 | 11.17 | 11.02 | 0.96 | 0.96 | 0.99 | 0.99 | 0.51 | 0.51 | 0.44 | 0.44 | 0.44 | 0.45 |
| | **Ensemble (Mix)** | 2888.22 | 22.81 | 3930.69 | 30.58 | 2101.56 | 16.93 | 4.95 | 4.94 | 0.99 | 0.99 | 0.99 | 0.99 | 0.41 | 0.46 | 0.36 | 0.40 | 0.36 | 0.40 |
| AMD 7513 | **AutoGluon** | **3297.57** | **315.32** | **6512.33** | **708.51** | **1709.93** | **174.70** | **1.92** | **14.36** | **1.00** | **0.74** | **1.00** | **0.86** | **0.46** | 0.55 | **0.40** | 0.48 | **0.40** | 0.48 |
| | **Ensemble (XGB)** | 4020.05 | 344.26 | 7071.85 | 724.33 | 2639.66 | 200.98 | 2.55 | 15.96 | 1.00 | 0.72 | 1.00 | 0.85 | 0.49 | 0.55 | 0.43 | 0.48 | 0.43 | 0.48 |
| | **Ensemble (LGB)** | 15744.34 | 367.84 | 18417.64 | 744.75 | 15733.28 | 256.65 | 11.25 | 19.37 | 0.97 | 0.71 | 1.00 | 0.85 | 0.57 | 0.56 | 0.49 | 0.48 | 0.49 | 0.49 |
| | **Ensemble (Mix)** | 4390.21 | 329.13 | 7277.61 | 717.62 | 3043.23 | 183.58 | 2.72 | 14.82 | 0.99 | 0.73 | 1.00 | 0.85 | 0.50 | **0.55** | 0.44 | **0.48** | 0.44 | **0.48** |
| AMD 7452 | **AutoGluon** | **6422.40** | **19.16** | **8400.06** | **25.91** | **4603.69** | **13.53** | **8.76** | **8.80** | **0.96** | **0.96** | **0.98** | **0.98** | **0.40** | **0.43** | **0.34** | **0.37** | **0.35** | **0.38** |
| | **Ensemble (XGB)** | 6583.99 | 19.77 | 8650.39 | 26.64 | 4800.54 | 14.12 | 9.01 | 9.17 | 0.96 | 0.96 | 0.98 | 0.98 | 0.46 | 0.47 | 0.40 | 0.41 | 0.41 | 0.42 |
| | **Ensemble (LGB)** | 8414.41 | 25.28 | 11082.78 | 33.29 | 6746.91 | 20.86 | 13.18 | 13.17 | 0.93 | 0.93 | 0.98 | 0.98 | 0.49 | 0.51 | 0.43 | 0.44 | 0.43 | 0.44 |
| | **Ensemble (Mix)** | 6535.60 | 19.66 | 8556.66 | 26.48 | 5013.65 | 14.41 | 8.93 | 9.06 | 0.96 | 0.96 | 0.98 | 0.98 | 0.46 | 0.47 | 0.40 | 0.41 | 0.40 | 0.41 |

Table 9: Various performance metrics of surrogates evaluated to predict latencies and energies on GPT-L for different GPUs. The arrow on the side of the metric name indicates if lower or higher is better.

| GPU | Surrogate | Accuracy | | | | | | | | | | | | Calibration | | | | | |
| | | MAE ↓ | | RMSE ↓ | | MDAE ↓ | | MARPD ↓ | | R² ↑ | | Corr. ↑ | | RMS Cal. ↓ | | MA Cal. ↓ | | Miscal. Area ↓ | |
| | | latencies | energies | latencies | energies | latencies | energies | latencies | energies | latencies | energies | latencies | energies | latencies | energies | latencies | energies | latencies | energies |
|---|---|---|---|---|---|---|---|---|---|---|---|---|---|---|---|---|---|---|---|
| A100 | **AutoGluon** | **1.43** | **0.34** | **1.97** | **0.56** | **1.14** | **0.16** | **0.70** | **6.63** | **1.00** | **0.98** | **1.00** | **0.99** | 0.31 | 0.39 | 0.24 | 0.34 | 0.24 | 0.34 |
| | **Ensemble (XGB)** | 2.07 | 0.36 | 2.85 | 0.58 | 1.55 | 0.18 | 1.00 | 7.55 | 1.00 | 0.97 | 1.00 | 0.99 | 0.30 | 0.35 | 0.22 | 0.31 | 0.23 | 0.32 |
| | **Ensemble (LGB)** | 9.28 | 0.67 | 10.63 | 0.84 | 9.24 | 0.54 | 4.75 | 18.27 | 0.97 | 0.95 | 1.00 | 0.99 | 0.46 | **0.13** | 0.41 | **0.10** | 0.41 | **0.10** |
| | **Ensemble (Mix)** | 2.16 | 0.38 | 2.94 | 0.58 | 1.68 | 0.21 | 1.04 | 8.26 | 1.00 | 0.97 | 1.00 | 0.99 | **0.29** | 0.34 | **0.22** | 0.30 | **0.22** | 0.31 |
| A40 | **AutoGluon** | **0.61** | **0.18** | **0.90** | **0.23** | **0.41** | **0.15** | **0.27** | **2.43** | **1.00** | **1.00** | **1.00** | **1.00** | **0.44** | 0.38 | **0.39** | 0.34 | **0.39** | 0.35 |
| | **Ensemble (XGB)** | 2.44 | 0.23 | 3.37 | 0.28 | 1.81 | 0.19 | 1.11 | 2.88 | 1.00 | 1.00 | 1.00 | 1.00 | 0.54 | 0.36 | 0.47 | 0.32 | 0.47 | 0.32 |
| | **Ensemble (LGB)** | 15.23 | 0.83 | 17.17 | 0.95 | 16.50 | 0.87 | 7.69 | 10.53 | 0.97 | 0.97 | 1.00 | 1.00 | 0.57 | **0.06** | 0.49 | **0.05** | 0.49 | **0.05** |
| | **Ensemble (Mix)** | 2.41 | 0.22 | 3.34 | 0.28 | 1.74 | 0.19 | 1.07 | 2.79 | 1.00 | 1.00 | 1.00 | 1.00 | 0.54 | 0.36 | 0.47 | 0.32 | 0.47 | 0.33 |
| H100 | **AutoGluon** | **0.15** | **0.13** | **0.21** | **0.17** | **0.11** | 0.09 | **0.15** | **5.37** | **1.00** | **0.96** | **1.00** | **0.98** | **0.22** | 0.16 | **0.20** | 0.14 | **0.20** | 0.14 |
| | **Ensemble (XGB)** | 0.62 | 0.13 | 0.83 | 0.18 | 0.47 | 0.09 | 0.63 | 5.50 | 1.00 | 0.96 | 1.00 | 0.98 | 0.48 | 0.16 | 0.42 | 0.14 | 0.42 | 0.14 |
| | **Ensemble (LGB)** | 3.61 | 0.18 | 4.11 | 0.23 | 3.79 | 0.15 | 3.84 | 8.09 | 0.97 | 0.93 | 1.00 | 0.98 | 0.56 | 0.27 | 0.48 | 0.25 | 0.49 | 0.25 |
| | **Ensemble (Mix)** | 0.57 | 0.13 | 0.78 | 0.17 | 0.41 | **0.08** | 0.57 | 5.41 | 1.00 | 0.96 | 1.00 | 0.98 | 0.47 | **0.16** | 0.41 | **0.14** | 0.41 | **0.14** |
| RTX2080 | **AutoGluon** | **2.40** | **1.80** | **3.43** | **2.58** | **1.53** | 1.12 | **0.59** | **10.68** | **1.00** | **0.90** | **1.00** | **0.95** | 0.16 | **0.24** | 0.14 | **0.22** | 0.14 | **0.22** |
| | **Ensemble (XGB)** | 5.21 | 1.83 | 6.98 | 2.63 | 4.06 | 1.15 | 1.30 | 10.91 | 1.00 | 0.90 | 1.00 | 0.95 | **0.16** | 0.29 | **0.13** | 0.25 | **0.13** | 0.25 |
| | **Ensemble (LGB)** | 28.00 | 2.09 | 31.84 | 2.92 | 30.68 | 1.42 | 7.61 | 13.14 | 0.97 | 0.88 | 1.00 | 0.95 | 0.35 | 0.35 | 0.32 | 0.30 | 0.32 | 0.31 |
| | **Ensemble (Mix)** | 4.79 | 1.83 | 6.60 | 2.62 | 3.46 | **1.09** | 1.14 | 10.92 | 1.00 | 0.90 | 1.00 | 0.95 | 0.16 | 0.30 | 0.13 | 0.26 | 0.13 | 0.27 |
| RTX3080 | **AutoGluon** | **1.49** | **0.68** | **2.13** | **0.77** | **0.99** | 0.66 | **0.50** | **6.50** | **1.00** | **0.99** | **1.00** | **1.00** | **0.13** | 0.07 | **0.12** | 0.06 | **0.12** | 0.06 |
| | **Ensemble (XGB)** | 3.95 | 0.70 | 5.40 | 0.81 | 2.89 | 0.65 | 1.36 | 6.59 | 1.00 | 0.99 | 1.00 | 1.00 | 0.21 | 0.07 | 0.15 | 0.06 | 0.16 | 0.06 |
| | **Ensemble (LGB)** | 22.68 | 1.46 | 25.65 | 1.71 | 24.92 | 1.42 | 8.86 | 12.08 | 0.97 | 0.96 | 1.00 | 1.00 | 0.52 | 0.12 | 0.45 | 0.11 | 0.46 | 0.11 |
| | **Ensemble (Mix)** | 3.86 | 0.69 | 5.32 | 0.81 | 2.78 | **0.65** | 1.32 | 6.56 | 1.00 | 0.99 | 1.00 | 1.00 | 0.23 | **0.07** | 0.17 | **0.06** | 0.17 | **0.06** |
| A6000 | **AutoGluon** | **1.56** | **0.45** | **2.24** | **0.61** | **1.08** | **0.33** | **0.74** | **7.13** | **1.00** | **0.98** | **1.00** | **0.99** | 0.48 | 0.20 | **0.42** | 0.17 | **0.42** | 0.18 |
| | **Ensemble (XGB)** | 2.22 | 0.47 | 3.06 | 0.63 | 1.57 | 0.34 | 1.11 | 7.21 | 1.00 | 0.98 | 1.00 | 0.99 | 0.51 | 0.19 | 0.44 | 0.17 | 0.45 | 0.17 |
| | **Ensemble (LGB)** | 13.40 | 0.84 | 15.18 | 1.04 | 14.58 | 0.67 | 7.47 | 12.16 | 0.97 | 0.95 | 1.00 | 0.99 | 0.57 | **0.03** | 0.49 | **0.02** | 0.49 | **0.02** |
| | **Ensemble (Mix)** | 2.08 | 0.46 | 2.95 | 0.63 | 1.45 | 0.34 | 0.99 | 7.36 | 1.00 | 0.98 | 1.00 | 0.99 | 0.50 | 0.17 | 0.43 | 0.15 | 0.44 | 0.15 |
| V100 | **AutoGluon** | 11.61 | **0.30** | **13.29** | 2.99 | 11.30 | **0.15** | 4.23 | **3.74** | **0.98** | 0.86 | 0.99 | 0.93 | **0.52** | 0.21 | **0.45** | 0.19 | **0.45** | 0.19 |
| | **Ensemble (XGB)** | **11.50** | 0.44 | 13.66 | 3.02 | **10.46** | 0.25 | **4.16** | 6.43 | 0.98 | 0.86 | 0.99 | 0.93 | 0.54 | **0.05** | 0.47 | **0.05** | 0.47 | **0.05** |
| | **Ensemble (LGB)** | 17.77 | 1.32 | 22.53 | 3.31 | 13.23 | 1.34 | 6.55 | 25.45 | 0.96 | 0.83 | 0.99 | 0.93 | 0.54 | 0.34 | 0.46 | 0.30 | 0.47 | 0.30 |
| | **Ensemble (Mix)** | 11.84 | 0.43 | 13.63 | 3.03 | 10.69 | 0.24 | 4.35 | 6.30 | 0.98 | 0.86 | 0.99 | 0.93 | 0.55 | 0.07 | 0.47 | 0.06 | 0.48 | 0.06 |
| P100 | **AutoGluon** | **5.59** | **0.78** | **7.81** | **1.27** | **4.28** | **0.40** | **0.52** | **1.31** | **1.00** | **1.00** | **1.00** | **1.00** | **0.29** | 0.13 | **0.26** | 0.11 | **0.26** | 0.12 |
| | **Ensemble (XGB)** | 21.84 | 1.17 | 29.68 | 1.77 | 16.52 | 0.73 | 1.61 | 2.18 | 1.00 | 1.00 | 1.00 | 1.00 | 0.49 | **0.06** | 0.43 | **0.06** | 0.43 | **0.06** |
| | **Ensemble (LGB)** | 131.33 | 5.74 | 146.45 | 6.60 | 139.03 | 5.69 | 10.74 | 13.53 | 0.97 | 0.97 | 1.00 | 1.00 | 0.57 | 0.50 | 0.49 | 0.44 | 0.49 | 0.44 |
| | **Ensemble (Mix)** | 23.97 | 1.64 | 31.74 | 2.20 | 18.94 | 1.30 | 1.76 | 3.66 | 1.00 | 1.00 | 1.00 | 1.00 | 0.49 | 0.20 | 0.42 | 0.18 | 0.43 | 0.18 |

Table 10: Various performance metrics of surrogates evaluated to predict latencies and energies on GPT-L for different CPUs. The arrow on the side of the metric name indicates if lower or higher is better.

| CPU | Surrogate | Accuracy | | | | | | | | | | | | Calibration | | | | | |
| | | MAE ↓ | | RMSE ↓ | | MDAE ↓ | | MARPD ↓ | | $R^2$ ↑ | | Corr. ↑ | | RMS Cal. ↓ | | MA Cal. ↓ | | Miscal. Area ↓ | |
| | | latencies | energies | latencies | energies | latencies | energies | latencies | energies | latencies | energies | latencies | energies | latencies | energies | latencies | energies | latencies | energies |
|---|---|---|---|---|---|---|---|---|---|---|---|---|---|---|---|---|---|---|---|
| Xeon Silver | **AutoGluon** | **636.15** | 5.61 | **864.06** | **7.70** | **494.13** | 3.93 | **2.54** | **9.84** | **1.00** | **0.96** | **1.00** | **0.98** | 0.17 | 0.44 | 0.15 | 0.38 | 0.15 | 0.39 |
| | **Ensemble (XGB)** | 753.47 | 5.72 | 1025.05 | 7.81 | 578.99 | 4.01 | 2.96 | 10.02 | 1.00 | 0.96 | 1.00 | 0.98 | 0.08 | **0.42** | 0.07 | **0.36** | 0.07 | **0.37** |
| | **Ensemble (LGB)** | 2824.18 | 8.02 | 3236.41 | 9.97 | 2733.85 | 6.99 | 11.64 | 14.87 | 0.97 | 0.93 | 1.00 | 0.98 | 0.41 | 0.48 | 0.37 | 0.41 | 0.37 | 0.42 |
| | **Ensemble (Mix)** | 794.59 | **5.60** | 1061.38 | 7.73 | 619.52 | **3.82** | 3.12 | 10.00 | 1.00 | 0.96 | 1.00 | 0.98 | **0.06** | 0.42 | **0.05** | 0.36 | **0.06** | 0.37 |
| Xeon Gold | **AutoGluon** | **1808.83** | **2.94** | **4993.72** | **9.06** | **961.39** | **1.64** | **12.44** | **12.52** | **0.65** | **0.60** | **0.81** | **0.77** | **0.04** | **0.22** | **0.03** | **0.20** | **0.03** | **0.20** |
| | **Ensemble (XGB)** | 2071.91 | 3.33 | 5172.30 | 9.21 | 1192.99 | 2.01 | 14.28 | 14.39 | 0.62 | 0.58 | 0.79 | 0.76 | 0.23 | 0.36 | 0.21 | 0.32 | 0.21 | 0.32 |
| | **Ensemble (LGB)** | 2276.83 | 3.72 | 5208.21 | 9.42 | 1546.31 | 2.52 | 16.82 | 17.10 | 0.62 | 0.56 | 0.80 | 0.76 | 0.32 | 0.42 | 0.28 | 0.37 | 0.28 | 0.37 |
| | **Ensemble (Mix)** | 1956.75 | 3.15 | 5056.95 | 9.17 | 1058.07 | 1.74 | 13.15 | 13.29 | 0.64 | 0.59 | 0.80 | 0.77 | 0.21 | 0.35 | 0.18 | 0.31 | 0.18 | 0.31 |
| AMD 7502 | **AutoGluon** | **979.59** | 10.85 | **1401.41** | 15.13 | **637.77** | 7.47 | 3.47 | 4.78 | 0.99 | 0.99 | 1.00 | 0.99 | 0.32 | 0.46 | **0.29** | 0.40 | **0.29** | 0.41 |
| | **Ensemble (XGB)** | 1061.60 | 11.30 | 1488.69 | 15.60 | 747.04 | 8.16 | 3.90 | 5.16 | 0.99 | 0.99 | 1.00 | 0.99 | 0.36 | 0.47 | 0.32 | 0.41 | 0.32 | 0.41 |
| | **Ensemble (LGB)** | 2963.13 | 24.72 | 3487.50 | 29.94 | 2788.52 | 22.54 | 12.75 | 13.26 | 0.96 | 0.96 | 1.00 | 0.99 | 0.53 | 0.54 | 0.46 | 0.47 | 0.46 | 0.47 |
| | **Ensemble (Mix)** | 1077.05 | 11.22 | 1504.92 | 15.53 | 777.37 | 8.19 | 3.88 | 4.97 | 0.99 | 0.99 | 1.00 | 0.99 | 0.34 | **0.44** | 0.30 | **0.38** | 0.31 | **0.39** |
| AMD 7513 | **AutoGluon** | **1341.23** | **138.82** | **2336.42** | **193.98** | **747.38** | **92.12** | **1.55** | 13.84 | **1.00** | **0.91** | **1.00** | **0.96** | **0.48** | **0.55** | **0.42** | **0.48** | **0.42** | **0.48** |
| | **Ensemble (XGB)** | 1821.68 | 141.56 | 2867.87 | 198.08 | 1203.03 | 97.95 | 2.32 | 14.26 | 1.00 | 0.91 | 1.00 | 0.95 | 0.53 | 0.56 | 0.45 | 0.48 | 0.46 | 0.48 |
| | **Ensemble (LGB)** | 8438.54 | 158.86 | 9602.56 | 224.77 | 8622.26 | 129.98 | 12.73 | 18.19 | 0.97 | 0.88 | 1.00 | 0.95 | 0.57 | 0.56 | 0.49 | 0.48 | 0.49 | 0.49 |
| | **Ensemble (Mix)** | 1990.32 | 141.12 | 2967.07 | 197.03 | 1402.25 | 103.07 | 2.56 | 14.02 | 1.00 | 0.91 | 1.00 | 0.95 | 0.53 | 0.56 | 0.46 | 0.48 | 0.46 | 0.49 |
| AMD 7452 | **AutoGluon** | **766.81** | **4.16** | **1101.54** | **6.15** | **535.75** | **2.72** | **2.52** | **4.20** | **1.00** | **0.99** | **1.00** | **1.00** | **0.05** | **0.26** | **0.04** | **0.23** | **0.05** | **0.23** |
| | **Ensemble (XGB)** | 895.53 | 4.40 | 1262.25 | 6.44 | 636.53 | 3.02 | 2.98 | 4.55 | 1.00 | 0.99 | 1.00 | 1.00 | 0.14 | 0.29 | 0.12 | 0.26 | 0.13 | 0.26 |
| | **Ensemble (LGB)** | 3265.62 | 10.37 | 3747.71 | 12.55 | 3294.40 | 9.69 | 12.28 | 12.92 | 0.97 | 0.96 | 1.00 | 0.99 | 0.50 | 0.50 | 0.43 | 0.43 | 0.44 | 0.44 |
| | **Ensemble (Mix)** | 920.19 | 4.45 | 1277.03 | 6.47 | 681.07 | 2.98 | 3.07 | 4.57 | 1.00 | 0.99 | 1.00 | 0.99 | 0.14 | 0.29 | 0.13 | 0.26 | 0.13 | 0.26 |

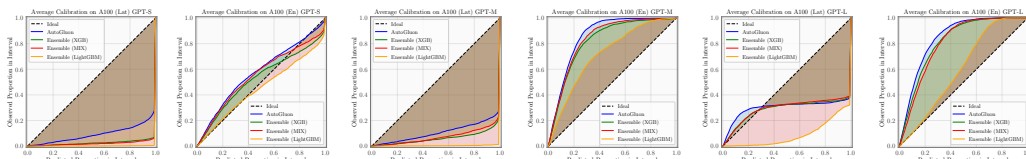

Figure 9: Calibration Areas A100

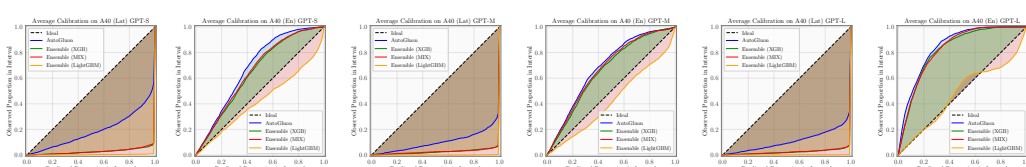

Figure 10: Calibration Areas A40

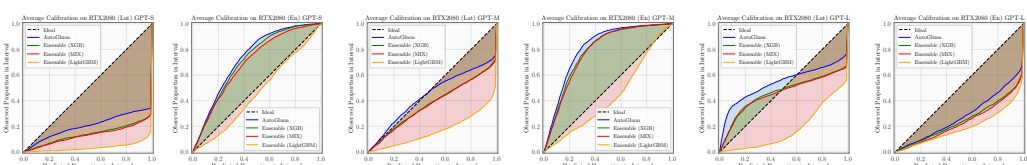

Figure 11: Calibration Areas RTX2080

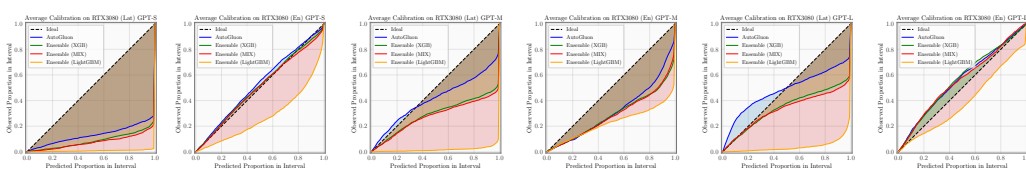

Figure 12: Calibration Areas RTX3080

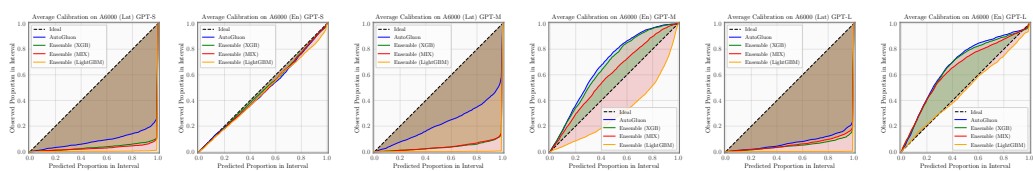

Figure 13: Calibration Areas A6000

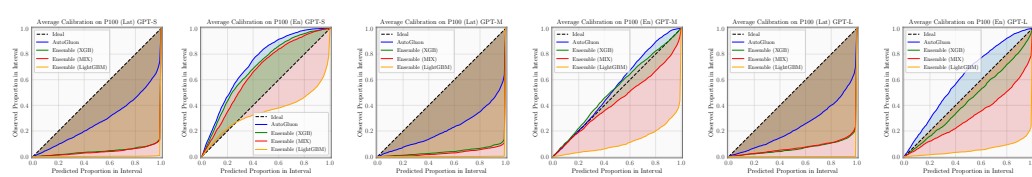

Figure 14: Calibration Areas P100

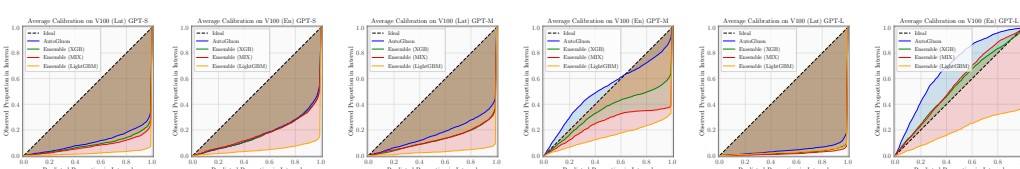

Figure 15: Calibration Areas V100

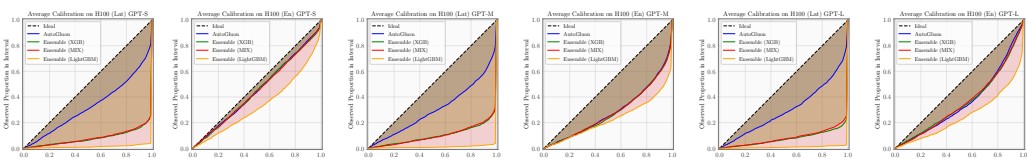

Figure 16: Calibration Areas H100

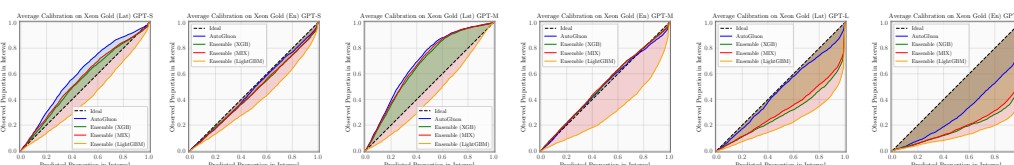

Figure 17: Calibration Areas Xeon Gold

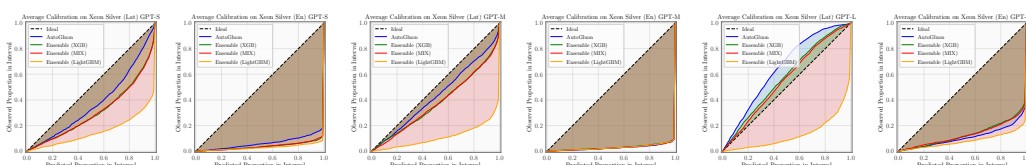

Figure 18: Calibration Areas Xeon Silver

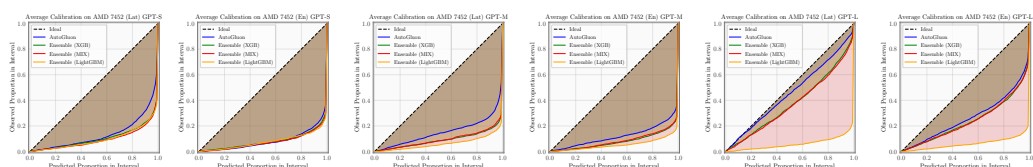

Figure 19: Calibration Areas CPU AMD 7452

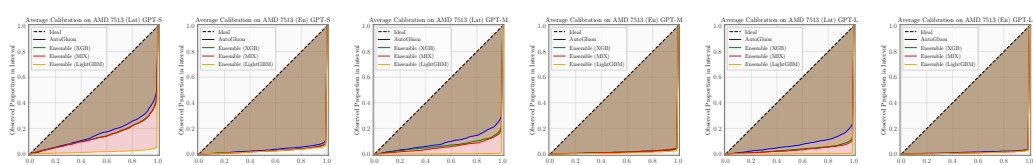

Figure 20: Calibration Areas CPU AMD 7513

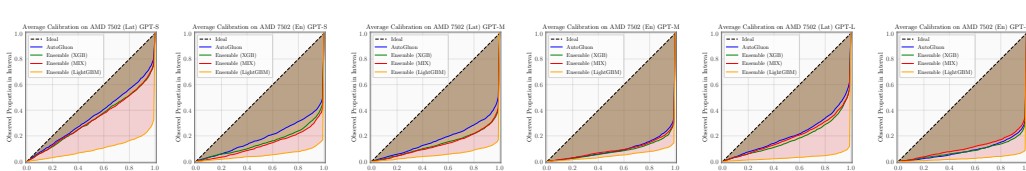

Figure 21: Calibration Areas CPU AMD 7502

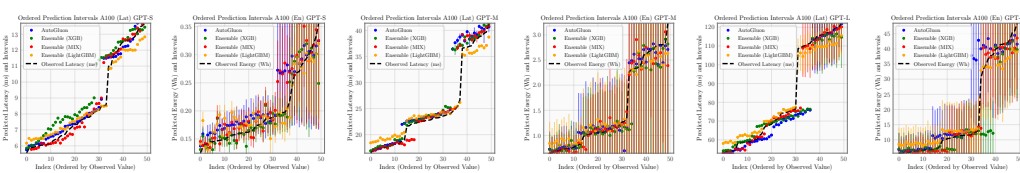

Figure 22: Prediction Intervals A100

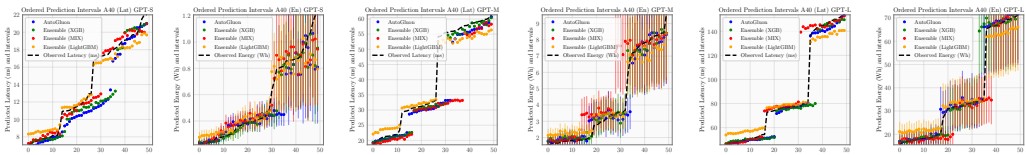

Figure 23: Prediction Intervals A40

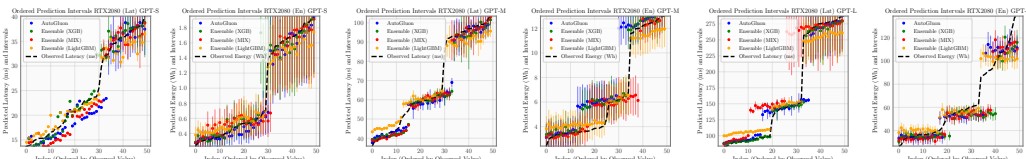

Figure 24: Prediction Intervals RTX2080

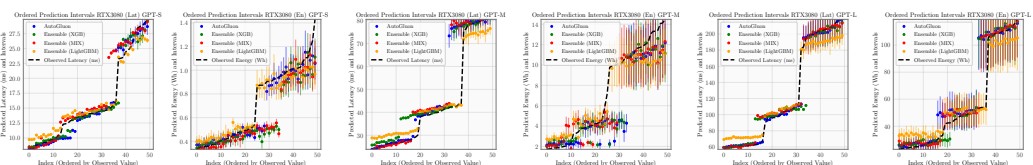

Figure 25: Prediction Intervals RTX3080

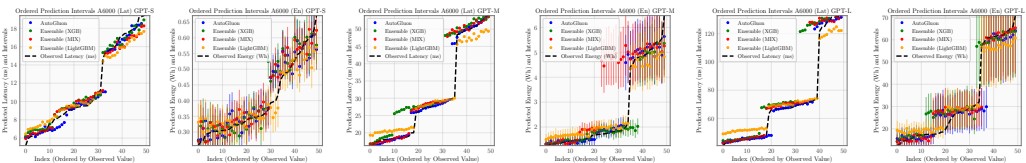

Figure 26: Prediction Intervals A6000

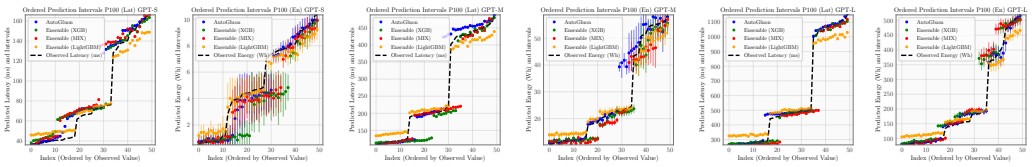

Figure 27: Prediction Intervals P100

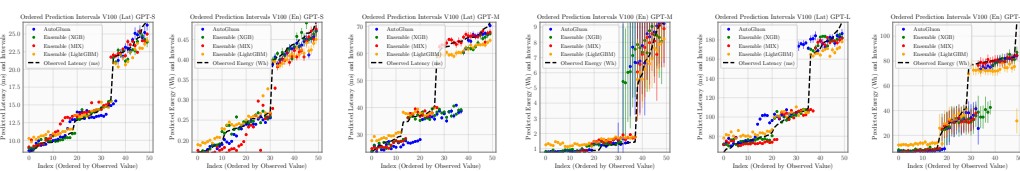

Figure 28: Prediction Intervals V100

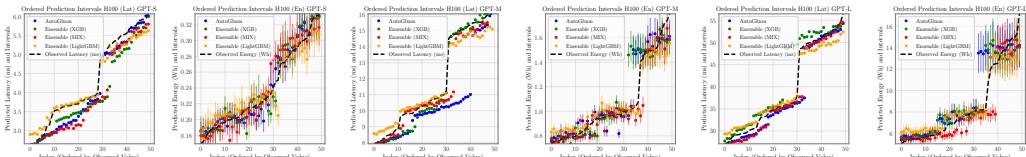

Figure 29: Prediction Intervals H100

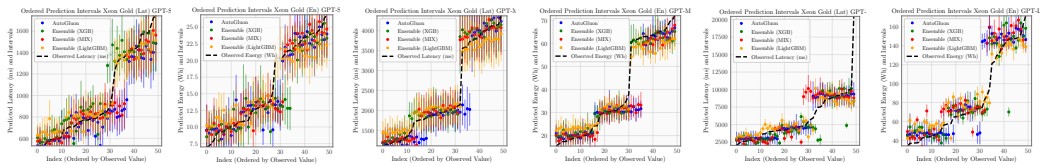

Figure 30: Prediction Intervals Xeon Gold

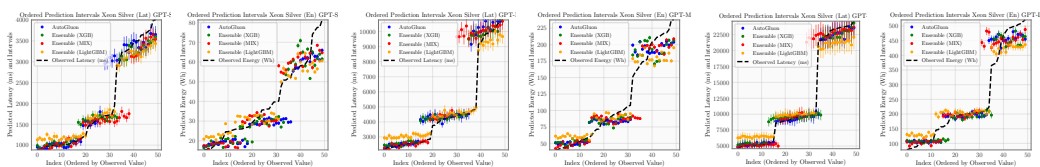

Figure 31: Prediction Intervals Xeon Silver

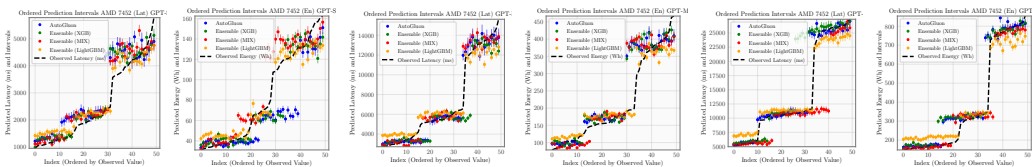

Figure 32: Prediction Intervals CPU AMD 7452

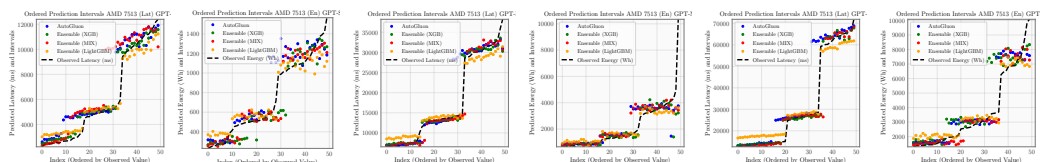

Figure 33: Prediction Intervals CPU AMD 7513

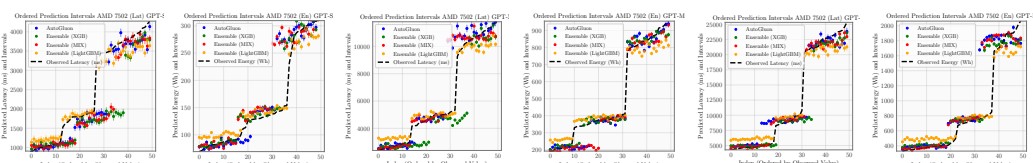

Figure 34: Prediction Intervals CPU AMD 7502

# E Comparison with NAS-Bench-301

Given the immense size of the search spaces (approximately $10^{36}$), training architectures from scratch is impractical. Our work is inspired by surrogate benchmarks such as those proposed by [88]. However, HW-GPT-Bench presents several key differences compared to NB301. First, unlike NB301, which trains architectures from scratch—impractical for larger models and dataset sizes—we utilize an efficient weight-sharing-based supernet. The performance of the inherited subnetwork directly serves as a reliable performance estimation proxy, and these architectures can be fine-tuned if necessary. We also employ a sandwich scheme to train the supernet by sampling the largest, smallest, and a set of random architectures. Second, while NB301 uses the DARTS pre-training pipeline, we introduce three novel search spaces and design a training pipeline specifically for supernet training. Additionally, we focus on the application domain of language modeling, in contrast to NB301's primary focus on image classification. Lastly, unlike NB301, our work supports a range of hardware devices and provides well-calibrated latency predictions. To our knowledge, we are the first to study the calibration of surrogate models for latency predictions.

# F Importance Analysis

In this section, we provide details on the methods we used throughout the paper to analyze and interpret the data collected from the HW-GPT-Bench search space.

## F.1 OLS Covariate Analysis.

Ordinary Least Squares (OLS) is a statistical method used for estimating the parameters of a linear regression model. The primary goal of OLS is to minimize the sum of the squared residuals, which are the differences between the observed values and the values predicted by the model. In the context of OLS, covariate analysis involves examining the relationships between independent variables (covariates) and the dependent variable. This analysis helps to understand how each covariate contributes to the prediction of the dependent variable and the overall model performance. To fit the linear regression model and conduct the analysis we used `statsmodel.regression.linear_model.OLS` [10].

### F.1.1 Key Concepts in OLS

**Linear Regression Model**: The model assumes a linear relationship between the dependent variable $Y$ (perplexity) and one or more independent variables (covariates). The general form of the model is:

$$Y = \beta_0 + \beta_1 e + \beta_2 l + \sum_{i=1}^{l} \beta_{i+2} h^i + \sum_{i=1}^{l} \beta_{i+2+l} m^i + \beta_{2l+3} b + \epsilon,$$

where $\beta_0$ is the intercept, $\beta_i$ (for $i = 1, \ldots, 2l + 3$) are the coefficients of the covariates, and $\epsilon$ represents the error term. $l$ is the number of layers, $e$ is the embedding dimension, $m^i$ and $h^i$ are the MLP ratio and number of heads on layer $i$, respectively, and b is the bias.

**Objective of OLS**: The goal is to estimate the coefficients $\beta$ such that we minimize the sum of the squared residuals (SSR):

$$SSR = \sum_{i=1}^{N} (Y_i - \hat{Y}_i)^2,$$

where $Y_i$ is the observed ground truth perplexity and $\hat{Y}_i$ is the predicted perplexity by the Linear Regression model.

### F.1.2 Covariate Analysis

Covariate analysis in the context of OLS involves investigating the effect of each independent variable (embedding dimension, number of layers, etc.) on the dependent variable, i.e. perplexity. This includes:

---

[10]https://www.statsmodels.org/dev/examples/notebooks/generated/ols.html

- **Estimating Coefficients**: Determining the values of $\beta_1, \beta_2, \ldots, \beta_{2l+3}$ which represent the change in perplexity for a one-unit change in the respective architectural dimension, holding other variables constant.

- **Statistical Significance**: Assessing the significance of each coefficient using t-tests. The null hypothesis $H_0$ states that the coefficient is zero (no effect). The p-value indicates whether the null hypothesis can be rejected.

- **Standard Errors**: Providing a measure of the variability of the coefficient estimates. Smaller standard errors suggest more precise estimates.

- **Goodness-of-Fit**: Evaluating how well the model explains the variability of perplexity using metrics such as R-squared and adjusted R-squared. R-squared indicates the proportion of the variance in perplexity that is predictable from the independent variables.

The results of the OLS analysis for perplexity on the collected samples are presented below. These results include:

1. **Coefficients** ($\beta$): Estimates for each covariate.

2. **Standard Errors**: Indicate the precision of the coefficient estimates.

3. **t-Values**: Used to test the hypothesis that a coefficient is significantly different from zero.

4. **P-Values**: Indicate the significance level of each coefficient.

5. **$R^2$**: The proportion of the variance in the dependent variable explained by the model.

6. **Adjusted R-squared**: Adjusted for the number of predictors in the model, providing a more accurate measure of goodness-of-fit for models with multiple covariates.

```
==============================================================================
                      OLS Regression Results GPT-L
==============================================================================
Dep. Variable:             perplexity   R-squared:                       0.901
Model:                            OLS   Adj. R-squared:                  0.901
Method:                 Least Squares   F-statistic:                 1.817e+04
No. Observations:               10000   AIC:                         3.395e+04
Df Residuals:                    9994   BIC:                         3.399e+04
Df Model:                           5
Covariance Type:            nonrobust
==============================================================================
                   coef    std err          t      P>|t|      [0.025      0.975]
------------------------------------------------------------------------------
const           23.6263      0.013   1788.886      0.000      23.600      23.652
num_layers      -0.0426      0.013     -3.227      0.001      -0.069      -0.017
embed_dim       -3.9787      0.013   -301.209      0.000      -4.005      -3.953
mean_mlp_ratio  -0.1164      0.013     -8.812      0.000      -0.142      -0.091
mean_heads      -0.0612      0.013     -4.634      0.000      -0.087      -0.035
bias            -0.0630      0.013     -4.769      0.000      -0.089      -0.037
==============================================================================
Omnibus:                    75511.319   Durbin-Watson:                   2.026
Prob(Omnibus):                  0.000   Jarque-Bera (JB):             1537.046
Skew:                          -0.635   Prob(JB):                         0.00
Kurtosis:                       1.559   Cond. No.                         1.03
==============================================================================

==============================================================================
                      OLS Regression Results GPT-M
==============================================================================
Dep. Variable:             perplexity   R-squared:                       0.914
Model:                            OLS   Adj. R-squared:                  0.914
Method:                 Least Squares   F-statistic:                 2.117e+04
No. Observations:               10000   AIC:                         3.791e+04
Df Residuals:                    9994   BIC:                         3.796e+04
```

```
Df Model:                          5
Covariance Type:            nonrobust
==============================================================================
                 coef    std err          t      P>|t|      [0.025      0.975]
------------------------------------------------------------------------------
const          28.1305      0.016   1747.009      0.000      28.099      28.162
num_layers     -0.0983      0.016     -6.104      0.000      -0.130      -0.067
embed_dim      -5.2315      0.016   -324.795      0.000      -5.263      -5.200
mean_mlp_ratio -0.1445      0.016     -8.975      0.000      -0.176      -0.113
mean_heads     -0.0744      0.016     -4.623      0.000      -0.106      -0.043
bias           -0.0650      0.016     -4.035      0.000      -0.097      -0.033
==============================================================================
Omnibus:                   66013.096   Durbin-Watson:                   2.021
Prob(Omnibus):                 0.000   Jarque-Bera (JB):             1509.031
Skew:                         -0.600   Prob(JB):                         0.00
Kurtosis:                      1.524   Cond. No.                         1.03
==============================================================================

==============================================================================
                     OLS Regression Results GPT-S
==============================================================================
Dep. Variable:            perplexity   R-squared:                       0.949
Model:                           OLS   Adj. R-squared:                  0.949
Method:                Least Squares   F-statistic:                 3.754e+04
No. Observations:              10000   AIC:                         3.667e+04
Df Residuals:                   9994   BIC:                         3.671e+04
Df Model:                          5
Covariance Type:            nonrobust
==============================================================================
                 coef    std err          t      P>|t|      [0.025      0.975]
------------------------------------------------------------------------------
const          32.3973      0.015   2141.130      0.000      32.368      32.427
num_layers     -0.4912      0.015    -32.459      0.000      -0.521      -0.462
embed_dim      -6.5321      0.015   -431.656      0.000      -6.562      -6.502
mean_mlp_ratio -0.2199      0.015    -14.530      0.000      -0.250      -0.190
mean_heads     -0.2739      0.015    -18.096      0.000      -0.304      -0.244
bias           -0.0251      0.015     -1.661      0.097      -0.055       0.005
==============================================================================
Omnibus:                   88675.889   Durbin-Watson:                   2.044
Prob(Omnibus):                 0.000   Jarque-Bera (JB):             1321.623
Skew:                         -0.544   Prob(JB):                     1.03e-287
Kurtosis:                      1.589   Cond. No.                         1.03
==============================================================================
```

## F.2  Power-laws for GPT-wide spaces

We define the power-law fits for GPT-wide spaces below. Similar to the observation in 4, we see that the embedding size has the most effect on perplexity, followed by number of layers, mlp expansion ratio and number of heads. The bias plays a minimal role in determining perplexity of an architecture.

$$\text{GPT-S-wide:} \quad y = 1116.453 \cdot l^{-0.212} \cdot e^{-0.3770} \cdot m^{-0.0514} \cdot h^{-0.0647} \cdot b^{-0.000190},$$

$$\text{GPT-M-wide:} \quad y = 618.0753 \cdot l^{-0.1795} \cdot e^{-0.3401} \cdot m^{-0.0711} \cdot h^{-0.0556} \cdot b^{-0.0050},$$

$$\text{GPT-L-wide:} \quad y = 498.9920 \cdot l^{-0.1659} \cdot e^{-0.3204} \cdot m^{-0.0692} \cdot h^{-0.053} \cdot b^{-0.0081}.$$

## F.3  Recursive Feature Elimination

Recursive Feature Elimination (RFE) is a feature selection method used in machine learning to identify the most relevant features in a dataset. It works by recursively fitting a model and removing the least important feature(s) based on model coefficients or importance scores until the maximum

number of features is reached. Features in our case are the architectural choices, i.e. embedding dimension size, number of layers, etc. The process involves the following steps:

1. **Model Fitting**: The model is initially trained on the entire set of features.
2. **Feature Ranking**: Features are ranked based on their importance scores derived from the fitted model.
3. **Feature Elimination**: The least important feature(s) are removed from the dataset.
4. **Repetition**: Steps 1-3 are repeated recursively on the pruned feature set until the desired number of features is reached or only a single feature remains.

We implement RFE using `sklearn` [11], which provides an efficient and easy-to-use RFE function. We employ a `RandomForest` regressor as the base model due to its ability to handle high-dimensional data and capture non-linear relationships. Specifically, we use a `RandomForest` regressor with 50 estimators, which balances model complexity and computational efficiency. In Figures 35, 36 and 37, we present the ranking of all search space dimensions for GPT-S, GPT-M and GPT-L spaces.

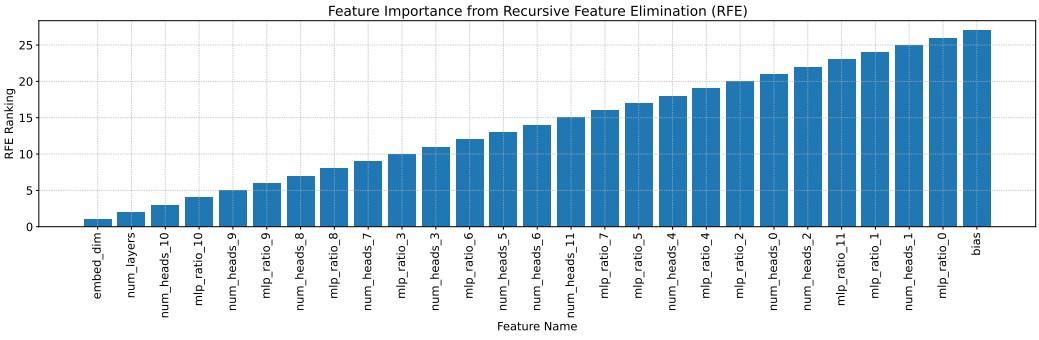

Figure 35: Detailed feature ranking from RFE for GPT-S.

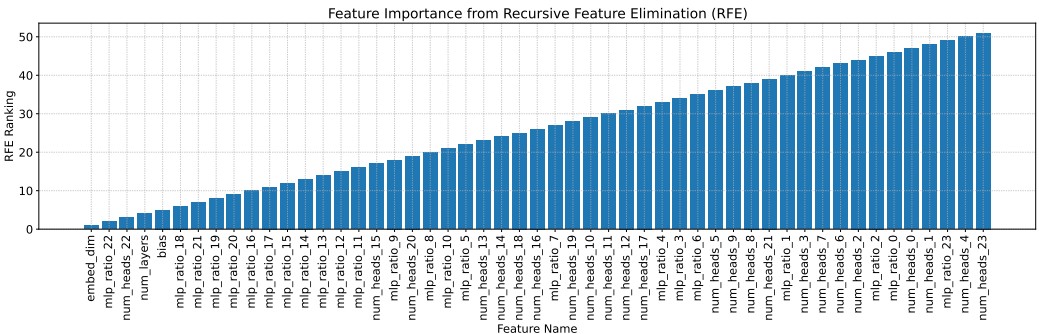

Figure 36: Detailed feature ranking from RFE for GPT-M.

# G Details on Empirical Attainment Function (EAF)

Running multi-objective optimization algorithms multiple times can yield different Pareto fronts. Computing uncertainty estimates of Pareto fronts over multiple runs of an algorithm is important to ensure that we appropriately compare different algorithms in a statistically meaningful way. Simply superimposing the Pareto fronts across multiple runs is insufficient in depicting how the Pareto fronts tend to vary. The Empirical Attainment Function (EAF) [25, 47] is a statistical measure used in

---

[11] https://scikit-learn.org/stable/auto_examples/feature_selection/plot_rfe_digits.html

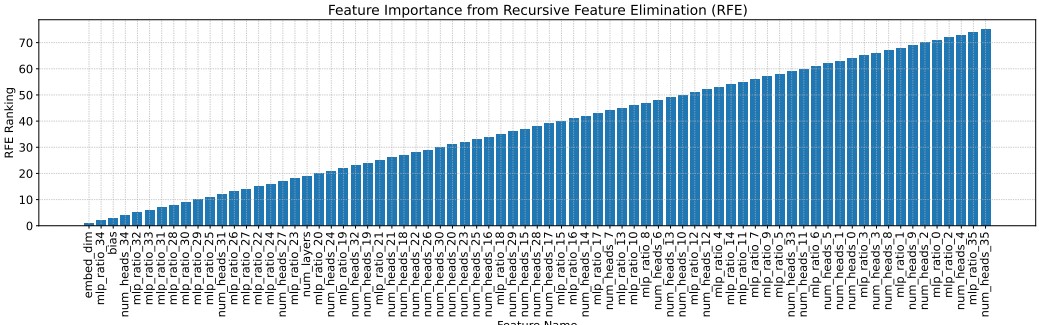

Figure 37: Detailed feature ranking from RFE for GPT-L.

multi-objective optimization to describe the distribution of outcomes achieved by an optimization algorithm over multiple runs. It provides a way to empirically estimate the probability that a given point in the objective space is attained (i.e., dominated or matched) by the solutions generated by the algorithm.

For two solution vectors of the multi-objective function, $\mathbf{y}$ and $\mathbf{z}$, $\mathbf{y}$ weakly dominates $\mathbf{z}$ ($\mathbf{y} \preceq \mathbf{z}$) if the following conditions hold:

1. **Not worse on any objective:** $\mathbf{y}$ is at least as good as $\mathbf{z}$ on all objectives. This means for each objective function, the value in $\mathbf{y}$ is either equal to or better than the corresponding value in $\mathbf{z}$.

2. **Strictly better on at least one objective (or indifferent on all):** $\mathbf{y}$ must be strictly better than $\mathbf{z}$ on at least one objective function. Alternatively, it can be equal on all objectives.

Given a set of Pareto fronts $\{F^1, \ldots, F^n\}$ obtained from $n$ independent runs with different random seeds, the EAF is defined by the following empirical attainment function:

$$\varepsilon(\mathbf{z}) = \frac{1}{n} \sum_{i=1}^{n} \mathbb{I}[F^i \preceq \mathbf{z}],$$

where $\varepsilon(\mathbf{z})$ represents the EAF value for a specific objective vector $\mathbf{z}$ in the objective space. $\mathbb{I}[F^i \preceq \mathbf{z}]$ is an indicator function that is 1 if the objective vector $\mathbf{z}$ is weakly dominated by the $F^i$, i.e. the Pareto front from the i-th run, and 0 otherwise. $F^i \preceq \mathbf{z}$ means that there exists at least an objective vector $\mathbf{y}$ in $F^i$ at least as good as $\mathbf{z}$. This doesn't necessarily mean every single run achieved a better outcome than $\mathbf{z}$, but at least one did. In simpler terms, the EAF value at a given objective vector $\mathbf{z}$ represents the portion of independent runs where a Pareto front achieved at least that good of an objective vector (weakly dominated $\mathbf{z}$).

The EAF can be used to visualize uncertainty in the Pareto front by plotting different EAS levels. For example, $S^{1/2}$ represents the set of objective vectors that are weakly dominated by at least half of the independent runs (50% EAF level). To compute the Empirical Attainment Surfaces (EAS) in this paper we used the implementation from Watanabe [80] [12].

## H   Inheriting v/s Finetuning Subnetworks

We validate the effectiveness of out perplexity surrogate by inheriting randomly sampled 100 subnetworks and comparing the correlation between the perplexity on simply inheriting the subnetworks v/s finetuning the subnetworks further upon inheritance for 5000 update steps. We observe that the inherited subnetwork performance strongly correlates with fine-tuned subnetworks as indicated by Table 11.

## I   Distribution of Architecture Latencies

We fit a kernel-density-estimator to the collected subnetwork latencies. As observed in Figure 38 and 39, the distribution of CPU latency is more noisier than GPU latency.

---

[12] https://github.com/nabenabe0928/empirical-attainment-func

| Supernet | Kendall-Tau |
|----------|-------------|
| GPT-S | 0.9626 |
| GPT-M | 0.9580 |
| GPT-L | 0.9286 |

Table 11: Kendall-Tau values between perplexities after inheriting and perplexities after finetuning (for 5000 steps) for different Supernets for a set of 100 random architectures

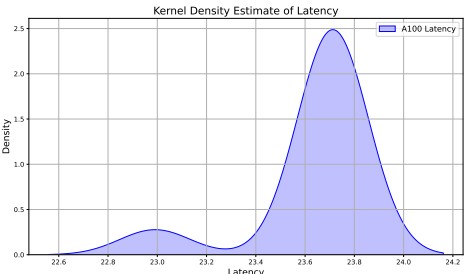

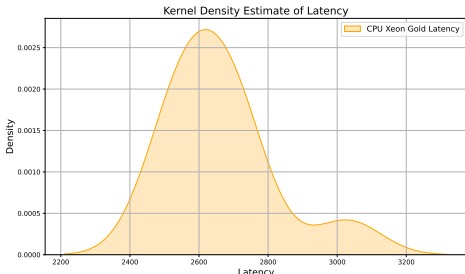

Figure 38: GPU-A100      Figure 39: CPU-Xeon Gold

Figure 40: Distribution of Latencies on different devices

## J   Scatter Plots for GPT-wide

We introduced 4 new search spaces (with more widely spaced choices), mainly GPT-S-wide, GPT-M-wide, GPT-L-wide and GPT-XL-wide and present the scatter plots on the collected ground truth subnetworks for these spaces in Figure 41.

## K   Multi-objective NAS algorithms

In this section, we provide more details on the multi-objective NAS methods we run on HW-GPT-Bench in Section 5. We use their implementation in SyneTune [62][13].

- **Random Search (RS)**. RS has been shown to be a strong baseline for single [40] and multi-objective [10, 31, 69] architecture search. For this baseline we sample architectures uniformly at random from the search space and then compute the Pareto-Front from these architecture samples. In larger search space, random search, while being embarrasingly parallel and often performant, is not guaranteed to yield the optimal architectures.

- **Multi-Objective Regularized Evolution Algorithm (MOREA).** Regularized Evolution (RE) or Aging Evolution [59] has been quite successfully applied for neural architecture search. Regularized Evolution works by evolving a population of candidates using mutation and periodically discarding the oldest members of the population, inducing a *regularization* effect. In SyneTune RE is extended to Multi-Objective Regularized Evolution (MOREA) by scoring the population via a multi-objective priority based on non-dominated sorting. Parents are then be sampled from the population based on this priority score.

- **Non-dominated Sorting Genetic Algorithm II (NSGA-II)**. NSGA-II [16], is a multi-objective evolutionary algorithm to obtain a Pareto-Set of architectures. It employs non-dominated sorting to rank architecures based on their dominance relationships and **crowding distance** to maintain a diverse population. Through selection, crossover, and mutation, NSGA-II iteratively evolves populations toward the Pareto front, offering a range of trade-off solutions.

- **Local Seach (LS).** SyneTune adapts LS to explore the vicinity of Pareto-optimal points in multi-objective optimization problems, aiming to iteratively refine Pareto-optimal solutions solutions within defined neighborhoods. The method is described in more detail in Klein et al. [36].

- **Bayesian Optimization with Random Scalarizations (RSBO).** RSBO [55] uses an acquisition function that takes as input multiple random scalarizations of the objectives being optimized, to obtain the Pareto-optimal set which minimizes the Bayesian regret.

---

[13]https://syne-tune.readthedocs.io/en/latest/getting_started.html#supported-multi-objective-optimization-methods

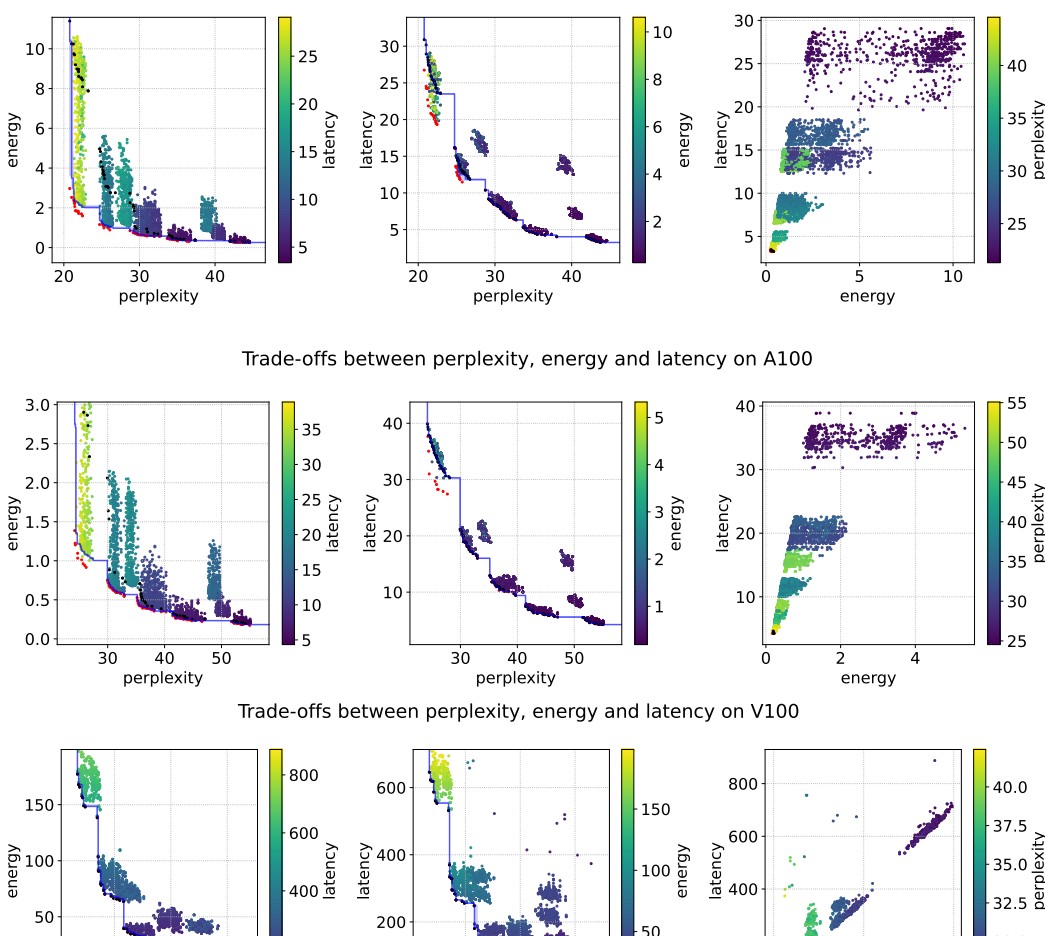

Figure 41: Trade-off scatter plots for GPT-M-wide, GPT-L-wide and GPT-XL-wide

- **Bayesian Optimization with Linear Scalarizations (LSBO).** Similar to RSBO, this method optimizes a single objective corresponding to a fixed linear combination of two objectives instead of randomizing the scalarizations at each BO iteration.

- **Expected Hypervolume Improvement (EHVI).** EHVI [15] is a Bayesian optimization method with an acquisition function designed to efficiently explore the Pareto front in multi-objective optimization problems. It quantifies the expected improvement in hypervolume, which measures the volume of the objective space dominated by Pareto-optimal solutions, that a candidate solution can offer. EHVI guides the search towards regions of the objective space likely to contain better trade-offs, aiding in the discovery of diverse Pareto-optimal solutions.

- **Multi-objective Asynchronous Successive Halving (MOASHA).** MOASHA [65] is a multi-fidelity approach that leverages an asynchronous successive halving scheduler [41] along with non-dominating sorting for budget allocation. It employs the NSGA-II selection mechanism and the $\epsilon$-net [63] exploration strategy, which ranks candidates within the same Pareto set by iteratively choosing the one with the greatest Euclidean distance from the previously selected candidates.

## L  Additional experiments with MOO methods

In addition to the results presented in the paper we also run MOO methods on our benchmark for latencies and perplexity across different devices and search spaces in Figures 42-54. We present the EAFs resulting from running the baselines for multiple seeds and the hypervolume of the baselines

over the number of surrogate evaluations. Furthermore, we also present the results of running different MOO methods on the energy-perplexity objectives for different devices on GPT-L in Figures 55-59. Interestingly for these two objectives local search is often very performant at higher budgets, outperforming other baselines like NSGA-2 and EHVI.

## L.1 Experiments with 2 Objectives

We observe from Figures 42-59 that NSGA-II and EHVI are amongst the top performing methods (even at lower budgets). LS typically has a low hypervolume in the beginning, however, often outperforms other methods with enough budget.

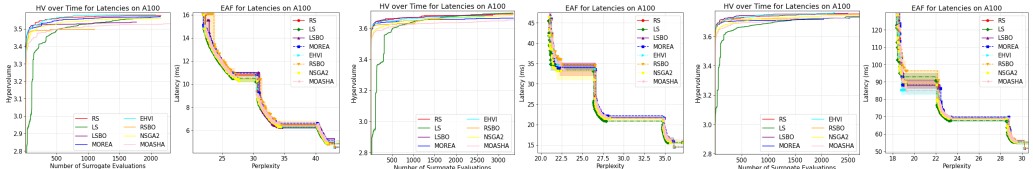

Figure 42: Pareto fronts and HV over time on A100 for GPT-S (first two), GPT-M (second two) and GPT-L (last two).

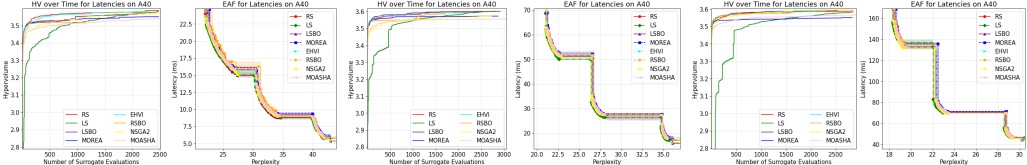

Figure 43: Pareto fronts and HV over time on A40 for GPT-S (first two), GPT-M (second two) and GPT-L (last two).

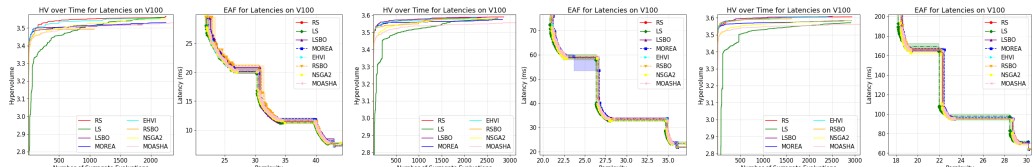

Figure 44: Pareto fronts and HV over time on V100 for GPT-S (first two), GPT-M (second two) and GPT-L (last two).

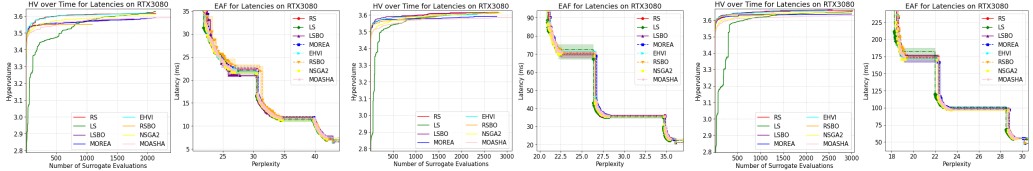

Figure 45: Pareto fronts and HV over time on RTX3080 for GPT-S (first two), GPT-M (second two) and GPT-L (last two).

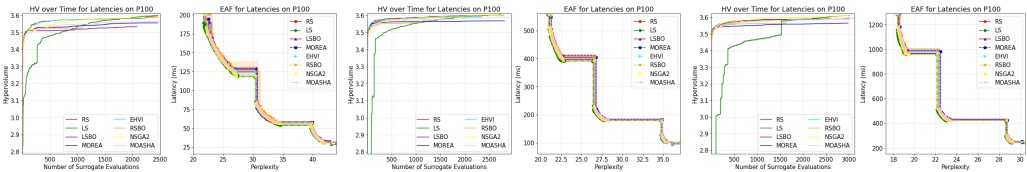

Figure 46: Pareto fronts and HV over time on P100 for GPT-S (first two), GPT-M (second two) and GPT-L (last two).

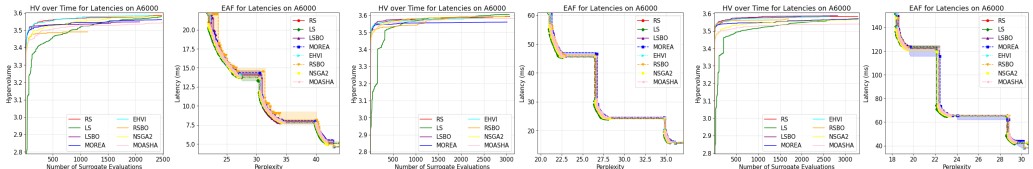

Figure 47: Pareto fronts and HV over time on A6000 for GPT-S (first two), GPT-M (second two) and GPT-L (last two).

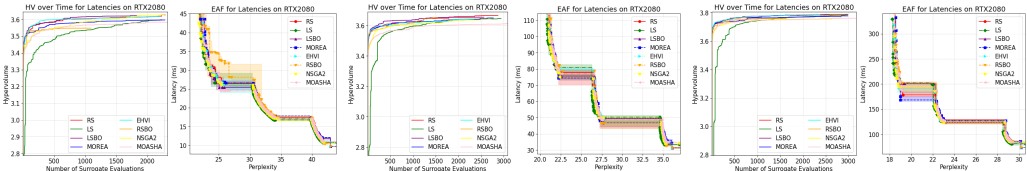

Figure 48: Pareto fronts and HV over time on RTX2080 for GPT-S (first two), GPT-M (second two) and GPT-L (last two).

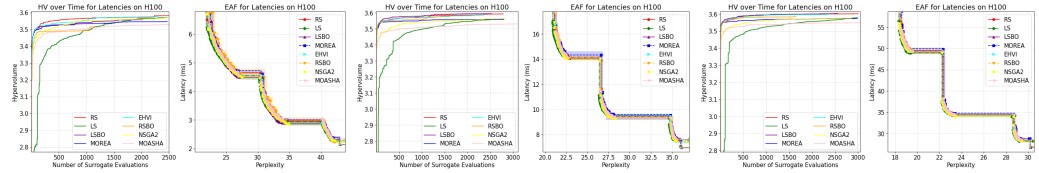

Figure 49: Pareto fronts and HV over time on H100 for GPT-S (first two), GPT-M (second two) and GPT-L (last two).

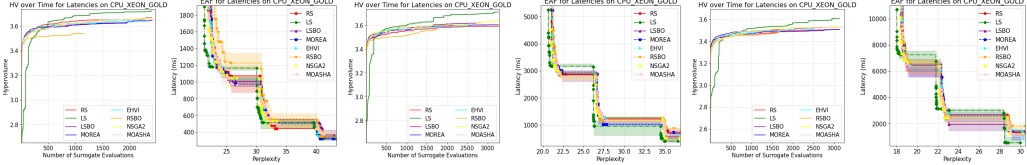

Figure 50: Pareto fronts and HV over time on CPU Xeon Gold for GPT-S (first two), GPT-M (second two) and GPT-L (last two).

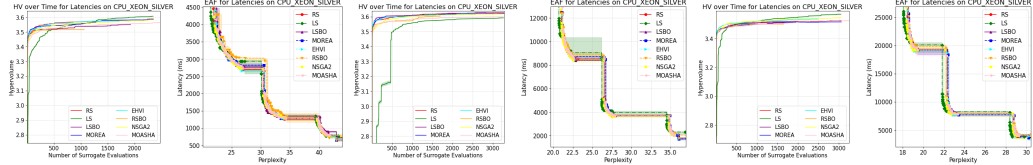

Figure 51: Pareto fronts and HV over time on CPU Xeon Silver for GPT-S (first two), GPT-M (second two) and GPT-L (last two).

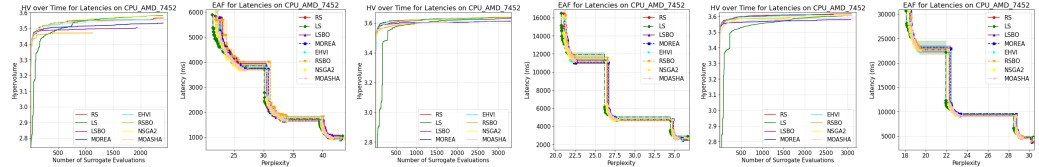

Figure 52: Pareto fronts and HV over time on CPU AMD 7452 for GPT-S (first two), GPT-M (second two) and GPT-L (last two).

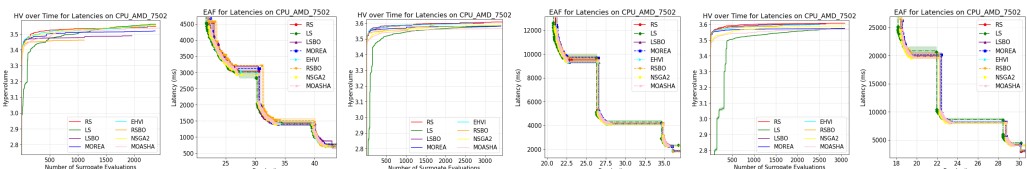

Figure 53: Pareto fronts and HV over time on CPU AMD 7402 for GPT-S (first two), GPT-M (second two) and GPT-L (last two).

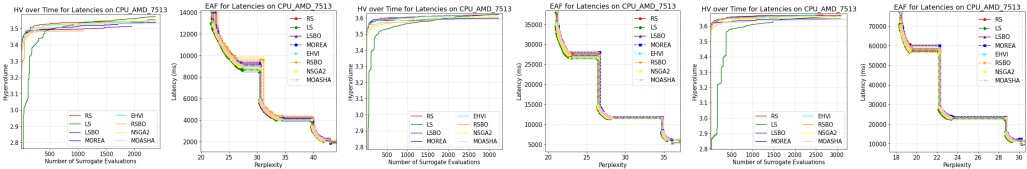

Figure 54: Pareto fronts and HV over time on CPU AMD 7513 for GPT-S (first two), GPT-M (second two) and GPT-L (last two).

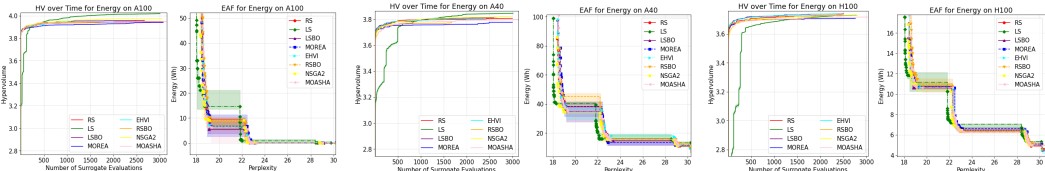

Figure 55: Pareto Fronts and HV on A100 (first 2), A40 (second 2) and H100 (last 2) for GPT-L

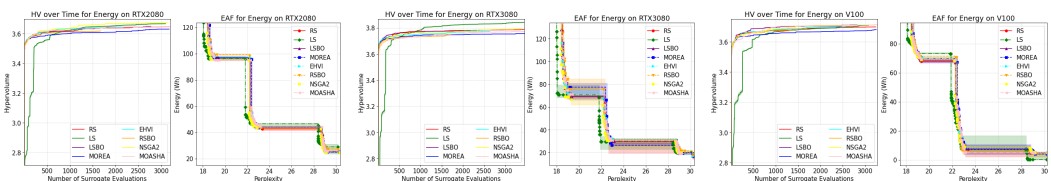

Figure 56: Pareto Fronts and HV on RX2080 (first 2), RTX3080 (second 2) and V100 (last 2) for GPT-L

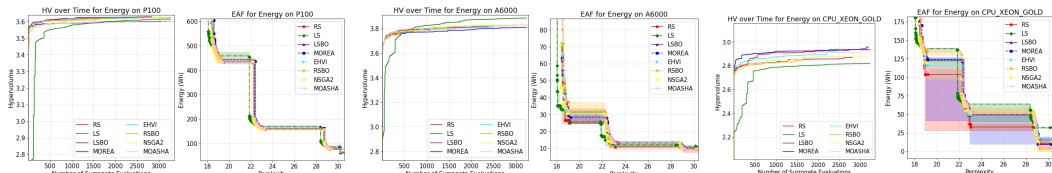

Figure 57: Pareto Fronts and HV on P100 (first 2), A6000 (second 2) and Xeon Gold CPU (last 2) for GPT-L

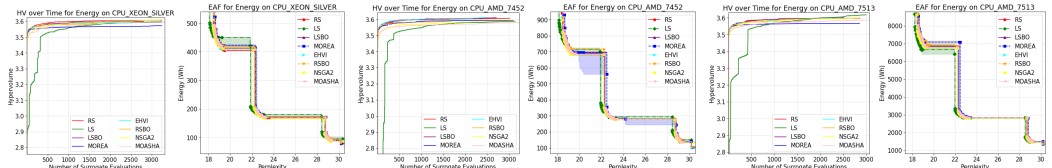

Figure 58: Pareto Fronts and HV on Xeon Silver (first 2), AMD 7452 (second 2) and AMD 7513 (last 2) for GPT-L

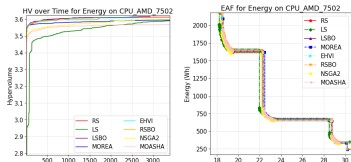

Figure 59: Pareto Fronts and HV on AMD 7502 for GPT-L.

## L.2 Experiments with 3 Objectives

In addition, we also use our benchmark to optimize energies and latencies, in conjunction with perplexity, on different devices for the GPT-L space, as presented in Figure 60 and Figure 61. We run these experiments for a smaller time budget of 3 hours using SyneTune. We observe that at smaller time budgets random search and MOO methods based on Bayesian optimization are the top methods.

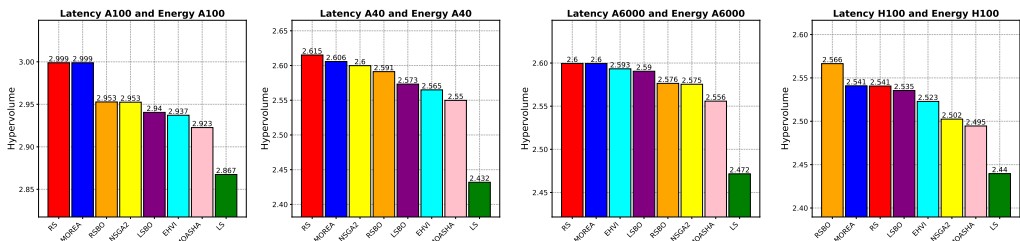

Figure 60: Hypervolumes of baselines optimizing for perplexity, latency and energy usage on A100, A40, A6000 and H100.

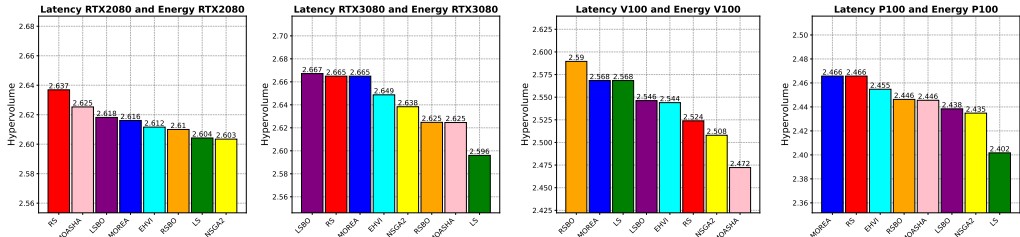

Figure 61: Hypervolumes of baselines optimizing for perplexity, latency and energy usage on RTX2080, RTX3080, V100 and P100.

## M  Correlations between different metrics

Figures 62,63, 64 show the Kendall-$\tau$ rank correlation coefficient across all the metrics supported in HW-GPT-Bench. Given two sets of $n$ observations $\{y_i\}_{i=1}^{n}$ and $\{z_i\}_{i=1}^{n}$, the $\tau$ coefficient is computed as:

$$\tau = \frac{C - D}{\frac{1}{2}n(n-1)},$$

where $C$ and $D$ are the number of concordant and discordant pairs, respectively. For every pair $(i, j)$ where $1 \leq i < j \leq n$:

1. A pair is **concordant** if the order of both elements in the pair is the same in both datasets: $(y_i < y_j$ and $z_i < z_j)$ or $(y_i > y_j$ and $z_i > z_j)$.

2. A pair is **discordant** if the order of the elements in the pair is different in the two datasets: $(y_i < y_j$ and $z_i > z_j)$ or $(y_i > y_j$ and $z_i < z_j)$.

To compute these values, we use the same 10k ($n = 10000$) ground truth observations that we use to train the surrogate models. For metrics that contain multiple observations, such as latency and energy usage, we use the median value. For easier visualization, we stratify the aforementioned correlation plots by metrics relevant to GPUs (Figures 68,69, 70) and CPUs (Figures 65,66, 67).

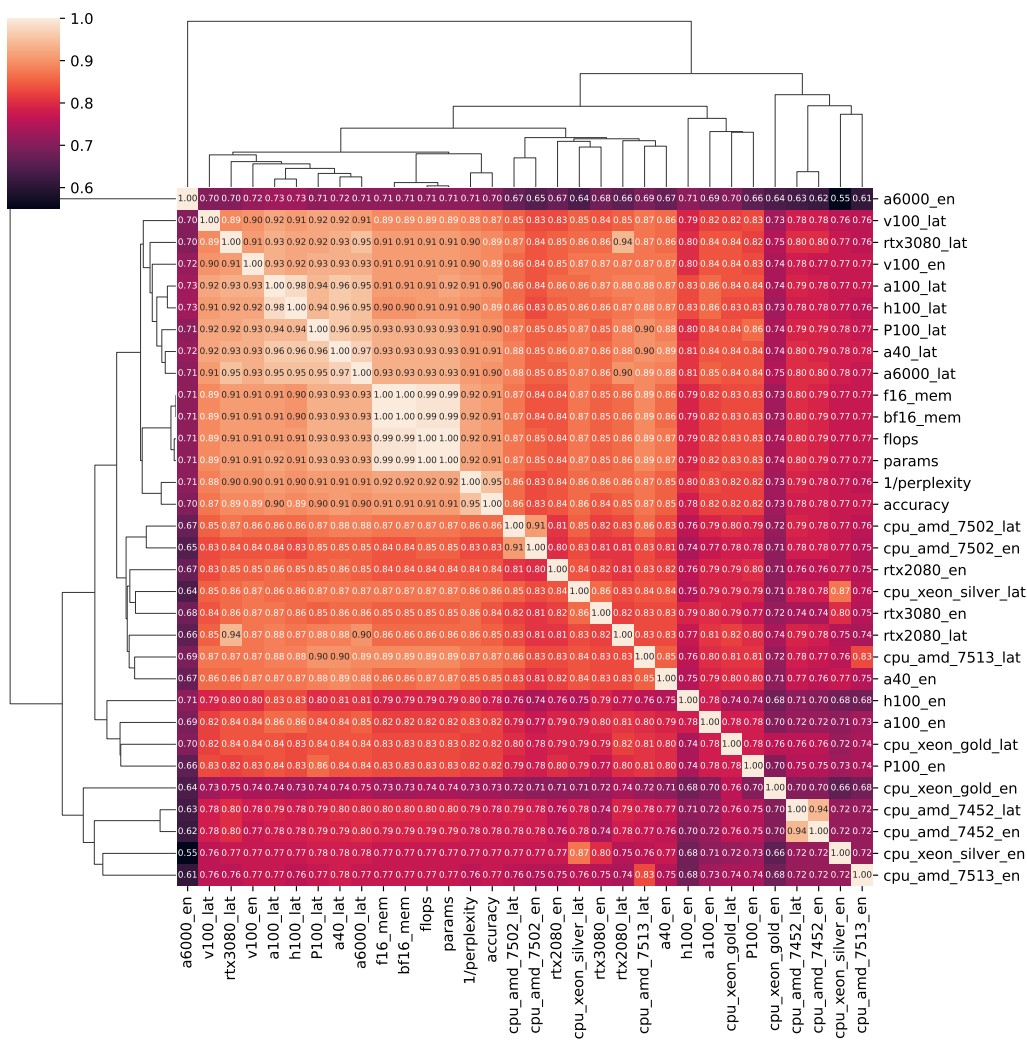

Figure 62: Cross-Metric kendall-tau correlation plots for GPT-S

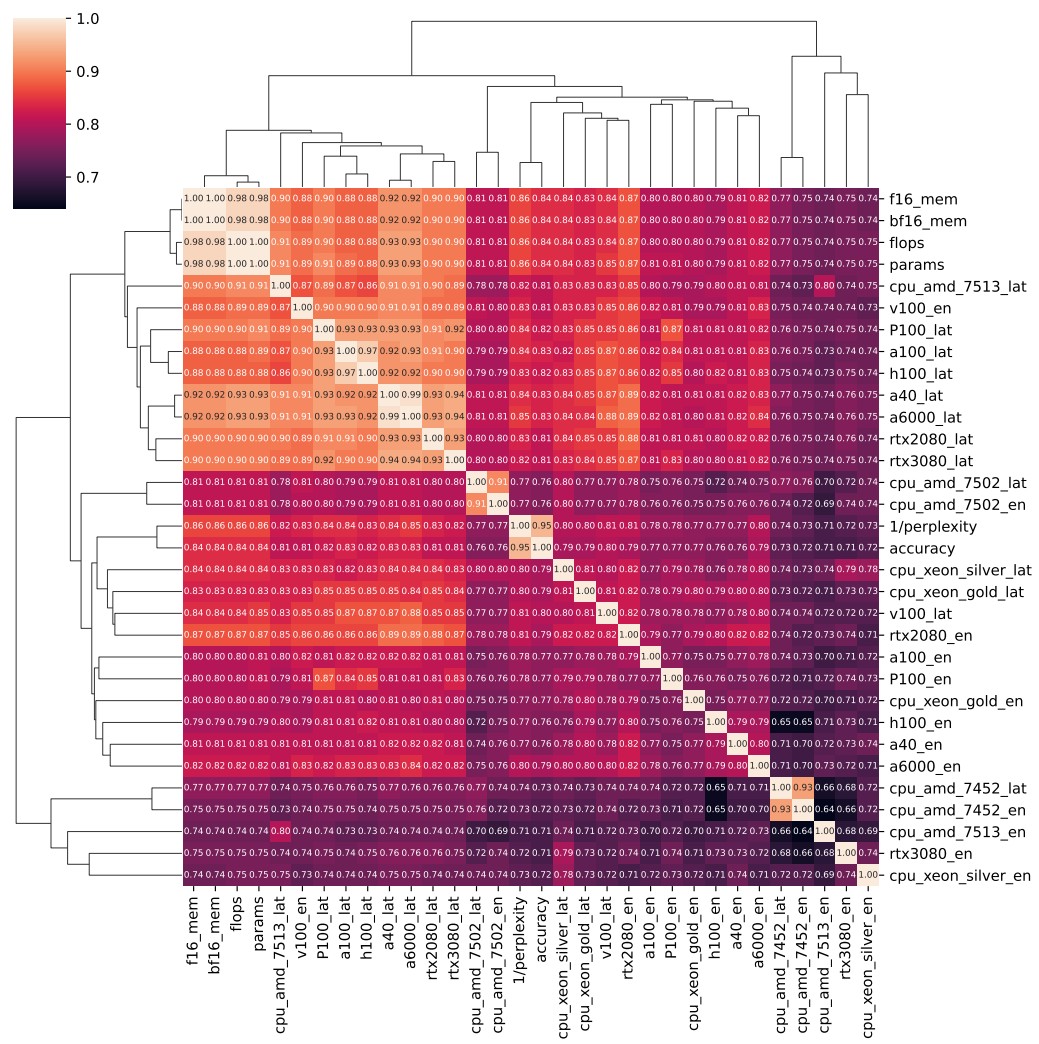

Figure 63: Cross-metric Kendall-$\tau$ correlation plots for GPT-M.

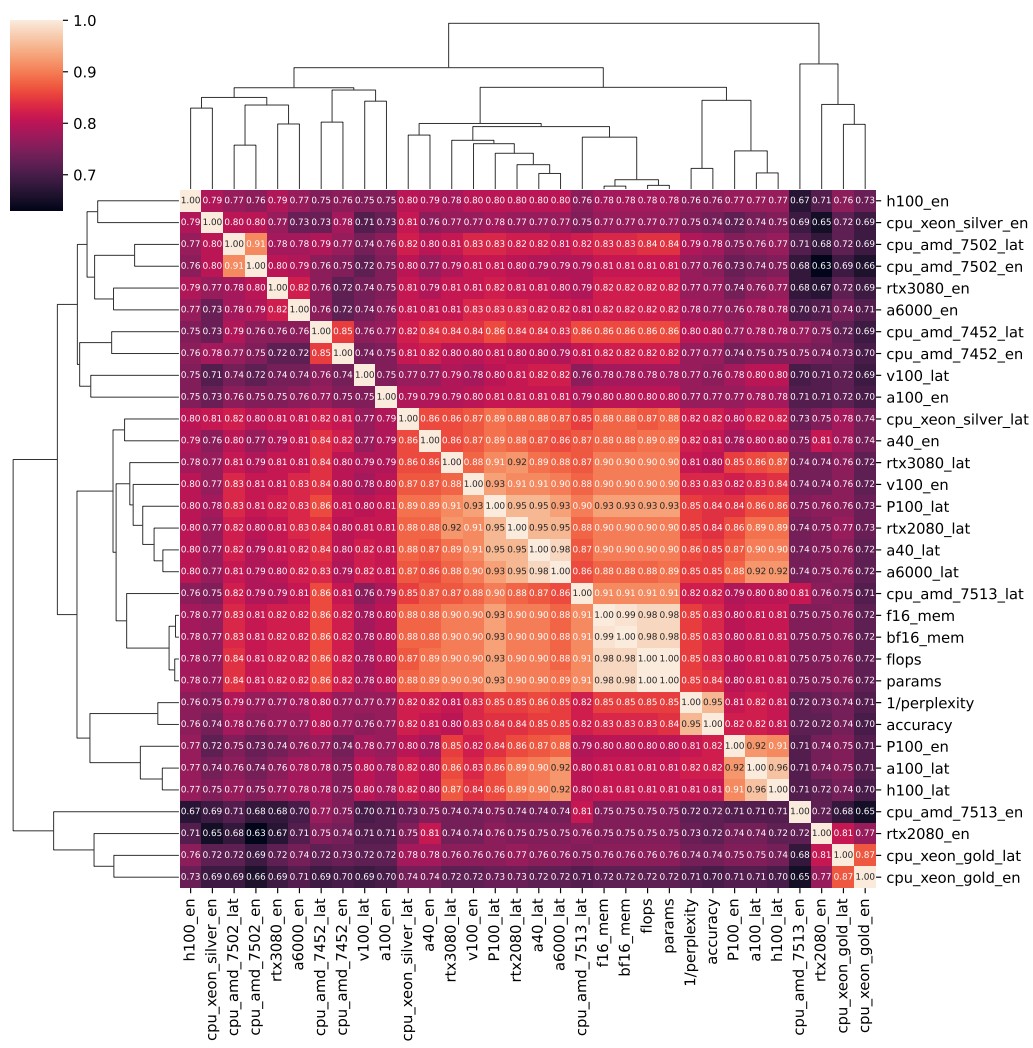

Figure 64: Cross-metric Kendall-$\tau$ correlation plots for GPT-L.

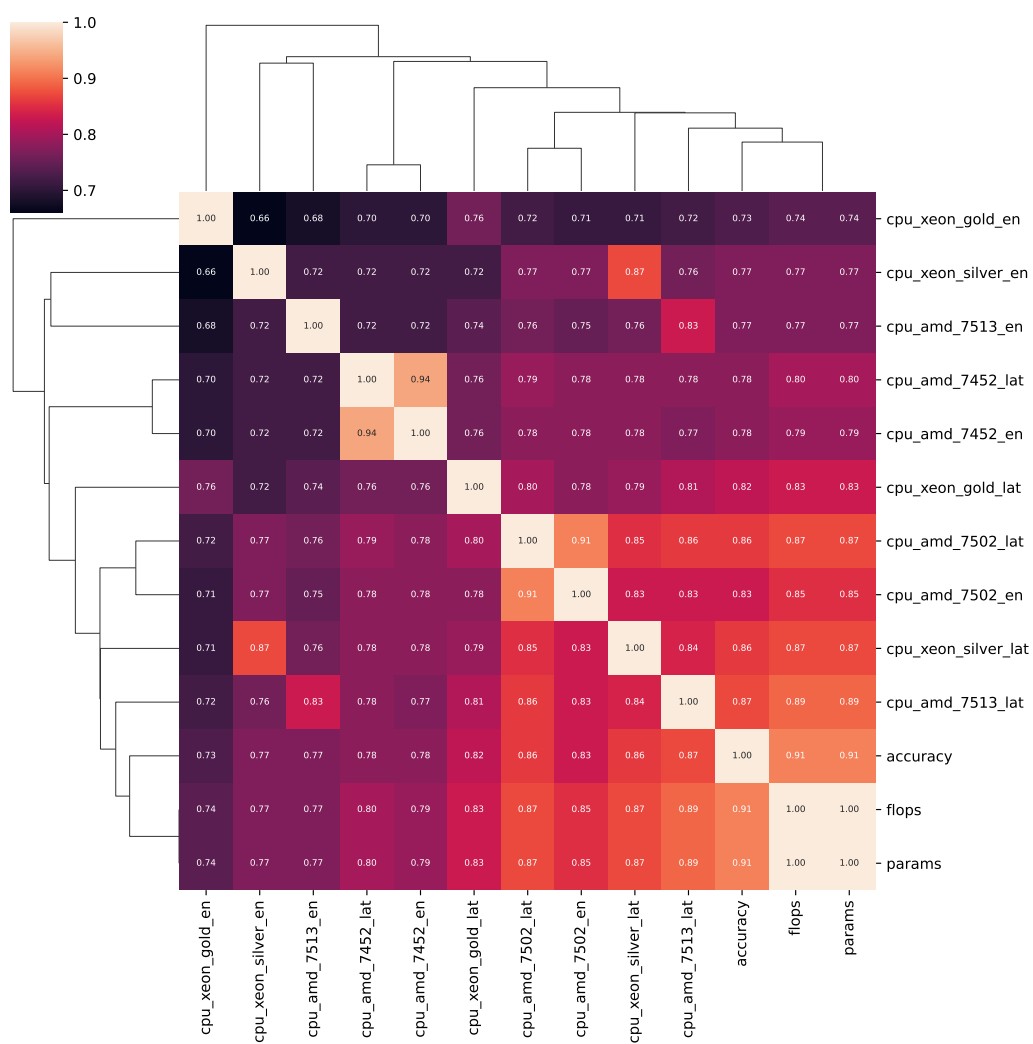

Figure 65: Cross-metric Kendall-$\tau$ correlation plots for GPT-S (only CPU devices).

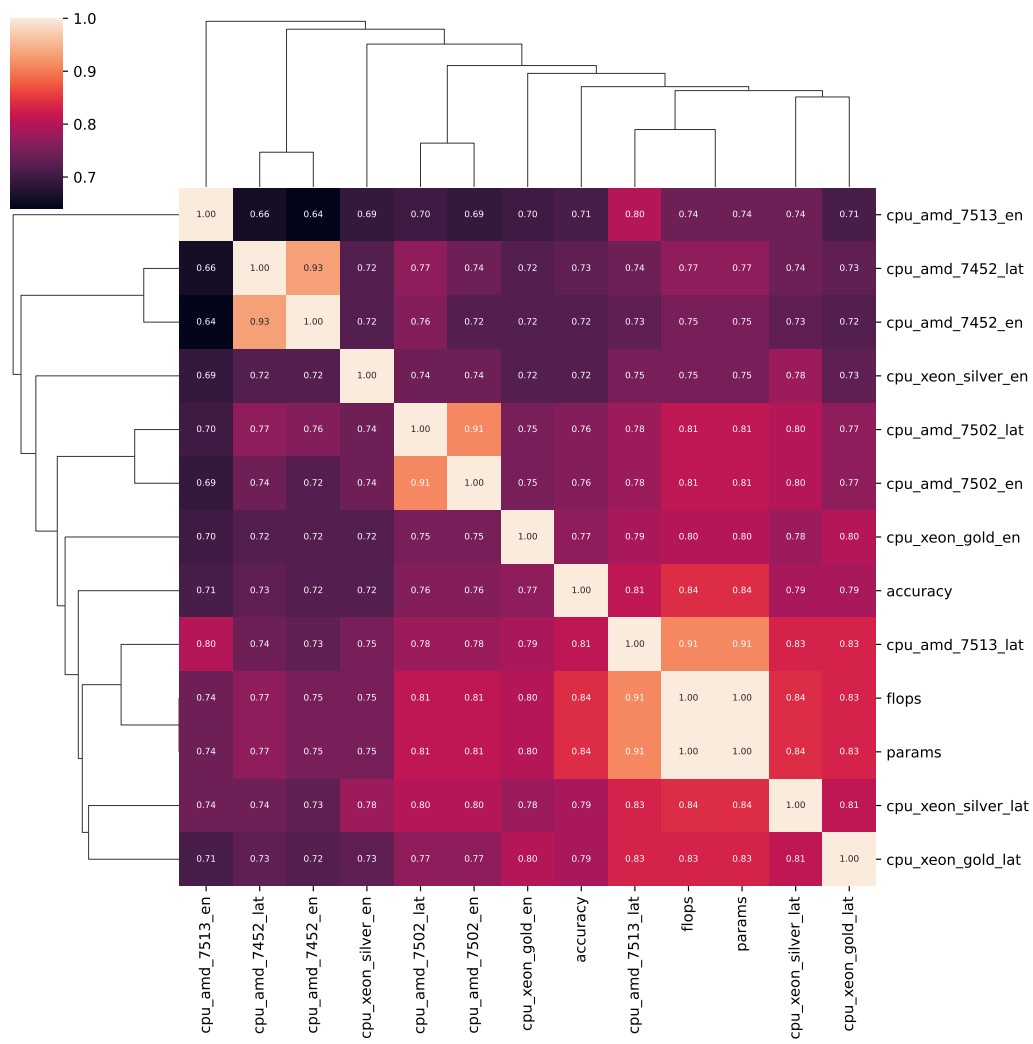

Figure 66: Cross-metric Kendall-$\tau$ correlation plots for GPT-M (only CPU devices).

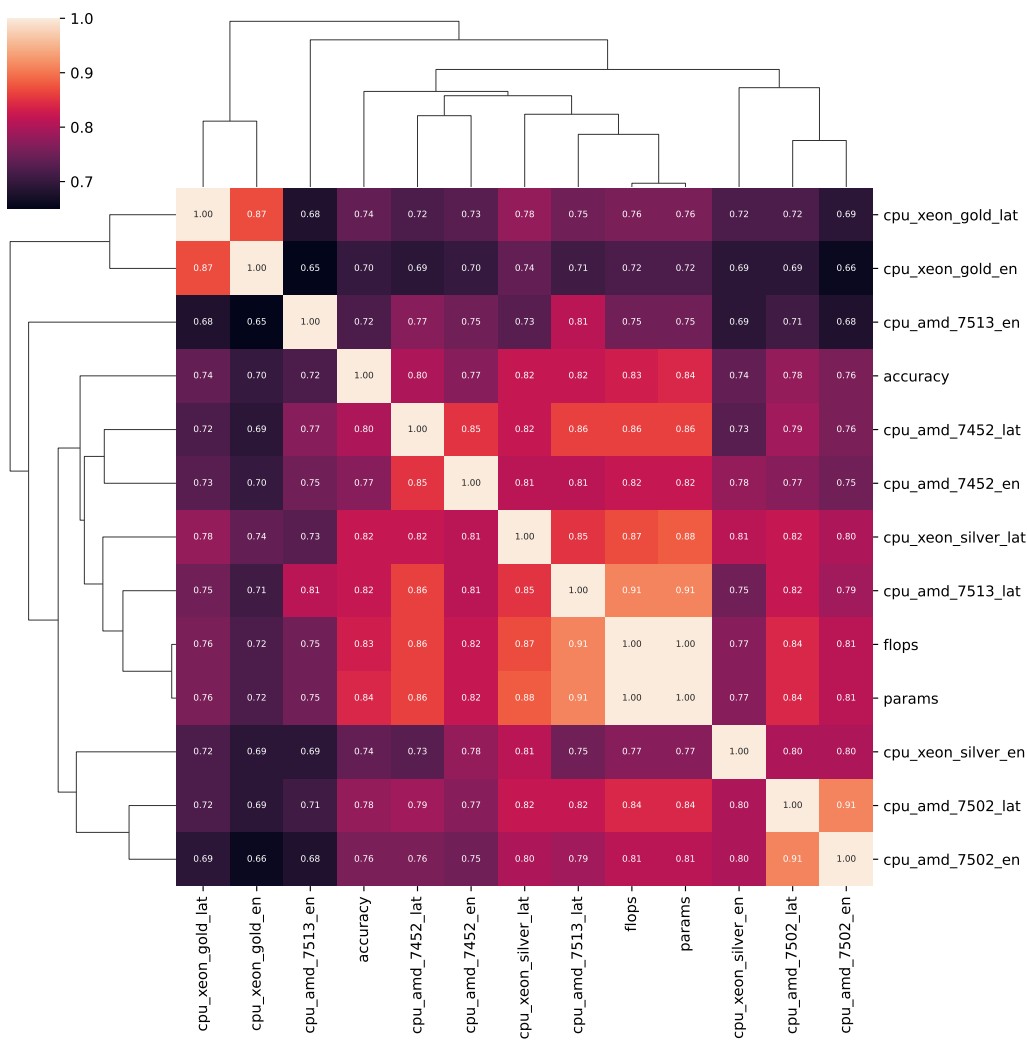

Figure 67: Cross-metric Kendall-$\tau$ correlation plots for GPT-L (only CPU device).

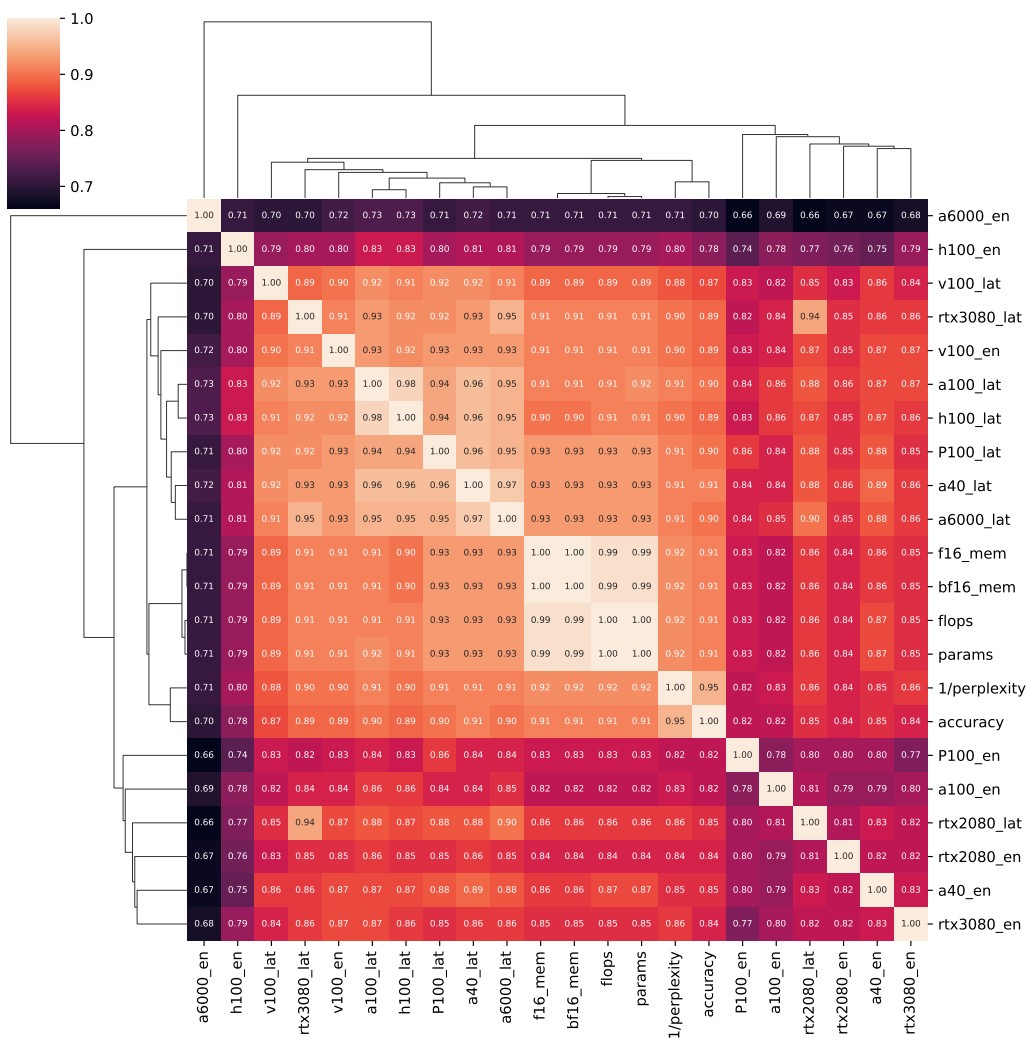

Figure 68: Cross-metric Kendall-$\tau$ correlation plots for GPT-S (only GPU devices).

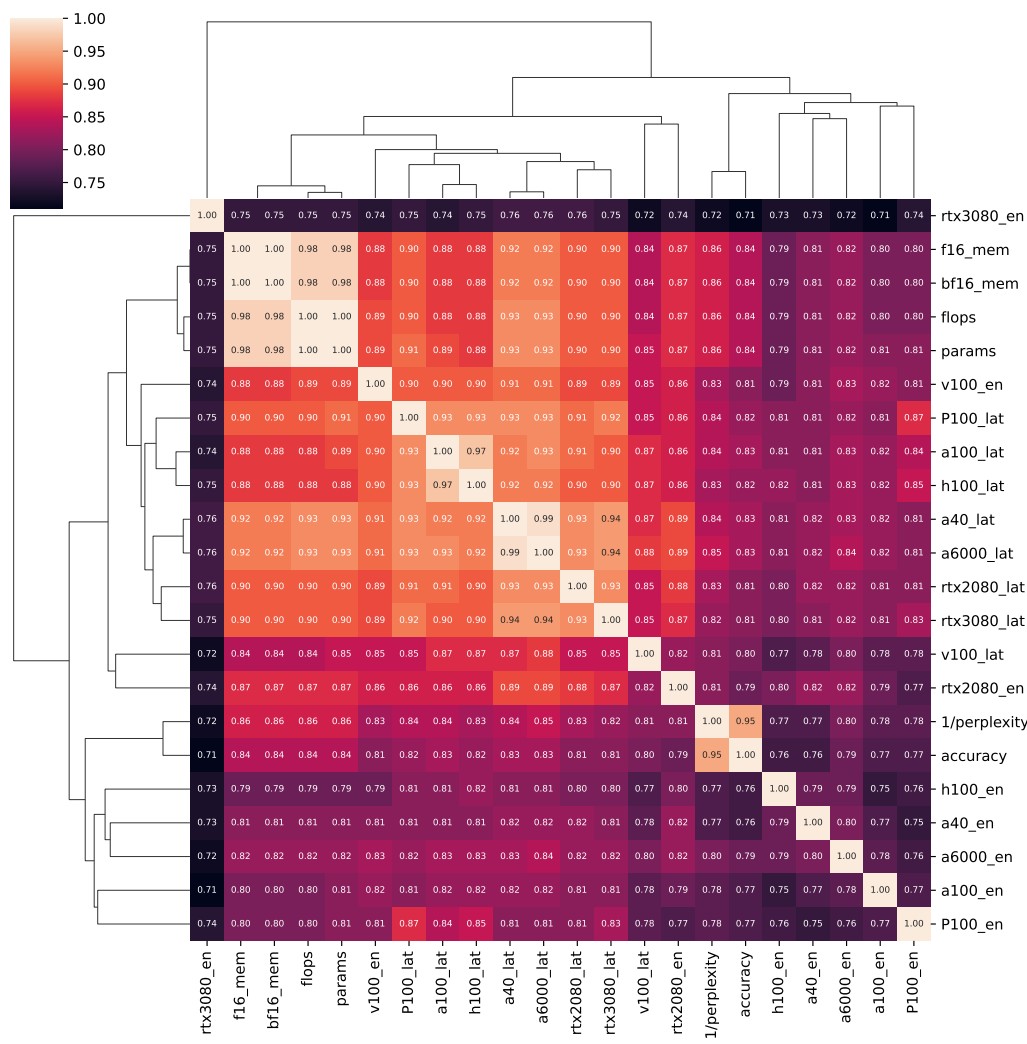

Figure 69: Cross-metric Kendall-$\tau$ correlation plots for GPT-M (only GPU devices).

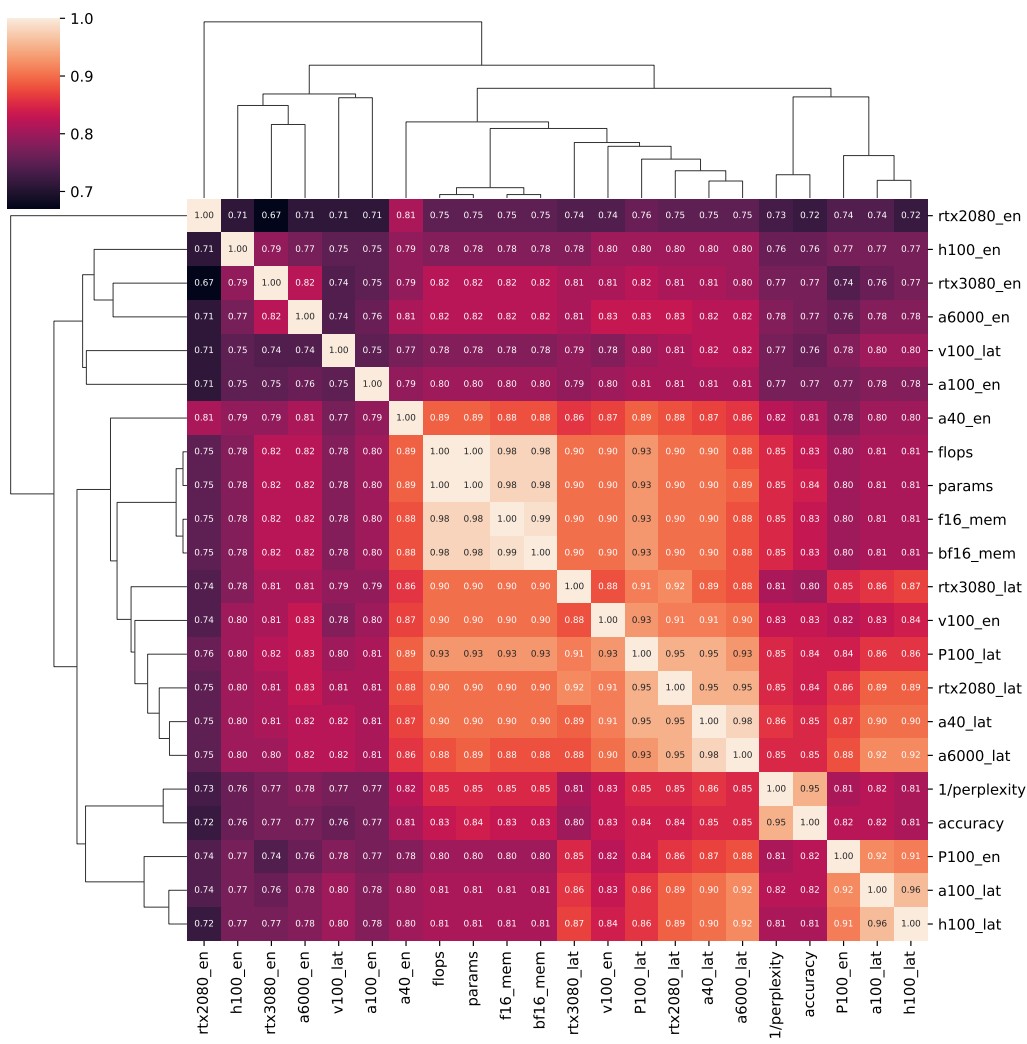

Figure 70: Cross-metric Kendall-$\tau$ correlation plots for GPT-L (only GPU devices).

# N  Additional ECDF plots

In this section, we present the ECDF plots of perplexity on different search space scales, computed using the 10k ground truth observations from the supernetwork. The largest set, $C$, contains all architectures for a fixed embedding dimension size. $B$, a subset of $C$, contains all architectures that, in addition to the fixed embedding dimension $e$, have the number of layers set to the largest possible value $l = l_3$, namely $12,\ 24,\ $ and $36$ for GPT-S, -M and -L, respectively. $A$, a subset of $B$, contains all architectures that, in addition to the number of layers set to largest possible values, have the average MLP ratio and number of heads greater than a fixed threshold. We show results for all 3 Transformer scales: GPT-S, -M and -L, in Figures 71, 72 and 73, respectively.

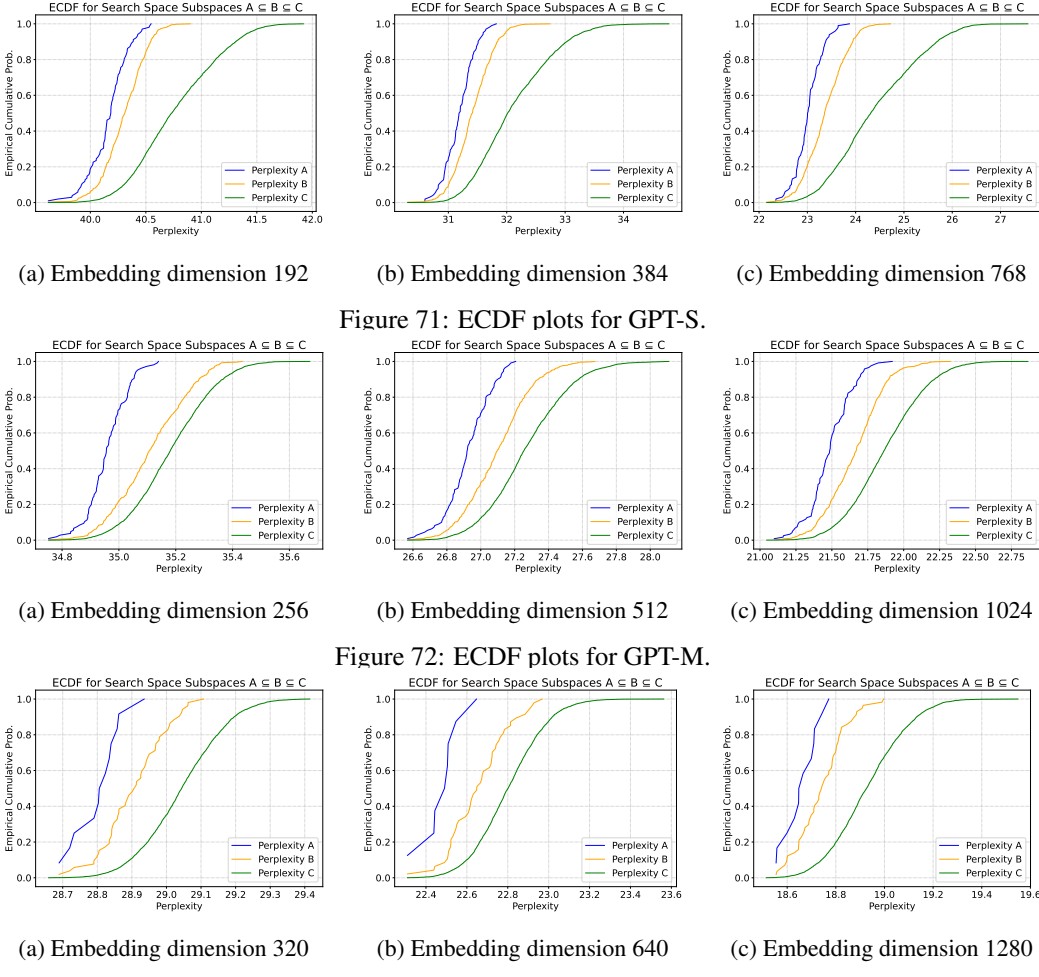

(a) Embedding dimension 192    (b) Embedding dimension 384    (c) Embedding dimension 768

Figure 71: ECDF plots for GPT-S.

(a) Embedding dimension 256    (b) Embedding dimension 512    (c) Embedding dimension 1024

Figure 72: ECDF plots for GPT-M.

(a) Embedding dimension 320    (b) Embedding dimension 640    (c) Embedding dimension 1280

Figure 73: ECDF plots for GPT-L.

# O  HW-GPT-Bench API Examples

```python
from hwgpt.api import HWGPT
api = HWGPT(search_space="m") # initialize API for GPT-M
random_arch = api.sample_arch() # sample random arch
api.set_arch(random_arch) # set  arch
flops = api.query(metric="flops") # query flops for the architecture
params = api.query(metric="params") # query params for the architecture
float16_memory = api.query(metric="f16mem") # query float16 memory for the architecture
bfloat16_memory = api.query(metric="bf16mem") # query bfloat16 memory for the architecture
```

Snippet 2: Hardware agnostic metric using the HW-GPT-Bench API.

```
from hwgpt.api import HWGPT
api = HWGPT(search_space="m") # initialize API for GPT-M
random_arch = api.sample_arch_gt() # sample random arch from amongst the 10k ground truth archs
api.set_arch(random_arch) # set  arch
results = api.query(gt=True) # get all ground truth results for the architecture
energy = api.query(metric="energy",gt=True)  # get ground truth energy observations for all
    architectures
rtx2080 = api.query(device="rtx2080",gt=True) # get all hw metrics for rtx2080 device
```

Snippet 3: Ground truth observations using the HW-GPT-Bench API.

```
from hwgpt.api import HWGPT
api = HWGPT(search_space="m") # initialize API for GPT-M
nsga2_results_2d = api.run_baseline(method="nsga2", device="h100", metrics=["energy","perplexity"],
    ppl_predictor="mlp") # nsga-2d
nsga2_results_3d = api.run_baseline_3d(method="nsga2", device="h100", metrics=["energy","perplexity","
    latency"],ppl_predictor="mlp") #nsga-3d
```

Snippet 4: Running MOO with 3 objectives using the HW-GPT-Bench API.

# P  HW-GPT-Bench Release and Maintainance

HW-GPT-Bench will be distributed under the Apache 2.0 License, tailored explicitly for academic research. The Apache 2.0 License is chosen for its permissive characteristics within the open-source community, permitting users to freely utilize, modify, and distribute the software under the condition of proper attribution and adherence to Apache 2.0 stipulations. This licensing strategy is pivotal in fostering broad adoption of HW-GPT-Bench.

In addition to its release, we are committed to fostering community engagement with the benchmark. We will actively monitor and respond to inquiries, issues, and suggestions related to HW-GPT-Bench, thereby cultivating a collaborative environment conducive to ongoing improvement and innovation.

Looking forward, our development roadmap includes plans to expand HW-GPT-Bench to encompass a wider range of devices and language model spaces. This expansion aims to bolster the benchmark's utility and relevance, accommodating emerging research demands and technological advancements in the field

# Q  Limitations and Future Work

While our work is the first one to efficiently benchmark different decoder-only architectures on a variety of gpu and cpu devices, there are several possible enhancements possible, which we leave to future work. Firstly, currently the benchmark is limited to decoder only models and we believe it would be interesting to extend to encoder-decoder models and state-space models. Secondly, currently the benchmark trains supernetworks from scratch and scaling to very large models (eg: Llama 3.1 405b), would require expensive retraining. Initializing from pretrained models and exploring parameter-efficient finetuning methods for supernet finetuning, is important to avoid retraining and make most efficient use of available compute. Furthermore, since the benchmark is developed primarily in an academic setting, we couldn't profile the architectures on edge devices and specifically edge devices which are optimized for LLM inference. However, provide a plug and play framework by releasing all our profiling scripts and hope for community contributions to enhance the benchmark for newer hardware devices.

