# OpenReview forum: "HW-GPT-Bench: Hardware-Aware Architecture Benchmark for Language Models"
_NeurIPS.cc/2024/Datasets_and_Benchmarks_Track — NeurIPS 2024 Track Datasets and Benchmarks Poster_

### Official Review · Reviewer_oM4X · 2024-07-16
**Comprehensive Analysis of a Surrogate-Based Benchmark for Optimizing GPT-2 Configurations Across Desktop Hardware**

**Rating:** 6
**Confidence:** 4

**Review:**

**Evaluation of the Quality**

The experiments are well conducted, with a robust methodology ensuring reliable and reproducible results. Data points are collected by running multiple iterations, which helps in capturing noise and uncertainty, enhancing the reliability of the results. The statistical methods used are appropriate and clearly presented, ensuring that the analysis is sound. However, the scope of hardware devices included could be expanded to cover mobile and embedded devices, and NPUs.

**Clarity**

The paper is well-structured, with a logical flow of ideas. Figures and tables are well-labeled, contributing to the clarity of the presentation. The surrogate predictors' performance is clearly explained, and the trade-offs between energy, latency, and perplexity are well-illustrated. The clear presentation makes the complex analysis accessible to the reader.

**Originality and Significance**

The methodology is not highly novel, as it originates from NAS-Bench-301 and similar works that generate tabular benchmarks using surrogate models. The paper does not highlight significant methodological differences from previous work. However, the benchmark is relevant to the broader research community for identifying optimal model configurations for various hardware platforms.

**Pros**
* Clear presentation
* Sound methodology
* Strong experimental results

**Cons**
* Relevant benchmark for the research community
* Limited novelty in methodology
* Insufficient diversity of hardware devices included in the benchmark

**Strengths:**

* Clear presentation
* Sound methodology
* Strong experimental results
* Relevant benchmark for the research community

**Additional Feedback:**

* How do you plan to address the current limitations in terms of the diversity of hardware devices in future work?

**Clarity:**

The paper is clear and well-organized, with well-labeled figures and tables. The README in the open-source repository is concise and provides a useful API with examples, aiding in the reproducibility of the benchmark.

**Correctness:**

* The experiments are well conducted, with a robust methodology ensuring reliable and reproducible results.
* Data points are collected by running multiple iterations capturing noise and uncertainty.
* The statistical methods used are appropriate and clearly presented.

**Documentation:**

The documentation is clear and well-organized. The README in the open-source repository is concise and provides an API with examples, making it easy for users to understand and utilize the benchmark.

**Ethics:**

No ethics concern.

**Limitations:**

The authors have adequately addressed the limitations and potential negative societal impact of their work in Section 6.

**Opportunities For Improvement:**

* Expanding the range of hardware devices included in the benchmark, particularly in mobile, embedded spaces, and NPUs.
* Highlighting the methodological differences from existing surrogate benchmarks would strengthen the paper's originality.
* Addressing the redundancy in references (reference 81 and 82 being the same paper) is also necessary.

**Relation To Prior Work:**

The paper references existing works like NAS-Bench-301 but does not sufficiently highlight the differences in methodologies. More emphasis on how this work advances or differs from previous surrogate benchmarks would be beneficial.

**Summary And Contributions:**

The paper presents a benchmark using surrogate predictors of perplexity, latency, memory usage, and energy consumption to analyze optimal model configurations for different hardware devices. This benchmark targets GPT-2-like models on 7 desktop GPUs and 5 desktop CPUs. It is designed to evaluate multi-objective optimization algorithms efficiently. The experiments conducted are rigorous, with data points collected through multiple runs to capture noise and uncertainty. The paper demonstrates convincing trade-offs between energy, latency, and perplexity, and the surrogates' performance is validated in terms of errors and correlations.

---

> ### Author Rebuttal · Authors · 2024-08-17
>
> Thank you for your thorough review of our work. We are encouraged by your positive feedback on the unique aspects of our work, including efficient benchmarking, rigorous experiments, reproducible results, and modeling uncertainties. We’re pleased that you found the benchmark and paper accessible and understandable. Below, we address each of your questions. Please also refer to the "General Response" for answers to common questions raised by reviewers:
>
> > **Expanding range of hardware devices and future work**
>
> Please see our response to this in the “General Response”.
>
> > **Differences and comparison with NB301**
>
> Given the immense size of the search spaces, training architectures from scratch is impractical. Our work is inspired by surrogate benchmarks like those proposed by [1], but HW-GPT-Bench has several key differences from NB301:
> - Unlike NB301, which trains architectures from scratch—a method impractical for larger models and datasets—we use an efficient weight-sharing supernet. The performance of the inherited subnetwork serves as a reliable performance proxy and can be fine-tuned if needed. We also use a sandwich scheme [2] to train the supernet by sampling the largest, smallest, and a set of random architectures.
> - While NB301 uses the DARTS pre-training pipeline, we introduce three novel search spaces and a training pipeline specifically designed for supernet training, focusing on language modeling instead of NB301's primary focus on image classification.
> - Unlike NB301, our work supports a range of hardware devices and provides well-calibrated latency predictions, enabling both single-objective and multi-objective NAS evaluations. To our knowledge, we are the first to study the calibration and uncertainty estimates of surrogate models for latency prediction.
>
> These differences have been added to the appendix of the paper, which you can find in the code repository link of this submission.
>
> Thank you again for thoroughly reviewing our work. If you have any follow-up questions, we’re happy to engage with you during the discussion period.
>
> *--References--*
>
> [1] Zela et al. Surrogate NAS Benchmarks: Going Beyond the Limited Search Spaces of Tabular NAS Benchmarks. In ICLR 2022
>
> [2] Yu et al. Slimmable neural networks. In ICLR 2019

---

> ### Comment · Reviewer_oM4X · 2024-08-28
>
> Thank you for your response. I am keeping my score as is.

---

### Official Review · Reviewer_r3pB · 2024-07-24

**Rating:** 8
**Confidence:** 4
**Correctness:** Yes looks correct.
**Clarity:** Yes, the paper is relatively easy to …

**Review:**

Strengths

* Exhaustive evaluation of language models across performance and hardware metrics.

* Use of surrogate models to predict performance across 13 devices and three different model sizes.

* Application of multi-objective optimization to search for Pareto-optimal configurations that balance model performance and hardware efficiency.

* Open-source nature. The benchmark has significant potential provided the authors offer a way to keep it updated with different hardware and model types.


Weakness

* While the principles applied can be extended to larger models, at present the benchmark focuses on GPT-2 models. While the paper addresses small, medium, and large scales, it remains to be seen how well the benchmark scales to even larger models beyond 774M parameters, such as those with billions of parameters used in state-of-the-art applications. How well do you think your benchmark scales to models with parameters beyond the 774M range, such as those with billions of parameters? Have you considered evaluating the benchmark on other types of language model architectures, beyond GPT-2?

* The surrogate models inherit weights from a pretrained supernetwork. The effectiveness and bias introduced by this approach depend heavily on the quality and diversity of the pretraining process, which may not be fully representative of all possible configurations.

* The results are based on evaluations across specific hardware devices. Variations in hardware configurations, software environments, and usage patterns could lead to different performance outcomes, limiting the generalizability of the results to other hardware setups. Given the rapid evolution of hardware, how do you plan to keep HW-GPT-Bench up-to-date with new devices and configurations?

**Strengths:**

Please see the strengths in the review above.

**Additional Feedback:**

Please address the questions in the review above.

**Documentation:**

Yes, the provided anonymous GitHub link works. The documentation seems adequate.

**Ethics:**

No ethical issues found.

**Limitations:**

Please see the comments in the weakness section of the review. I feel the authors can address these concerns.

**Opportunities For Improvement:**

Please see the questions in the weakness section above. I feel this work can be extended to other language models beyond 774 million parameters. I also feel community engagement would be key to keep this benchmark updated with more models and new hardware configurations.

**Relation To Prior Work:**

Yes.

**Summary And Contributions:**

The authors introduce HW-GPT-Bench, a hardware-aware benchmark designed to optimize the configurations for large language models training across various hardware metrics and devices. By using surrogate models to predict performance (perplexity) and hardware metrics (latency, energy consumption, memory usage), the benchmark enables efficient exploration of multi-objective Neural Architecture Search (NAS) methods. Key contributions include establishing a benchmark with different model scales, providing calibrated predictions for hardware metrics, and evaluating various optimization algorithms to identify Pareto-optimal configurations.

---

> ### Author Rebuttal · Authors · 2024-08-17
>
> Thank you for your encouraging comments and positive score on our work. We appreciate your recognition of key aspects of our benchmark, such as different model scales, calibrated predictions for hardware metrics, and the evaluation of various multi-objective algorithms. Below, we address each of your questions. Please also refer to the “General Response” for answers to common questions raised by reviewers:
>
> > **Application to larger scales**
>
> Please see our response to this in the “General Response.”
>
> > **The surrogate models inherit weights from a pretrained supernetwork**
>
> Thank you for raising this important point. Previous works, such as HAT [1] (Table 5), AutoFormer [2] (Table 4), and Bender et al. [3] (Figure 5), have shown high correlations between the performance of architectures with inherited weights from a supernet and those trained from scratch. We expect similar results for HW-GPT-Bench.
>
>  > **Limiting the generalizability of the results to other hardware setups**
>
> We have open-sourced the profiling pipelines for the devices we have access to, and we anticipate minimal modifications will be needed when profiling on newer devices. As we work primarily in an academic setting, our access is limited to certain CPU and GPU devices. However, we welcome pull requests from the community to add results on new devices, and we will periodically update the benchmark as we gain access to more hardware.
>
> Thank you again for carefully reviewing our work. We look forward to updating you with new results as they become available. If you have any follow-up questions, we are happy to engage with you during the discussion period.
>
> *--References--*
>
> [1] Wang et al. Hat: Hardware-aware transformers for efficient natural language processing. In ACL 2020
>
> [2] Chen et al. Autoformer: Searching transformers for visual recognition. In ICCV 2021
>
> [3] Bender et al. Understanding and simplifying one-shot architecture search. In ICML 2018

---

> > ### Comment · Reviewer_r3pB · 2024-08-30
> > **Response to the authors**
> >
> > Thank you for your response. I hope the authors will continue to update this dataset with more hardware setups while actively engaging with the community. I will keep my score to 8.

---

### Official Review · Reviewer_6da8 · 2024-07-24
**Review of #199 HW-GPT-Bench**

**Rating:** 6
**Confidence:** 4
**Correctness:** The claims made in the submission see…
**Clarity:** The writing is good.

**Review:**

Strengths

- HW-GPT-Bench introduces a hardware-aware benchmark for language models, which is a valuable contribution to the field of neural architecture search (NAS) for NLP.
- The authors provide an open-source API for latency and perplexity predictors, supernetwork weights, and different baselines, facilitating easy integration of new methods into the benchmark.
- The benchmark assesses trade-offs across multiple crucial hardware metrics, including latency, energy consumption, GPU memory usage, and performance.

Weaknesses

- The difference between weight-sharing NAS (Once for all[1] and HAT[2]) and converting pretrained model to supernet method should be emphasized. Recently, some methods like LLaMA-NAS[3], LoNAS[4] and Shears [5] have made progress in this field and should be acknowledged.
- Benchmark Relevance in the Context of Scaling Laws:
As per the scaling law, the bigger the language model is, the better the performance is. There is evidence from LiteTransformerSearch[6] showing a strong correlation between performance and the size of the decoder. The paper should justify the relevance of this benchmark given the consensus on scaling laws.
- Rationale for Multiple Search Spaces:
In Figure 3, the perplexity should be continuous but there is an obvious gap forming three clusters related to the three search spaces -S, -M and -L. The necessity of designing these three search spaces and rebuilding supernets for -S and -M when -L could potentially cover them needs clarification.
- Generalizability to Larger Language Models:
The paper trains three supernets from scratch and summarizes insights using proxies like AutoGlun. It should address whether these insights can be applied to larger language models.
- Comprehensive Factors in Scaling Laws:
The proposed scaling law about the search space might ignore an important factor - data or the number of tokens. This should be addressed.
- Diversity of Evaluation Datasets:
The paper primarily uses the OpenWebText 2 dataset for training and evaluation. This limited scope may not fully capture the generalizability of the benchmark across different types of datasets and real-world applications.
- Long-term Relevance and Scalability:
The scalability of the proposed benchmark for significantly larger models or different architecture types is not thoroughly discussed. Given the rapid evolution of language models, the benchmark's applicability to future models should be addressed.
- Alignment of Model Scale and Hardware Focus:
As the model scale is limited to 774M, the target of this benchmark should focus on edge devices including mobile devices and Jetson etc. The evaluation of performance on powerful GPUs, which can even contain 7B LLaMA, seems misaligned.
- Typo:
The reference in line 18 is missing and should be added.

[1] Cai, Han et al. “Once for All: Train One Network and Specialize it for Efficient Deployment.” *ArXiv* abs/1908.09791 (2019): n. pag.

[2] Wang, Hanrui et al. “HAT: Hardware-Aware Transformers for Efficient Natural Language Processing.” *Annual Meeting of the Association for Computational Linguistics* (2020).

[3] Sarah, Anthony et al. “LLaMA-NAS: Efficient Neural Architecture Search for Large Language Models.” *ArXiv* abs/2405.18377 (2024): n. pag.

[4] Muñoz, J. Pablo et al. “LoNAS: Elastic Low-Rank Adapters for Efficient Large Language Models.” *International Conference on Language Resources and Evaluation* (2024).

[5] Muñoz, J. Pablo et al. “Shears: Unstructured Sparsity with Neural Low-rank Adapter Search.” *ArXiv* abs/2404.10934 (2024): n. pag.

[6] Javaheripi, Mojan et al. “LiteTransformerSearch: Training-free On-device Search for Efficient Autoregressive Language Models.” *ArXiv* abs/2203.02094 (2022): n. pag.

**Strengths:**

- HW-GPT-Bench introduces a hardware-aware benchmark for language models, which is a valuable contribution to the field of neural architecture search (NAS) for NLP.
- The authors provide an open-source API for latency and perplexity predictors, supernetwork weights, and different baselines, facilitating easy integration of new methods into the benchmark.
- The benchmark assesses trade-offs across multiple crucial hardware metrics, including latency, energy consumption, GPU memory usage, and performance.

**Additional Feedback:**

Nil.

**Documentation:**

Yes.

**Ethics:**

No.

**Limitations:**

Yes.

**Opportunities For Improvement:**

Please refer to the above weaknesses.

**Relation To Prior Work:**

Some related works are missing. Please refer to the weaknesses.

**Summary And Contributions:**

This paper introduces HwGPT-Bench, a hardware-aware benchmark for language models. It uses surrogate predictions to approximate hardware metrics across 13 devices in the GPT-2 family, with models up to 774M parameters.

The benchmark models the noise in energy and latency measurements and employs weight-sharing techniques from Neural Architecture Search to estimate perplexity. It demonstrates its utility by quickly simulating optimization trajectories for multi-objective optimization algorithms.

---

> ### Author Rebuttal · Authors · 2024-08-17
>
> Thank you for your comprehensive feedback. We're encouraged by your recognition of the value our contribution brings to Neural Architecture Search for NLP. We also appreciate your acknowledgment of our easily integrable open-source API, model weights, and thorough analysis across multiple hardware devices. Below, we address each of your concerns. Please refer to the "General Response" for common questions raised by reviewers:
>
> > **Rationale for Multiple Search Spaces**
>
> We chose three different search spaces to account for variability in subnetwork scales and architecture dimensions, as shown in Table 1 of the paper. Our initial prototyping revealed that including too many choices (e.g., embedding dimensions) in a single search space (e.g., GPT-L) causes the validation loss to saturate early, likely due to gradient interference, which worsens in larger, fine-grained spaces. Instead of merging multiple choices into one supernet, we opted to train supernets at three different scales, allowing for more variability in parameter size and perplexity while reducing interference during training.
>
> > **Three clusters and a gaps in the Pareto-front**
>
> Thank you for pointing this out. We clarify that Figure 3 in the paper represents a single search space, GPT-L, with three different embedding dimensions (corresponding to the three clusters). This highlights the importance of the embedding dimension in transformer spaces. We have now extended HW-GPT-Bench to include more uniform strides across all architecture dimensions, aiming for a more continuous Pareto front. Please refer to Figure 3 and Table 1 in the rebuttal PDF for updated results. As shown, the Pareto front is more continuous. Additionally, multiple architectures with similar perplexity have significantly different latencies, underscoring the importance of appropriate parameter allocation across architecture dimensions.
>
> > **Diversity of Evaluation Datasets**
>
> Thank you for the recommendation to evaluate on diverse benchmarks. We now provide a unified API to lm-eval-harness for evaluating supernets at different scales on various downstream datasets in a few-shot manner. Please see Figure 3 for an initial scatter plot showing perplexity vs. performance on a downstream task (ARC-easy). While perplexity generally guides downstream accuracy, improvements in perplexity don't always translate to accuracy gains. As we expand the benchmark, we plan to include other downstream tasks and evaluations on larger scales.
>
> > **LiteTransformerSearch [6] showing a strong correlation and neural scaling laws...**
>
> Thank you for bringing this work to our attention. The initial scaling law papers [1, 2] focused on compute-optimal training, which differs significantly from our focus on inference efficiency, such as energy usage and latency. Moreover, scaling laws often overlook on-device latency and energy usage, which are central to our benchmark. While scaling laws suggest a strong correlation between parameter size and performance, they typically don't account for how different architectural parameters contribute to model size. For example, different methods to achieve a target parameter size (e.g., increasing depth, embedding dimension, intermediate dimension, number of heads) aren't considered. It's crucial to allocate parameter size wisely across architecture dimensions.
>
> In the three new spaces introduced (see Table 1 and Figure 4), we observe that similar perplexities can lead to vastly different latencies and energy consumption. This highlights the importance of appropriately distributing the transformer parameter budget across architecture dimensions.
>
> > **The difference between weight-sharing NAS and pretrained model to supernet …**
>
> Thank you for raising this important point. There are two primary approaches to supernet training: training from scratch and warmstarting from a pretrained model. While the latter is computationally efficient, it introduces various confounding factors into supernet training. The impact of warmstarting on performance metrics like perplexity, compared to training from scratch, is not well-studied, making it a relevant area for future research. For this benchmark, we chose to restrict ourselves to models and scales that we could train from scratch.
>
> > **Long-term Relevance and Scalability**
>
> We agree that long-term relevance and scalability of HW-GPT-Bench are crucial, especially with the increasing number of open-sourced language models. We list our long-term goals for the benchmark below:
> - We are currently developing a library [whittle](https://github.com/whittle-org/whittle) based upon [litgpt](https://github.com/Lightning-AI/litgpt) to support weight-sharing based NAS for different types of decoder-only transformers and also different training and fine-tuning schemes, which allows us to expand the benchmark to more recent architectures.
> - Establishing mechanisms for community contributions of new supernets and multi-objective baselines on an open-access MO-NAS leaderboard as the benchmark and library mature.
> - Seamlessly integrating few-shot evaluation of sub-networks on different downstream tasks through *whittle*.
>
> > **Missing citations**
>
> Thank you for bringing these recent works to our attention. We have updated the paper to include citations to these references.
>
> Thank you again for taking time to carefully review our work. We also look forward to updating you with the newer results, when they become available. If you have any followup questions we are happy to engage with you during the discussion period.
>
> --References--
>
> [1] Kaplan et al. Scaling laws for neural language models. arXiv preprint arXiv:2001.08361.
>
> [2] Hoffmann et al. Training compute-optimal large language models. In NeurIPS 2022

---

> > ### Comment · Reviewer_6da8 · 2024-08-28
> >
> > Thanks for the rebuttal. After reading your responses and other review comments, I'd like to update my rating to 6.

---

### Official Review · Reviewer_74Qi · 2024-08-01
**Interesting topic with extensive results**

**Rating:** 9
**Confidence:** 4
**Correctness:** The benchmark evaluation is appropria…
**Clarity:** yes

**Review:**

The topic of benchmarking hardware-aware architecture for language models is interesting, and important for its broader applicability. Latency and energy consumption on multiple hardware device types are profiled and reported. A power law model is used to model the collected perplexity data. And open-source code and apis are available.

**Strengths:**

1. Extensive experimental results are provided with open-source code and apis available.
2. Measurements are conducted with multiple evaluations to account for uncertainty.
3. Multiple hardware device types are evaluated.

**Additional Feedback:**

N/A

**Documentation:**

yes

**Opportunities For Improvement:**

1. Please explain why the specific form of power law in Line 253 is used. Is there any citation for it?
2. The authors mention that potential limitations are discussed (Line 590) but I can't find it in the paper.
3. Considering that a hardware device (e.g., CPU) may have other current tasks running, how would it interfere with the latency and energy consumption metrics?


Some minor points:

1. A few typos: Line 18, it seems missing citations []
2. Font size in Figure 1 and 4 are too small to read.

**Relation To Prior Work:**

yes

**Summary And Contributions:**

This paper proposes HW-GPT-Bench, a hardware-aware surrogate benchmark for evaluating language models across various hardware devices, metrics, and scales on a single CPU in just a few seconds. Surrogate predictions are used to approximate various hardware metrics. Heteroscedastic noise inherent in the energy and latency measurements are modeled.

---

> ### Author Rebuttal · Authors · 2024-08-17
>
> Thank you for your positive and encouraging feedback on our work. We are pleased that you find our research interesting and valuable for its broader applicability. We appreciate your recognition of our efforts in testing across multiple hardware devices, accounting for uncertainty, and conducting extensive analyses and experiments to validate our findings. We are also grateful that you highlighted our open-source code, which we believe promotes transparency and enables further community development of the benchmark. Below, we respond to each of your questions. Please also refer to the “General Response” for answers to common questions raised by reviewers.
>
> > **Explain power law form **
>
> Please see our clarification on the power law form in the “General Response”.
>
> > **Considering that a hardware device (e.g., CPU) may have other current tasks running, how would it interfere with the latency and energy consumption metrics?**
>
> We allocate a fixed number of dedicated CPU cores to avoid interference from other tasks. Additionally, we simulate multiple latency observations to capture variability and ensure well-calibrated latency predictions. However, as you noted, CPU observations can indeed be noisier compared to those from a GPU. Please refer to Figures 1 and 2 in the rebuttal PDF for the distribution of latencies for a single architecture on both GPU and CPU devices.
>
> > **The authors mention that potential limitations are discussed (Line 590) but I can't find it in the paper.**
>
> Thank you for pointing this out. While we included the broader impact in Section 6, we missed adding the limitations. This has now been corrected in the updated paper, which is available in the repository along with the API code. For your reference, the limitations are:
>
> “*While our work is the first one to efficiently benchmark different decoder-only architectures on a variety of GPU and CPU devices, there are several possible enhancements possible, which we leave to future work. Firstly, the current benchmark is limited to decoder-only models and we believe it would be interesting to extend to encoder-decoder models [1] and state-space models [2]. Secondly, in HW-GPT-Bench we train supernetworks from scratch and scaling to very large models (e.g.: Llama 3.1 405b), would require expensive retraining. Fine-tuning pretrained models  and exploring parameter-efficient finetuning methods [3] for supernet finetuning, in order to avoid retraining and make most efficient use of available compute time, is an important future direction. Furthermore, since the benchmark is developed primarily in an academic setting, we couldn’t profile the architectures on edge devices and specifically edge devices which are optimized for LLM inference. However, since our API provides a plug-and-play framework, by releasing all our profiling scripts, we hope for community contributions to enhance the benchmark with other hardware devices.*”
>
> > **Missing citations on line 18 and increasing font size in Figures 1 and 4 to improve readability.**
>
> Thank you for your careful review. We have fixed these issues and uploaded the updated PDF to the repository linked in the paper.
>
> Thank you again for taking the time to review our work. If you have any follow-up questions, we are happy to engage with you during the discussion period.
>
> *-- References --*
>
> [1] Raffel et al. Exploring the limits of transfer learning with a unified text-to-text transformer. JMLR 2020
>
> [2] Gu and Dao. Mamba: Linear-time sequence modeling with selective state spaces. 2023, arXiv preprint arXiv:2312.00752.
>
> [3] Pfeiffer et al. Modular deep learning. 2023, arXiv preprint arXiv:2302.11529.

---

### Author Rebuttal · Authors · 2024-08-17

We thank the reviewers for their time, positive scores, and valuable feedback, which helped us significantly improve our paper. We appreciate that the reviewers found our work broadly applicable (**74Qi**) and our contributions valuable (**6da8**). We are pleased that they recognized several positive aspects of our work, such as exhaustive experimentation and reproducibility (**r3pB24**,**74Qi**,**oM4X**), our open-source API (**74Qi**, **6da8**), calibrated latency/energy modeling (**74Qi**, **r3pB24**, **oM4X**), and clear, accessible writing (**oM4X**). Below, we address some common questions raised by the reviewers:

> **Extension of the benchmark to larger scales**

Thank you for highlighting this important point. In response, we are conducting experiments to extend the benchmark to GPT-XL scale (1.55B parameters) and profiling on various GPU devices. We will include these larger-scale results during the rebuttal and discussion period. Our training is massively parallelized across multiple nodes and GPUs, so we foresee no issues with further scaling, although we lack the compute resources to train larger models from scratch. In the future, we plan to start from pretrained models [4,5,6,7] to extend the benchmark, but for this version, we trained all our supernets from scratch to avoid the confounding effects of pretrained initialization.

> **Extension to include embedded devices**

We currently evaluate the benchmark on 5 different CPUs and 8 GPU types. We agree that profiling on more devices would enhance the benchmark. If you know of any available simulators, we would be happy to use them, as our well-designed interface makes integration trivial. Unfortunately, as academics, we lack access to embedded devices. In the future, we welcome pull requests from the community to add results on new devices and will periodically update the benchmark with newer devices as we gain access to them.

> **Relation to Scaling Laws**

The power law presented in line 253 was chosen because it empirically models the data well. This equation models perplexity as a product of several dimensions that affect transformer parameter size (embedding size, number of layers, MLP expansion ratio, and number of heads), using an ordinary least squares fit on collected perplexity pairs. Our primary goal is to study the importance of different transformer dimensions for perplexity prediction. We also find a *strong* correlation between the performance modeled by this equation and our perplexity predictor (an MLP) on a set of held-out architectures, as shown in the table below, justifying the assumed functional form.

| Search Space | Correlation between Power Law and Perplexity Predictor |
|:------------:|:------------------------------------------------------:|
|     GPT-S    |                        0.9270                          |
|     GPT-M    |                        0.87893                         |
|     GPT-L    |                        0.87819                         |

We would like to clarify that this functional form is independent of the scaling laws proposed in the literature [1,2], which typically study model size but not the various dimensions contributing to model performance. Additionally, we allocate a fixed compute time for each supernet, updating multiple architectures simultaneously. Thus, all architectures in a space (e.g., GPT-S) are trained for the same compute time, and compute is not factored into our functional form. Exploring compute-optimal scaling laws for masked supernet training of language models with the sandwich update rule [3] is an intriguing and important direction for future work.

We thank the reviewers for their careful assessment of our work and look forward to updating them with new results during the discussion period.

*-- References --*

[1] Kaplan et al. Scaling laws for neural language models. arXiv preprint arXiv:2001.08361.

[2] Hoffmann et al. Training compute-optimal large language models. In NeurIPS 2022

[3] Yu et al. Universally slimmable networks and improved training techniques. In ICCV 2019.

[4] Klein et al. Structural pruning of pre-trained language models via neural architecture search. arXiv preprint arXiv:2405.02267, 2024.

[5] Cai et al. Flextron: Many-in-one flexible large language model. arXiv preprint arXiv:2406.10260, 2024.

[6] Muñoz et al. Shears: Unstructured sparsity with neural low-rank adapter search. arXiv preprint arXiv:2404.10934, 2024.

[7] Muñoz et al. Lonas: Elastic low-rank adapters for efficient large language models. In LREC-COLING 2024.

---

### Author Rebuttal · Authors · 2024-08-30

We once again thank all the reviewers for their feedback on our work, which helped us improve our benchmark significantly and also scale it up further. As the discussion period closes soon, we would like to update the reviewers with some new experiments and results:

> "Scaling up to larger models"

We have now added GPT-XL, a 1.55B parameter model, to the existing suite of 6 models in HW-GPT-Bench. Please find the plots on statistics collected from the GPT-XL supernet in Figure 1 of the new rebuttal pdf and the corresponding search space in Table 1. Interestingly, we again observe the variety of latency trace-offs for architectures with similar perplexities in Figure 1.

> "Reliability of Supernet Proxies"

Reviewer r3pB raised an interesting point that since the surrogate models inherit weights from a pretrained supernetwork, their reliability is crucial. Hence we randomly sample 100 architectures from the pretrained supernet and finetune them further for 5000 steps. As shown in Table 2 of the new rebuttal pdf, we observe a very strong rank correlation (**>0.9**) between the perplexities of the inherited and finetuned architectures, showing that the ranking between the inherited and finetuned architectures is largely preserved. This further shows the reliability of our supernet surrogate.

> "Additional Covariate Analysis and Feature Importance"

We would like to update the reviewers that we have performed some additional covariate analysis on the data collected for the GPT-S-wide, GPT-M-wide and GPT-L-wide supernets we introduced during the rebuttal period. Please refer to the power law fits in the new rebuttal pdf. Once again, we observe that the "embedding-dimension" or width of the transformer network is a crucial architectural choice in determining perplexities and latencies, despite the more uniform strides in GPT-wide spaces.

We will add the additional search spaces introduced and results from the rebuttal period to the updated version of our paper. We thank the reviewers for engaging with us during the discussion period and we are happy to address any remaining questions they may have. We are also committed to improving the benchmark further by adding newer devices, simulators and scaling up to larger models using different finetuning strategies.

---

### Decision · Program_Chairs · 2024-09-26

**Decision:**

Accept (Poster)

**Comment:**

In this paper, the authors focus on the problem of computational costs in LLMs, and then propose a hardware-aware benchmark (HW-GPT-Bench)  on several devices by NAS.  To accelerate the evaluation of those devices, this paper uses surrogate predictors to evaluate the performance of architectures. The key idea is to follow hardware-aware NAS to build the benchmark of LLMs. The problem is clear and important, the idea is novel and interesting.

All reviewers agreed that building a benchmark focused on hardware-aware LLM benchmark is an important contribution and found the proposed HW-GPT-Bench approach as compelling. However, some reviewers raise some minor points in their review which have been addressed in the rebuttal. From my understanding, the paper is incremental, the main component of surrogate predictors is from existing hardware-aware methods.

 Exploring the reasons behind the success of these techniques and providing intuitive explanations would contribute to the overall scientific contribution of the work.

We invite the authors to take these changes into account in the camera-ready version of the paper.